JCB Journal of Cell Biology

# Metabolic adaptations of micrometastases alter EV production to generate invasive microenvironments

Michalis Gounis[1,2], America V. Campos[1,2], Engy Shokry[1], Louise Mitchell[1], Ruhi Deshmukh[1,2], Emmanuel Dornier[1], Nicholas Rooney[1], Sandeep Dhayade[1], Luis Pardo[1], Madeleine Moore[1], David Novo[1,2], Jenna Mowat[3], Craig Jamieson[3], Emily Kay[1], Sara Zanivan[1,2], Nikki R. Paul[1], Claire Mitchell[1], Colin Nixon[1], Iain Macpherson[2], Saverio Tardito[1,2,4], David Sumpton[1], Karen Blyth[1,2], Jim C. Norman[1,2], and Cassie J. Clarke[1,2]

**Altered cellular metabolism has been associated with the acquisition of invasive phenotypes during metastasis. To study this, we combined a genetically engineered mouse model of mammary carcinoma with syngeneic transplantation and primary tumor resection to generate isogenic cells from primary tumors and their corresponding lung micrometastases. Metabolic analyses indicated that micrometastatic cells increase proline production at the expense of glutathione synthesis, leading to a reduction in total glutathione levels. Micrometastatic cells also have altered sphingomyelin metabolism, leading to increased intracellular levels of specific ceramides. The combination of these metabolic adaptations alters extracellular vesicle (EV) production to render the microenvironment more permissive for invasion. Indeed, micrometastatic cells shut down Rab27-dependent production of EVs and, instead, switch on neutral sphingomyelinase-2 (nSM2)-dependent EV release. EVs released in an nSM2-dependent manner from micrometastatic cells, in turn, influence the ability of fibroblasts to deposit extracellular matrix, which promotes cancer cell invasiveness. These data provide evidence that metabolic rewiring drives invasive processes in metastasis by influencing EV release.**

## Introduction

Breast cancer can metastasize to various organs including, the bone, liver, brain, and lung. Lung metastases are observed in approximately a quarter of patients with estrogen receptor–positive metastatic breast cancer and in almost half of patients with the HER2-positive, estrogen receptor–negative subtype (Kennecke et al., 2010). Thus, lung metastases represent a major contributor to breast cancer morbidity and mortality, highlighting the need to understand the processes through which this cancer type colonizes this organ. The first step in the metastatic cascade involves breaching the basement membrane and local invasion of the surrounding stroma, followed by entry into the circulatory and/or lymphatic system and survival of tumor cells in these environments (Joyce and Pollard, 2009; Lambert et al., 2017). To metastasize to the lung, circulating cells must then extravasate into the lung parenchyma and seed small colonies (micrometastases), which must then grow to yield clinically detectable metastases. This last stage of the metastatic cascade poses a major bottleneck for development of clinically

detectable metastases. Indeed, following extravasation, the majority of disseminated cancer cells are unable to grow in distant organ environments; some studies estimate that <0.02% of disseminated tumor cells can proceed to metastatic outgrowth (Luzzi et al., 1998). This inefficiency is likely due to the vulnerability of extravasated tumor cells to elimination by the immune system, but also by challenges posed by the very different microenvironment that the tumor cells encounter there (Aguirre-Ghiso, 2007).

Successful metastasis requires that cancer cells rewire their metabolism to adapt to the various nutrient and metabolite profiles and other factors, such as oxygen tension and tissue stiffness, encountered at points on the metastatic cascade (Faubert et al., 2020). There has been considerable focus on how such metabolic adaptations may support both growth and survival of metastasizing cells. For instance, to survive oxidative insults encountered following detachment from the extracellular matrix (ECM) environment of the primary tumor site and

[1]Cancer Research UK Scotland Institute, Glasgow, UK; [2]School of Cancer Sciences, University of Glasgow, Glasgow, UK; [3]Department of Pure and Applied Chemistry, Thomas Graham Building, University of Strathclyde, Glasgow, UK; [4]Medical University of Vienna, Center for Cancer Research and Comprehensive Cancer Center, Vienna, Austria.

Correspondence to Jim C. Norman: j.norman@crukscotlandinstitute.ac.uk; Cassie J. Clarke: c.clarke@crukscotlandinstitute.ac.uk

M. Gounis's current affiliation is Turku Bioscience Centre, University of Turku and Åbo Akademi University, Turku, Finland. S. Zanivan's current affiliation is MD Anderson Cancer Center, Experimental Therapeutics, Houston, TX, USA.

upon entering the circulation, disseminating cancer cells can increase pentose phosphate pathway activity to provide reducing equivalents for glutathione synthesis (Labuschagne et al., 2019; Pilley et al., 2023). Evidence is accumulating that metabolic adaptations may support other cellular processes—such as cell migration/invasion and ECM production/remodelling—that cancer cells need to execute at various stages of the metastatic cascade. For example, stiff microenvironments increase activity of the creatine-phosphagen ATP-recycling system to power cytoskeletal dynamics during the invasive migration and chemotaxis necessary to establish liver metastases of pancreatic ductal adenocarcinoma (Papalazarou et al., 2020). Similarly related to invasive behavior, transformation of mammary epithelial cells leads to metabolic stress, which upregulates expression of the glutamate-cystine exchanger, xCT (SLC7A11), without affecting rates of glutaminolysis (Dornier et al., 2017). Upregulated xCT leads to increased extracellular glutamate—which is manifest as increased levels of circulating glutamate in tumor-bearing individuals. Increased extracellular glutamate then activates metabotropic glutamate receptor signalling to promote intracellular trafficking of the pro-invasive matrix metalloprotease, MT1-MMP.

Having negotiated successful exit from the primary tumor, survival in the circulation, extravasation, and executed the metabolic rewiring necessary for these steps of the metastatic cascade, cancer cells must still adapt to the metabolic microenvironment of the metastatic target organ before they can form clinically detectable metastases. As discussed above, mammary cancer commonly metastasizes to the lung, and its metabolic microenvironment is very different from that of the mammary gland and the circulation. For instance, pyruvate is present in higher concentrations in lung interstitial fluid than in plasma (Christen et al., 2016), and once metastasizing cells have adapted to this, this nutrient supports several cellular processes necessary for metastatic outgrowth in this tissue. Indeed, metastasizing mammary cancer cells adapt to using pyruvate: (a) to provide α-ketoglutarate (αKG) to enable hydroxylation of proline residues in collagens (Elia et al., 2019); (b) to increase activation of anabolic signalling through the mTOR pathway (Rinaldi et al., 2021); and (c) to increase pyruvate carboxylase-dependent anaplerosis (Christen et al., 2016). Thus, adaptation to increased local pyruvate levels contributes to the ability of micrometastatic cells to condition the ECM niche—likely to increase their chances of survival and immune escape—and to provide building blocks and anabolic signalling for subsequent metastatic outgrowth.

Cancer aggressiveness is thought to be influenced by the primary tumor's ability to alter the microenvironment in other organs to generate niches that render them receptive to metastatic seeding (Peinado et al., 2017). Metastatic niche priming may involve mobilization of elements of the innate immune system—such as macrophages and neutrophils—to suppress acquired immunity in metastatic sites (Jackstadt et al., 2019). Also, alterations to the ECM, which would be expected to support cancer cell survival, growth, and invasiveness, have been observed in metastatic target organs very early in metastasis and prior to the arrival and extravasation of cancer cells in those organs (Novo et al., 2018). Release of extracellular vesicles (EVs) from primary tumors is now established to assist with niche priming—often by altering ECM deposition in metastatic target organs. EVs can be generated within the endosomal system, and key components of the endosomal EV production and release pathway in cancer cells—particularly the Rab27 GTPases—are, therefore, key to priming of metastatic niches in both the liver and lung in mammary cancer (Wang et al., 2023).

Evidence is accumulating that altered metabolic landscapes can influence EV production, and this is now thought to be a mechanism through which cells may communicate information concerning the state of their metabolism to other cells. For instance, in obesity, metabolically stressed/damaged hepatocytes secrete EVs, which can communicate with liver stellate cells to promote ECM deposition, thus increasing the fibrosis associated with fatty liver (Azparren-Angulo et al., 2021). Also, mitochondrial stress in mammary cancer cells activates the PINK1 kinase, which leads to packaging of mitochondrial DNA (mtDNA) into EVs (Rabas et al., 2021). These mtDNA-containing EVs can then drive invasive behavior in other cells via activation of a toll-like receptor signalling. The metabolic rewiring occurring in disseminated cancer cells as they adapt to the microenvironment of other organs may, therefore, modulate the release of factors such as EVs that are able to generate ECM niches permitting metastatic outgrowth.

Here we have deployed a genetically engineered mouse model of mammary cancer to generate cells from very early lung micrometastases, which are isogenically paired with their corresponding primary tumor cells. This approach has enabled in-depth characterization of the metabolic adaptations made by cancer cells very early in metastatic seeding and a description of how these control the release of EVs that can influence ECM deposition to generate a microenvironment conducive to subsequent invasive growth.

## Results

### Generation and characterization of micrometastatic cells

Orthotopic transplantation models can be used to generate organ-specific isogenic metastatic cell lines to study adaptations made by breast cancer cells as they move from their primary tumor site to metastatic target organs, such as the lung (Minn et al., 2005). As we are interested in how metabolic adaptations might sculpt the early metastatic microenvironment of mammary cancer, we sought to establish cultures of cells that have recently relocated from their primary site in the mammary gland to the lung. To do this, we reintroduced cells derived from primary tumors in the mouse mammary tumor virus (MMTV)-polyoma middle T (PyMT) model of mammary cancer (maintained in the FVB mouse strain) (Campbell et al., 2021)—termed parental (P) cells—into the fourth mammary fat pad (FP) of FVB mice (Fig. 1 A). We allowed these to grow into tumors of 8–10 mm in diameter, resected them from the mammary FP, placed them into culture, and thereafter referred to these as FP cells. After resection of the primary tumor, mice were maintained for ~1 mo to allow metastases to seed. Mice were then sacrificed, and their lungs were visually inspected. Although

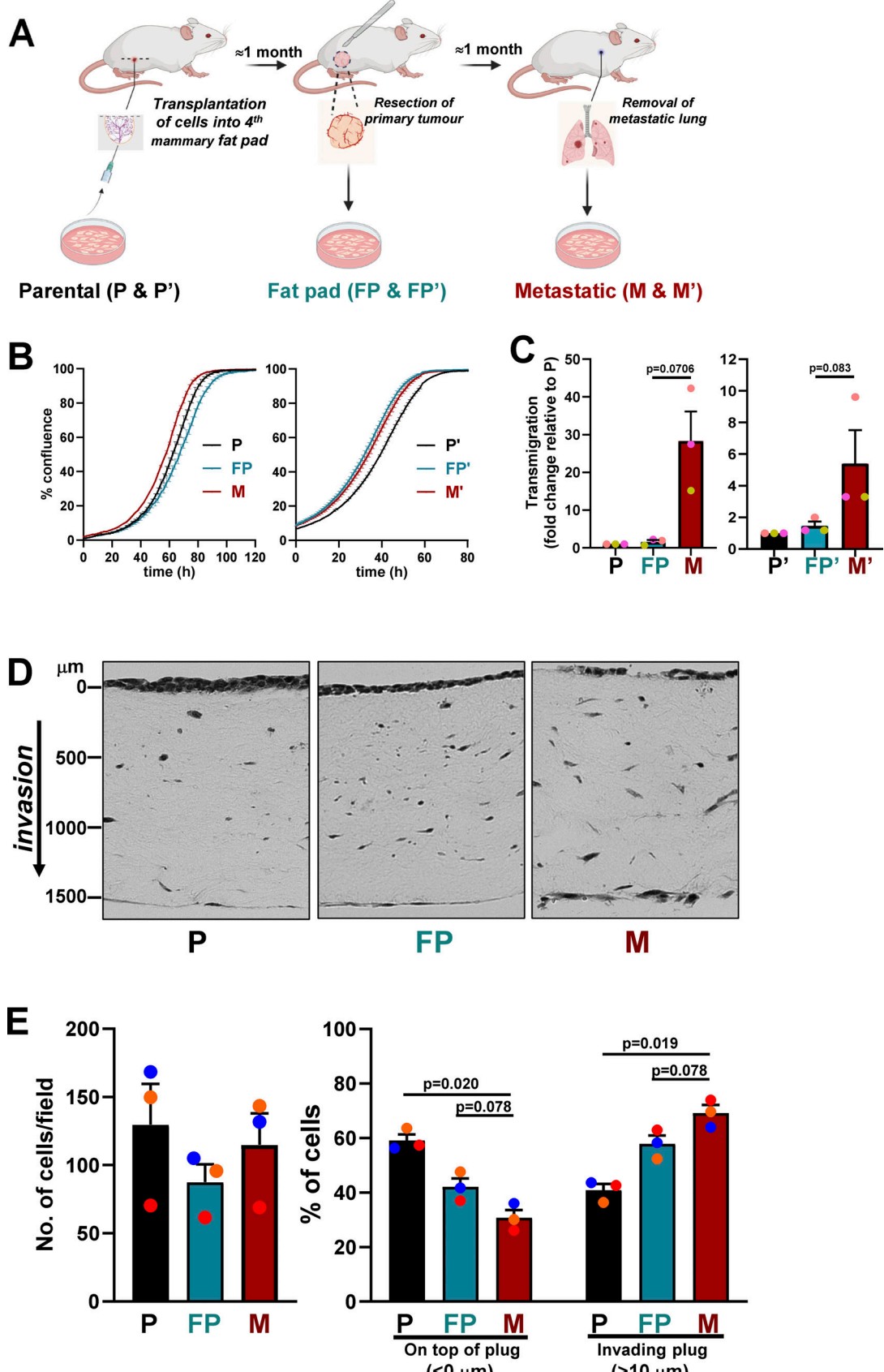

Figure 1. **Cells from lung micrometastases display increased invasiveness. (A)** Parental (P and P') cell lines were established from mammary tumors spontaneously arising in MMTV-*PyMT* female mice. These cells were then transplanted into the fourth mammary FP of syngeneic mice (FVB/N), and tumors

grew to a defined size (8–10 mm). Tumors resected from the mammary FP were used to establish "fat pad" (FP and FP') cell lines. Following tumor resection, mice were maintained for sufficient time (1 mo) to allow seeding of micrometastases in the lung. Lungs were then removed, and *PyMT*-positive "micro-metastatic" (M and M') cell lines were established from lung homogenates. **(B)** P, FP, or M and P', FP', or M' cells (as described in A) were plated onto 6-well dishes, and their growth was determined using the IncuCyte ZOOM live-cell imaging system. Values are mean ± SEM, n = 3 technical replicates/cell line. **(C)** P, FP, or M and P', FP', or M' cells were seeded into the upper chambers of Transwells (8-μm pore size), and transmigration over a 2-h period toward a gradient of serum and fibronectin (applied to the lower chamber) was determined. Quantification of the number of cells adherent to the upper surface of the lower chamber are displayed. Values are mean ± SEM, n = 3 independent experiments (colored dots); data were analyzed using a paired t test. **(D and E)** P, FP, or M cells were plated onto plugs of rat tail collagen that had previously been conditioned by TIFs for 4 days. Cancer cells were allowed to invade preconditioned plugs for 7 days, followed by fixation and visualization of cells using H&E. Representative images (D) and quantification of the total number (E, left graph) and % (E, right graph) of cells remaining on top of or invading collagen organotypic plugs to a depth of >10 μm are displayed. Values are mean ± SEM, n = 3 independent experiments (colored dots), up to 52 fields of view were analyzed per cell line in each experiment; data were analyzed using a paired t test.

---

macrometastases of sufficient size (2.2 ± 0.45 mm in length [longest axis]) were occasionally visible to the naked eye, most cancer cells in the lung were present as micrometastases (each with an average diameter of 149 ± 71 μm). Visible macrometastases were excised and placed into culture, whereas lungs, containing numerous micrometastases, were minced and cultured separately. We then cultured the *PyMT*-expressing cancer cells that grew out from these lung homogenates—thereafter referred to as micrometastatic (M) cells. We generated two series of these cells (P, FP, M and P', FP' and M', respectively) using independent MMTV-*PyMT* mice. All cells in these series expressed the *PyMT* antigen and E-cadherin, confirming that they were tumor cell—derived and of epithelial origin. Moreover, cells from primary tumors (P/P' and FP/FP') and micro-metastases (M/M') all grew at similar rates (Fig. 1 B). However, micrometastatic cells (M/M') were significantly more migratory than their primary tumor counterparts (P/P' or FP/FP') as determined by transmigration toward a gradient of fibronectin and serum (Fig. 1 C). Invasive behavior was assessed by measuring penetration of cancer cells into "organotypic" plugs of collagen that had been preconditioned with telomerase-immortalized dermal fibroblasts (TIFs) (Timpson et al., 2011). This indicated that the reintroduction of MMTV-*PyMT* (P) cells into the mammary FP to generate the FP cells led, in itself, to a moderate increase in invasiveness. However, micrometastatic (M) cells were significantly more invasive in this organotypic microenvironment than cells from primary tumors (P or FP) (Fig. 1, D and E). Finally, to assess the stability of the differences between the micrometastatic and primary tumor cells, we reintroduced M cells into the fourth mammary FP of FVB mice and monitored their growth. While all (5/5) of the primary tumor cells (P) grew efficiently to form lesions of 300–1,000 mm², micrometastatic cells were unable to establish tumors in the mammary FP within 10 wk. This indicated that, by moving to the lung to establish micrometastases, "M" cells have undergone a relatively stable non-transient selection or adaptation that renders them more migratory and invasive but no longer competent to grow in the microenvironment of the mammary FP.

## Micrometastatic (but not macrometastatic) cells increase proline synthesis at the expense of glutathione production

As we were interested in how metastasizing mammary cancer cells might rewire their metabolism during the early stages of seeding the lung, we compared how the cells we have established from MMTV-*PyMT* primary tumors (P and FP) and their micrometastases (M) altered the profile of extracellular metabolites over a 24-h period. Although most metabolites were either consumed (e.g., glucose and glutamine) or secreted (e.g., lactate and alanine) at similar rates in all cells, one metabolite (proline) had markedly altered consumption/secretion dynamics between primary tumor-derived (P and FP) and micrometastatic (M) cells (Fig. 2 A). Strikingly, proline was consumed by primary tumor-derived cells, whereas micrometastatic cells secreted this amino acid (Fig. 2 A). To determine whether increased proline secretion by metastatic cells might manifest in vivo, we profiled the circulating metabolome of MMTV-*PyMT* mice and healthy age-matched control animals. We determined the lung metastatic burden of tumor-bearing mice and subdivided them into those that did (Mets) and did not (no Mets) display lung metastases, as determined by retrospective histological examination. This indicated that MMTV-*PyMT* mice bearing lung metastases displayed significantly higher levels of circulating proline than MMTV-*PyMT* mice that had extensive primary tumor growth in the mammary gland but no metastases (Fig. 2 B). By contrast, other amino acids (such as asparagine and serine), whose levels did not differ between conditioned media from P, FP, or M cells (Fig. 2 A), were similar in mice that did and did not bear metastases (Fig. 2 B). This result from a mouse model of metastatic mammary cancer encouraged us to compare levels of proline in the circulation of healthy volunteers and patients with metastatic breast cancer. Circulating proline (but not asparagine or serine) was significantly elevated in the plasma of patients with metastatic breast cancer, indicating that this amino acid may represent a potentially useful biomarker for metastasis in breast cancer (Fig. 2 C). Consistently, elevated levels of plasma proline have been previously identified as being positively associated with breast cancer in several studies (Brantley et al., 2022; Li et al., 2020; Miyagi et al., 2011).

We then proceeded to look for intracellular polar metabolites that were elevated within micrometastatic cells by comparison with their primary tumor counterparts. Only one metabolite, proline, was increased in both series of micrometastatic cells (M and M') by comparison with their primary tumor counterparts (P & P'; FP & FP') (Fig. 3, A and B; and Fig. S1, A and B), consistent with our finding that micrometastatic cells release more proline and that this amino acid is specifically elevated in the circulation of mice with lung metastases. We detected only one metabolite—glutathione (in both its reduced [GSH] and oxidized [GSSG] forms)—that was significantly decreased in micrometastatic cells (M and M') by

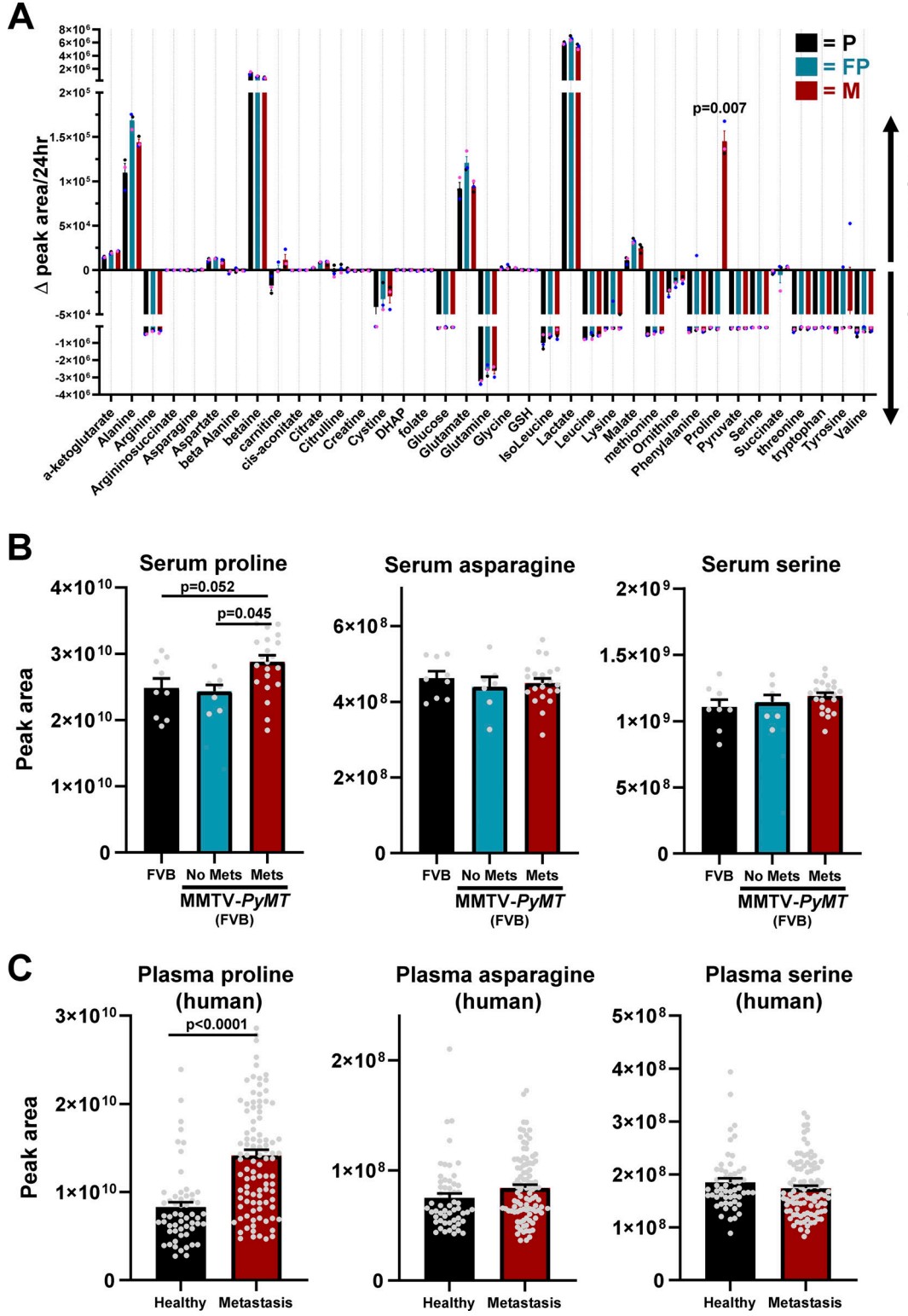

Figure 2. **Micrometastatic cells secrete proline. (A)** Parental (P), FP, or micrometastatic (M) cells were plated onto 6-well plates and incubated at 37°C for 24 h. Conditioned medium was collected, and levels of the indicated metabolites were determined using LC-MS–based metabolomics. Data are expressed as the difference between metabolite levels detected in cell-conditioned medium and those in medium incubated at 37°C in the absence of cells (Δ peak area). Thus, positive values indicate production and release of a metabolite by cells, whereas negative values indicate consumption of that metabolite during the 24-h period. Values are mean ± SEM, $n$ = 3 independent experiments; data were analyzed by two-way ANOVA with Tukey's multiple comparison. **(B)** MMTV-*PyMT* and non-tumor–bearing FVB control mice were culled at 11–14 wk of age, blood was collected by puncture of the posterior vena cava, sera were prepared by centrifugation, and levels of the indicated metabolites were determined using LC-MS. Lungs of MMTV-*PyMT* mice (MMTV-*PyMT* (FVB)) and non-tumor–bearing FVB control mice (FVB) ($n$ = 9 mice) were assessed by histology and categorized according to the presence (Mets; $n$ = 22 mice) or absence (no Mets; $n$ = 7 mice)

of lung metastases. Values are mean ± SEM; data were analyzed by ordinary one-way ANOVA with Sidak's multiple comparison. **(C)** LC-MS metabolomics was used to determine levels of the indicated metabolites in plasma collected from metastatic breast cancer patients ($n$ = 99) and healthy volunteers ($n$ = 56). Values are mean ± SEM; data were analyzed using an unpaired Student's $t$ test.

comparison with cells from primary tumors (P and P'; FP and FP') (Fig. 3, A and B; and Fig. S1, A and B). Moreover, the ratio of GSH/GSSG did not appear to differ between cells from primary tumors and micrometastases (Fig. 3 B and Fig. S1 B). To further confirm these alterations to GSH levels, we deployed the thiol alkylating reagent N-ethylmaleimide (NEM)—a cell-permeable agent that reacts with GSH (but not GSSG)—prior to metabolite extraction (Giustarini et al., 2013; Sun et al., 2020). This indicated that GS-NEM (the glutathione adduct of NEM) levels were decreased in NEM-treated micrometastatic cells, indicating that these cells display genuinely decreased levels of GSH (Fig. 3 C). De novo glutathione biosynthesis involves two ATP-dependent enzymatic steps, where ligation of cysteine to glutamate is catalyzed by glutamate-cysteine ligase (GCL), the rate-limiting enzyme for glutathione biosynthesis to form γ-glutamylcysteine (Bansal and Simon, 2018). Subsequently, glycine is added to this dipeptide by glutathione synthetase to form the tripeptide, GSH. Although cysteine levels remained unchanged, we found that levels of γ-glutamylcysteine were decreased in micrometastatic cells (Fig. 3 C). To determine whether expression of glutathione biosynthetic enzymes might explain these reductions in GSH and GSSG levels, we used qPCR to quantify the expression of *Gclc*, encoding the catalytic subunit of GCL. *Gclc* expression was suppressed in micrometastatic cells (M) by comparison with cells from primary tumors (P and FP), indicating that suppression of GCL activity may be one reason underlying decreased glutathione levels in micrometastatic cells (Fig. 3 D, left graph). By controlling intracellular levels of cystine and cysteine, the glutamate-cystine antiporter, xCT, can also influence glutathione levels (Bansal and Simon, 2018). xCT is upregulated in various human cancers, and previously published work has highlighted the ability of *PyMT*-derived primary tumor cells to release glutamate via increased xCT expression (Dornier et al., 2017). qPCR indicated that micrometastatic cells express almost 50% less xCT (*Slc7a11*) than their primary tumor-derived counterparts (Fig. 3 D, right graph). To further assess the (patho)physiological relevance of our finding, we used RNA in situ hybridization (ISH) to compare xCT expression in primary tumors and the micrometastases present in the lungs of matched MMTV-*PyMT* mice. This indicated that primary tumors express substantial amounts of xCT, whereas this was significantly decreased in lung micrometastases from the same animals (Fig. 3 E). Importantly, the surrounding lung parenchyma expressed more xCT than the micrometastases, indicating that ISH was effective in this tissue. These data indicate that xCT is downregulated in lung micrometastases by comparison with primary tumor cells and that this, in combination with the reduction in GCL levels, may lead to decreased glutathione in micrometastatic cells.

In addition to culturing cells from micrometastases (M and M'), we have dissected larger/frank metastases from the lung of MMTV-*PyMT* tumor-bearing mice and established cultures from

these (maM' cells). Metabolomic analysis indicated that cells cultured from frank metastases display similar GSH, GSSG, and proline levels to their primary tumor-derived counterparts (Fig. S1, C and D). Thus, adoption of a metabolic profile in which proline and glutathione levels are, respectively, elevated and suppressed is a feature that is unique to micrometastatic cells and is not apparent in cells derived from a larger/frank metastasis.

Because glutamate is a precursor of both glutathione and proline, we hypothesized that levels of these metabolites might move in opposing directions when cancer cells form micrometastases, perhaps due to competition for carbons from this shared precursor. We tested this by performing metabolic tracing using $^{13}C_5$-labelled glutamine. In cells from primary tumors (P), glutamine-derived carbons were found predominantly in glutathione and Krebs cycle intermediates, such as αKG (Fig. 4 A). Conversely, in micrometastatic cells (M), glutamine-derived carbons were present in increased levels in both the cellular and secreted pools of proline (Fig. 4 A). To directly test whether the glutathione and proline synthesis pathways compete for these carbons, we treated micrometastatic cells with an inhibitor of pyrroline-5-carboxylate reductase (PYCR), a key enzyme in the proline synthesis pathway (Kay et al., 2022; Milne et al., 2019). As expected, the PYCR inhibitor opposed synthesis (and secretion) of proline from glutamine (Fig. 4 B). In addition to inhibiting synthesis of proline from glutamine, this inhibitor led to significant dose-dependent increases in the flux of glutamine-derived carbons toward αKG and glutathione synthesis in micrometastatic cells (Fig. 4 B). To test whether this competition between glutathione and proline synthesis might shunt carbons bidirectionally, we metabolically traced the destinations of glutamine-derived carbons in cystine-depleted conditions. This indicated that reduction (from 20–1 μM) or removal of extracellular cystine (which, as expected, depleted intracellular glutathione) led to an approximately fourfold increase in the synthesis of proline from glutamine (Fig. 4 C).

Taken together, these data indicate that mammary cancer cells that have left the primary tumor and are in the early stages of seeding metastases in the lung increase the glutamine-dependent production of proline, which is then exported from the cells, and that this occurs at the expense of glutathione synthesis (Fig. 4 D). Moreover, this situation is likely to be transient because our data indicate that the balance of proline and glutathione synthesis returns to levels similar to those observed in primary tumors as micrometastases evolve into frank metastatic outgrowth.

## Decreased glutathione synthesis is associated with increased EV release

We and others have established that EV release from primary tumors contributes to priming of the lung metastatic niche (Novo et al., 2018; Peinado et al., 2017). Moreover, we have

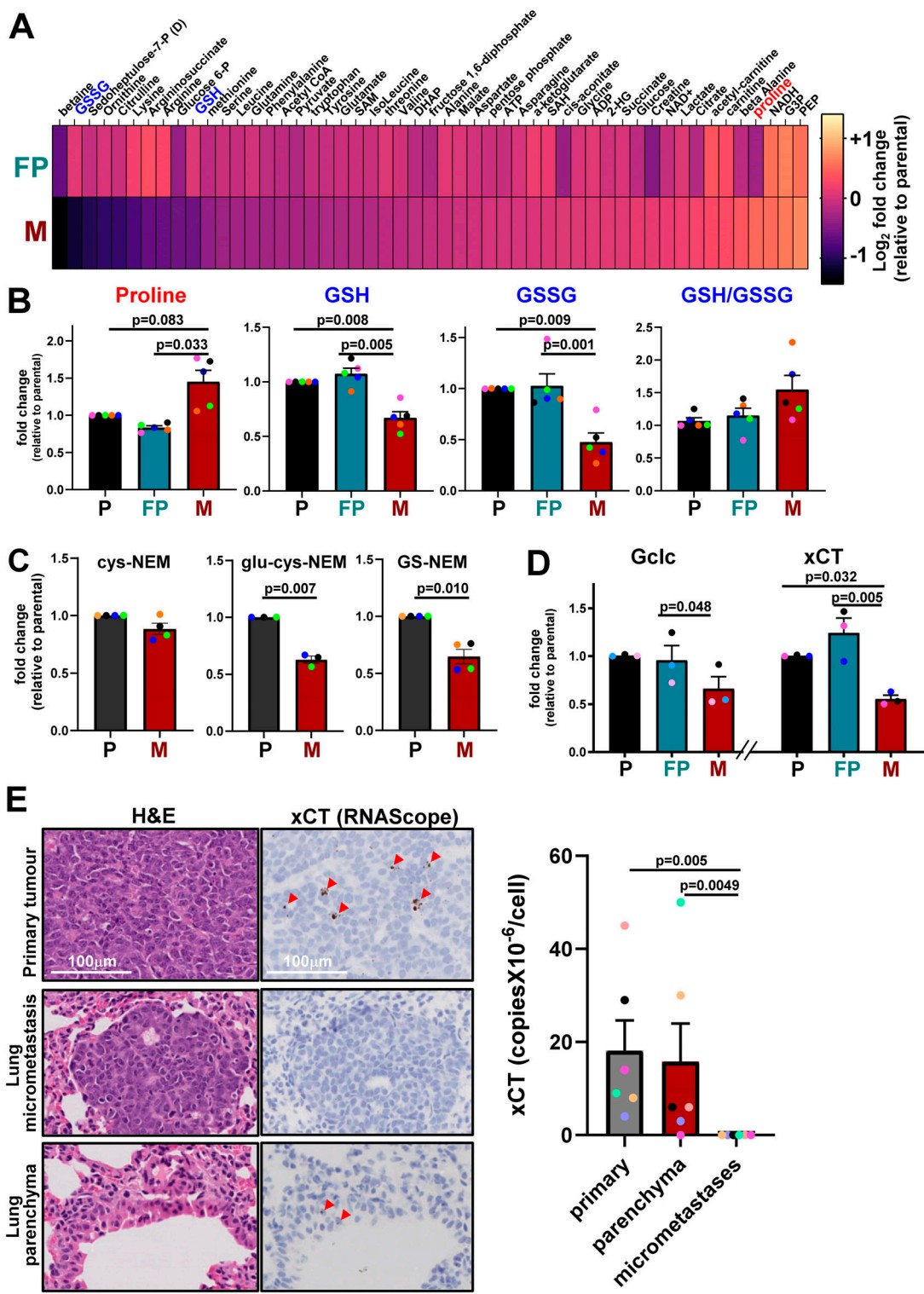

Figure 3. **Proline and glutathione metabolism is altered in micrometastatic cells. (A and B)** Parental (P), fat pad (FP), or micrometastatic (M) cells were cultured as for Fig. 2 A, and levels of intracellular metabolites were determined using LC-MS–based metabolomics. The abundance of intracellular metabolites in FP and micrometastatic (M) cells was normalized to cell number and expressed relative to levels of the same metabolites in parental (P) cells. For the heatmap in A, values are log$_2$-fold changes, and for B, values are mean fold-change; data were analyzed by repeated measures one-way ANOVA with Tukey's multiple comparison test, $n = 5$ experiments (each colored dot represents an individual experiment). (GSH, reduced glutathione; GSSG, oxidized glutathione; SAM, S-adenosyl methionine; DHAP, dihydroxyacetone phosphate; G3P, glycerol-3-phosphate; PEP, phosphoenolpyruvate; SAH, S-adenosylhomocysteine). **(C)** Parental (P) and micrometastatic (M) cells were cultured as for Fig. 2 A, and lysates derivatized with NEM. NEM adducts of cysteine (cys-NEM), γ-glutamylcysteine (glu-cys-NEM), and reduced glutathione (GS-NEM) were then detected using LC-MS and plotted as fold change relative to parental cells, $n = 3–4$ independent experiments (colored dots); values represent mean ± SEM, paired $t$ test. **(D)** Cells were cultured as for Fig. 2 A, and levels of the mRNAs

encoding GCL catalytic subunit (*Gclc*; left graph) and *Slc7a11* (xCT; right graph) were determined using qPCR. Values were normalized to *ARPP P0* mRNA levels and expressed as fold change relative to parental (P) cells. Values are mean ± SEM, *n* = 3 independent experiments (colored dots); data were analyzed using one-way ANOVA with Tukey's multiple comparison test. **(E)** Mammary tissue and lungs were resected from MMTV-*PyMT* mice at primary tumor endpoint (tumor size 15 mm). Tissues were formalin-fixed, paraffin-embedded, and sectioned for histological examination. Expression of the mRNA-encoding xCT/*Slc7a11* was visualized using ISH (RNAScope), and separate serial sections were counterstained with H&E. Sections representing primary mammary tumors, their matched lung micrometastases, and surrounding lung parenchyma are displayed. Red arrowheads highlight the brown dots that indicate hybridization with the xCT probe, bar 100 μm. Quantification of the average optical density of the xCT probe in stained tissue sections was achieved using HALO software; values are mean ± SEM, Kruskal–Wallis with Dunn's multiple comparisons test, *n* = 6 mice (each colored dot represents an individual mouse).

shown that EVs from tumors with high metastatic potential—such as those that have acquired particular gain-of-function mutations in p53—can generate pro-invasive microenvironments that are associated with tumor dissemination and metastatic seeding (Novo et al., 2018). We were, therefore, interested in measuring EV release from micrometastatic mammary cancer cells and how this might be related to their altered metabolism. We used differential centrifugation to purify EVs from cell-exposed culture medium and analyzed these using nanoparticle tracking to determine their number and size distribution and also western blotting for established EV markers such as protein associated with the endosomal sorting complex required for transport-ESCRT-I (TSG101) and the tetraspanins, CD63 and CD81. This indicated that micrometastatic cells (M and M') released significantly more EVs with a diameter in the range of ~100–200 nm—a size corresponding to that accepted to represent "small" EVs—than their primary tumor counterparts (P and P'; FP and FP') (Fig. 5 A). Consistently, western blotting of the EV pellets indicated that these EVs were positive for CD63, TSG101, and CD81 (Fig. 5 B). We then used buthionine sulfoximine (BSO), an irreversible inhibitor of GCL, to interrogate a causal relationship between decreased glutathione levels and increased EV release. BSO has been extensively used to evoke oxidative stress by depletion of GSH reserves (Lushchak, 2012; Wang et al., 2021). By careful titration, we identified concentrations of BSO (0.625–2.5 μM) that decreased total glutathione (GSSG and GSH) levels in cells from primary tumors to those observed in micrometastatic cells (~40–50% reduction) (Fig. S2 A). Moreover, we were able to decrease glutathione levels in a manner that did not compromise cell growth/viability (Fig. S2 B), was reproducibly stable for 48 h to allow for collection of EVs (Fig. S2 C) and did not detectably disturb the levels of other cellular metabolites (Fig. 5 C). Restricting glutathione synthesis in this way led to a significantly increased release of CD63-positive EVs from cells derived from primary tumors (P) (Fig. 5, D and E; and Fig. S2 D). This indicates that the increased capacity of micrometastatic cells to release EVs is likely to be, at least in part, due to the decreased synthesis of glutathione observed in these cells.

**Micrometastatic cells release EVs via a neutral sphingomyelinase-2–dependent and Rab27-independent mechanism**

Altered lipid metabolism is now recognized to be an acquired feature of cancer cells that enables them to meet anabolic and catabolic needs in the face of rapid cell growth (Cairns et al., 2011). Moreover, accumulation of lipids, increased uptake of fatty acids, and upregulation of genes encoding fatty acid biosynthesis or fatty acid transporters have been shown to enhance

migratory and invasive traits of cancer cells and to be associated with metastatic progression in many cancers (Antalis et al., 2011; Nath and Chan, 2016; Pascual et al., 2017). To gain a picture of lipid classes that might differ between cells from micrometastases (M) and primary tumors (P and FP), we performed an untargeted lipidomic analysis. This clearly identified two main lipid classes—cholesterol esters and sphingomyelin/ceramides (SM/Cer)—as being increased in micrometastatic (M) cells with respect to those from primary tumors (P and FP) (Fig. 5, F and G). As sphingomyelin metabolism and the production of ceramides have been previously shown to promote the budding of intraluminal vesicles within multivesicular endosomes and thereby influence EV biogenesis and membrane cargo sorting (Trajkovic et al., 2008), we decided to focus on studying the alterations in this class of lipids. Immunofluorescence (using a pan-ceramide antibody) confirmed that ceramide levels were generally elevated in micrometastatic cells and that these lipids appeared to be localized to the plasma membrane and to puncta distributed throughout the cytoplasmic space (Fig. 6 A). We then performed a targeted lipidomic analysis—focusing on ceramides and sphingomyelins—which clearly identified four distinct ceramide species whose levels were elevated in micrometastatic cells (M) with respect to their primary tumor counterparts (P and FP) (Fig. 6 B). As hydrolysis of sphingomyelins to ceramides contributes to EV production (Trajkovic et al., 2008), and these lipids are present in EVs, we also profiled the ceramide content of EVs. This indicated that two of the species that were elevated in micrometastatic cells (Cer 35:2:2 and Cer 35:3:2) were significantly enriched in EVs released by these cells (Fig. 6 C). We then hypothesized that increased EV-mediated release of these ceramide species by micrometastatic cells might indicate that these lipids may be used as metastatic biomarkers. To test this, we compared the levels of ceramide species in blood plasma from healthy subjects and from patients with metastatic breast cancer. This indicated that three of the ceramide species (Cer 35:2:2, Cer 35:3:2, and Cer 40:4:2), which were enriched in micrometastatic cells from the MMTV-*PyMT* mouse model (two of which were also elevated in EV pellets from these cells), were present at significantly increased levels in the plasma of metastatic breast cancer patients (Fig. 6 D). This indicates the possibility that the release of EVs rich in certain ceramides from metastatic cells may reflect the presence of metastases in humans with breast cancer.

Ceramides are formed from sphingomyelins by the hydrolytic removal of a phosphocholine group by neutral sphingomyelinases (Trajkovic et al., 2008). Furthermore, because ceramides promote inward budding of endosomal membranes,

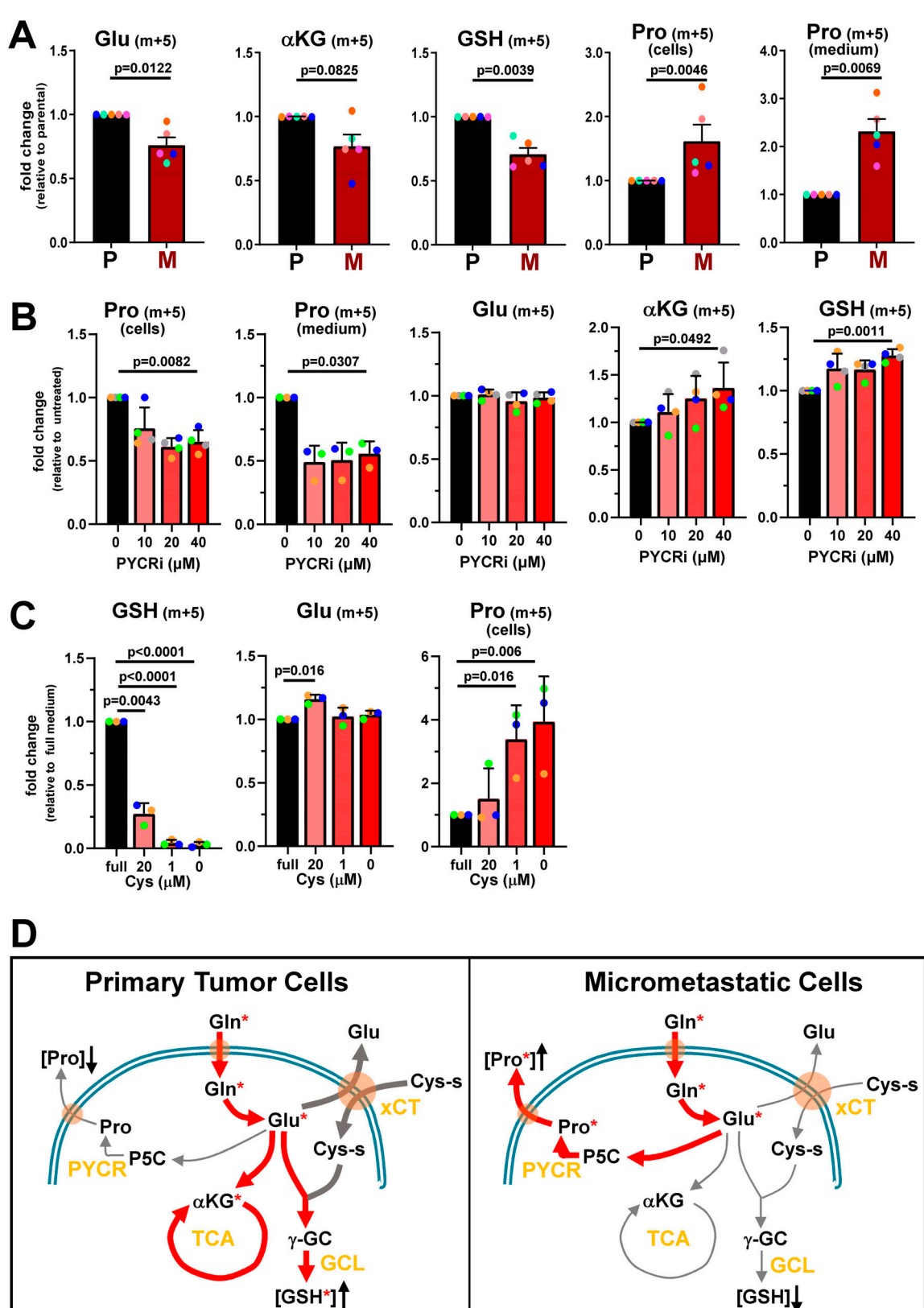

Figure 4. **Proline and glutathione synthesis pathways compete for glutamine-derived carbons in micrometastatic cells. (A and B)** Parental (P) or micrometastatic (M) cells were cultured for 24 h in the presence of $^{13}C_5$-labelled glutamine (A). Additionally, micrometastatic (M) cells were cultured with $^{13}C_5$-labelled glutamine in the presence of the indicated concentrations of PYCR inhibitor (PYCRi) for 24 h (B). The presence of $^{13}C_5$-labelled glutamine-derived carbons in the indicated metabolites was determined using LC-MS. Values are mean ± SEM, $n$ = 5 independent experiments (A), $n$ = 4 (for cellular metabolites) or $n$ = 3 (for secreted metabolites) independent experiments (B); data were analyzed by paired $t$ test. Each colored dot represents an independent experiment.

**(C)** Parental cells were cultured for 6 h in the presence of $^{13}C_5$-labelled glutamine in DMEM (full) or cystine-free DMEM supplemented with the indicated concentrations of cystine. Metabolomics were performed as for A. Values are mean ± SEM, $n$ = 3 independent experiments (colored dots); data were analyzed by two-way ANOVA with Dunnett's multiple comparison test. **(D)** Schematic representation of the utilization of $^{13}C_5$-glutamine–derived carbons (denoted with red asterisks) in *PyMT*-derived primary tumor (P) and micrometastatic (M) cells. Glutamine-derived carbons are primarily used for glutathione production in primary tumor cells, while micrometastatic cells reroute these carbons toward proline synthesis at the expense of glutathione production (Cys-s, cystine).

---

neutral sphingomyelinases contribute to multivesicular body formation and, in turn, EV release. We used CRISPR to disrupt the genes for the two major forms of neutral sphingo-myelinases (Fig. S3 A) and found that reduction of neutral sphingomyelinase-2 (nSM2), but not neutral sphingomyelinase-1 (nSM1) expression, significantly reduced levels of two of the ceramides that we had found most upregulated in micrometa-static cells (Fig. 6 E). We then proceeded to determine the neutral sphingomyelinase dependence of EV release by cells from primary tumors and micrometastases. This indicated that EV release from micrometastatic (M) cells was significantly reduced by CRISPR of nSM2 (but not nSM1), whereas EV release by cells from primary tumors (P and FP) was less dependent on nSM1 and nSM2 (Fig. 7, A and B; and Fig. S3, B and C). Moreover, treatment of primary tumor cells (P) with 1.88 μM BSO (to judiciously decrease glutathione levels) drove EV release that was nSM2 dependent (Fig. 7, C and D).

The Rab27 GTPases are also key regulators of EV release (Ostrowski et al., 2010). We, therefore, used CRISPR to reduce protein expression of Rab27a or Rab27b in both micrometa-static and primary tumor cells (Fig. S3 D) and measured EV release into their conditioned media. As anticipated, suppression of Rab27a or Rab27b expression strongly opposed the release of EVs from cells from primary tumors (P) (Fig. 7, E and F). However, EV release from micrometastatic cells (M) was not opposed by disruption of Rab27s and was even increased following CRISPR of Rab27a (Fig. 7, E and F). Taken together, these data indicate that, when mammary cancer cells begin to colonize the lung, they may upregulate ceramide levels to enable a switch from Rab27-dependent/nSM2-independent EV production to a mechanism of EV release that is nSM2-dependent but independent from Rab27s.

### The switch to nSM2-dependent EV release enables micrometastatic cells to generate pro-invasive microenvironments

Upregulated ceramide levels and the consequent switch to nSM2-dependent EV release prompted us to investigate the role of this enzyme in the invasiveness of micrometastatic cells. CRISPR of nSM2 did not influence transmigration toward a gradient of serum and fibronectin (Fig. 8 A), indicating that cell-autonomous haptotactic/chemotactic migratory behavior did not depend on nSM2-dependent EV release from micrometastatic cells. However, despite their intrinsically migratory capabilities, micrometastatic cells in which nSM2 had been disrupted by CRISPR displayed a reduced ability to invade through organotypic collagen plugs (Fig. 8 B). As the microenvironment of organotypic plugs is strongly influenced by fibroblasts, this indicates the possibility that nSM2-dependent EV release enables communication between micrometastatic

cells and fibroblasts to permit cancer cell invasiveness. To test this, we pre-treated fibroblasts with EVs from control or nSM2 CRISPR cells prior to introducing them into collagen plugs (Novo et al., 2018). EV-treated fibroblasts were then allowed to precondition collagen plugs for 4–5 days. Control or nSM2 CRISPR micrometastatic cells were then plated onto, and allowed to invade into, preconditioned plugs for a further 7 days (Fig. 8 C). This indicated that when collagen plugs were preconditioned with EVs from control (but not nSM2-CRISPR) micrometastatic cells, this restored the ability of nSM2-CRISPR cells to invade into the organotypic microenvironment (Fig. 8 D).

This result led us to propose that EVs released in an nSM2-dependent manner from micrometastatic cells may modify the microenvironment of the organotypic plugs by influencing how the fibroblasts deposit and/or remodel the ECM (Novo et al., 2018). To test this, we pre-treated the TIFs with EVs from control or nSM2-CRISPR micrometastatic cells (as for Fig. 8 C), but, instead of introducing them into collagen plugs, we allowed them to deposit ECM in 2D culture for 7 days. We then de-cellularized the ECM and measured the migration speed of MDA-MB-231 mammary cancer cells plated onto this. Pre-treatment of TIFs with EVs from control, but not nSM2 CRISPR, micrometastatic cells increased the migration speed of MDA-MB-231 cells subsequently plated onto ECM deposited by these fibroblasts (Fig. 8 E). Taken together, these data indicate that when disseminated mammary cancer cells move to the lung and form micrometastases, they alter their metabolism in a way that favors nSM2-dependent/Rab27-independent production of EVs. EVs released via this nSM2-dependent route then influence ECM deposition by fibroblasts to foster a microenvironment that may contribute to subsequent metastatic colonization of the lung (Fig. 8, F and G).

## Discussion

Metabolic reprogramming enables cancer cells to overcome challenges encountered on the path to metastasis and may enhance their potential to seed distant organs (Roshanzamir et al., 2022). We show here that cells from mammary cancer lung micrometastases have rewired their metabolism in a way that generates an invasive microenvironment. The metabolic adaptations of lung micrometastases are manifest in the rerouting of glutamine-derived carbons toward proline production (and secretion) at the expense of glutathione synthesis—likely owing to reduced xCT and GCL levels and increased nSM2-dependent synthesis of ceramides. The combination of decreased glutathione synthesis and increased ceramide levels promotes EV release via a Rab27-independent/nSM2-dependent route, which, in turn, influences ECM deposition/remodelling by fibroblasts to foster an invasive microenvironment.

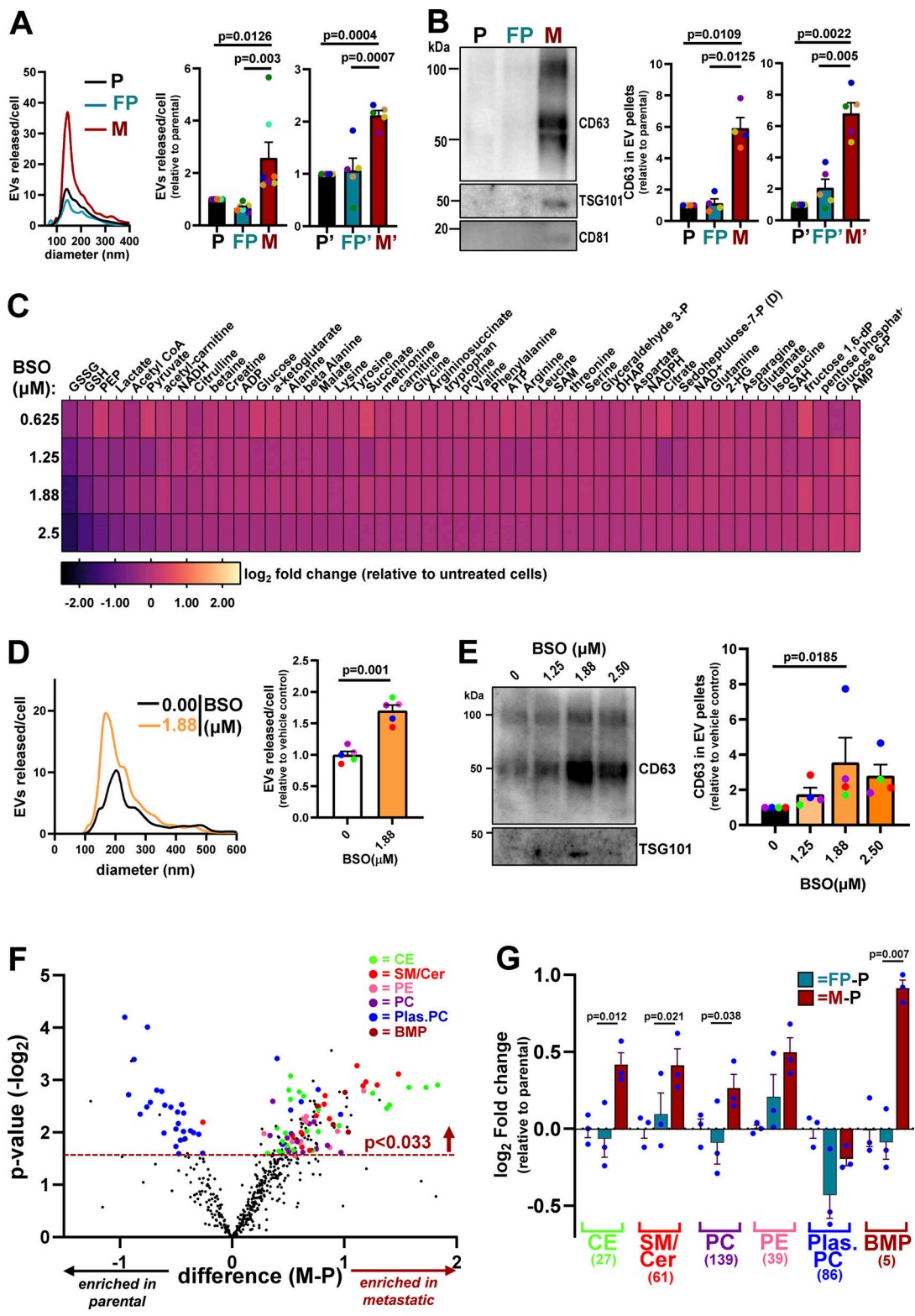

Figure 5. **Decreased glutathione synthesis in micrometastatic cells is associated with increased EV release. (A and B)** Conditioned media were collected from parental (P and P'), FP (FP and FP'), and micrometastatic (M and M') cells, and EVs were purified from these using differential centrifugation. The size distribution and number of EVs were analyzed by nanoparticle tracking (A), and the CD63, TSG101, and CD81 content of EV pellets was determined by western blotting (B). Values are mean ± SEM, $n = 7$ (P, FP, and M); $n = 5$ (P', FP', and M') (A) or $n = 4$ (P, FP, and M); $n = 5$ (P', FP', and M') (B) independent experiments; data were analyzed by one-way ANOVA with Tukey's multiple comparisons test. **(C–E)** Parental (P) cells were incubated with the indicated concentrations of BSO for 24 h, and the cells lysed for determination of intracellular metabolites by LC-MS (C). Following another 24 h, cell-conditioned medium was collected for isolation of EVs by differential centrifugation (D and E). The heatmap (C) displays levels of the indicated metabolites expressed as $\log_2$-fold change (normalized to cell number) relative to untreated (P) cells, $n = 1$ (three technical replicates/condition). The size distribution and number of EVs

released by cells incubated in the presence or absence of BSO (1.88 µM) were analyzed by nanoparticle tracking (D), and the CD63 and TSG101 content of EV pellets from untreated or BSO-treated (1.25 µM, 1.88 µM, and 2.5 µM) P cells was determined by western blotting. CD63 levels in EV pellets were quantified as for Fig. 5 A (E). Values represent mean ± SEM, n = 4 independent experiments (colored dots); data were analyzed by Friedman ANOVA with Dunn's multiple comparison. **(F and G)** Parental (P), FP, or micrometastatic (M) cells were cultured as for Fig. 2 A, and levels of intracellular lipidic metabolites were determined using LC-MS–based metabolomics. Data are expressed as a volcano plot (F) showing the mean differences (M minus P [M-P]; x axis) between the peak areas of lipids identified in M and P cells. The dotted line represents the P value (y axis) above which all the indicated lipid classes display significant differences across the conditions, n = 3 independent experiments, data were analyzed by paired t test. The classes of lipids that were detected at significantly different (P < 0.033) levels between M and P cells are denoted with colored dots. Color coding for the lipid classes is as follows: cholesterol esters (CE); sphingomyelin/ceramide (SM/Cer); PE, phosphatidylethanolamine (PE); phosphatidylcholine (PC); plasmanyl/plasmenyl phosphatidylcholine (Plas.PC); bis(monoacylglycerol) phosphate (BMP). $Log_2$-fold differences peak areas of the indicated lipid classes (normalized to cell number) between FP and P (FP minus P; blue bars) and M and P (M minus P; red bars) are displayed in the graph in G, n = 3 independent experiments; data were analyzed using two-way ANOVA with Tukey's multiple comparison test. Source data are available for this figure: SourceData F5.

Evidence is accumulating that alterations to proline metabolism can drive breast cancer metastasis (Elia et al., 2017, 2019; Kay et al., 2022), and we concur that proline metabolism is indeed rewired during early metastasis. Micrometastatic cells from the lungs of mice bearing MMTV-*PyMT* tumors increase proline production and release it into the medium at the expense of glutathione synthesis. Moreover, because pharmacological inhibition of proline synthesis (Milne et al., 2019) can, at least in part, restore glutathione synthesis to micrometastatic cells, this indicates that the need to increase proline production may be a metabolic priority during early metastatic seeding. Our data indicate that most of the proline synthesized by micrometastatic cells is not retained in the cells but exported into the medium, suggesting that a role for these cells may be to increase proline levels in the extracellular microenvironment during early metastasis. As ECM components, particularly collagens, have very high proline content, it is, therefore, possible that proline exported from micrometastatic cells may be used by other cells in the lung microenvironment to synthesize collagen-rich ECM. Indeed, a recent study has described how myofibroblastic-type carcinoma-associated fibroblasts rewire their metabolism to favor synthesis of proline via PYCR, which is then used for collagen production to drive tumor progression and aggression (Kay et al., 2022). Such a requirement for metabolic support to ECM production during early metastatic seeding has been highlighted by a study describing how breast cancer cells can generate metastatic niches in the lung by increasing production of αKG to activate the hydroxylation of prolines in collagens (Elia et al., 2019). In addition to supporting ECM production, evidence is accumulating that regulation of proline synthesis plays a key role in maintaining redox balance in metabolically stressed cells. Proline synthesis via PYCR can lead to NAD(P)H consumption, thus providing a route to regeneration of NAD(P)+ without the cell needing to activate oxidative phosphorylation and the consequent generation of damaging reactive oxygen species (ROS) (Schworer et al., 2020; Westbrook et al., 2022). Indeed, metabolic flux analysis indicates that the micrometastatic cells of the present study do not display significantly elevated oxygen consumption, despite considerable flux of glutamine-derived carbons into the Krebs cycle. This raises the possibility that micrometastatic cells exploit proline synthesis in a manner that allows sufficient Krebs cycle activity to maintain anaplerosis, but without passing high-energy electrons to the electron transport chain thus increasing the

likelihood of generating ROS. Indeed, oxidative stress is established to restrict tumor cell growth and, particularly, to reduce distant metastasis (Le Gal et al., 2015; Piskounova et al., 2015; Roshanzamir et al., 2022). Thus, micrometastatic cells may avoid excessive ROS generation to maximize their chances of survival in the highly oxidative environment of the lung and/or the circulation.

Increased proline synthesis may reduce ROS generation, but the fact that this occurs at the expense of glutathione production still presents micrometastatic cells with a redox challenge. Although there have been long-running controversies over the role of antioxidants in cancer therapy, it is now established that glutathione synthesis and maintenance of a reducing cellular environment are necessary for both primary tumor initiation and growth (Harris et al., 2015). Consistently, several studies now demonstrate that inhibition of glutathione synthesis—typically by targeting xCT—increases cancer cell death and inhibits tumor growth (Koppula et al., 2018, 2021). The reasons for this are multifarious and likely include roles for the glutathione/thioredoxin system in inhibiting ferroptosis (and other pathways driving cell death) and the ability of antioxidants to prolong oncogene-induced senescence during early tumorigenesis by reducing oxidative stress. Furthermore, metastatic outgrowth—a later event in metastasis—also relies heavily on glutathione synthesis, as highlighted by a recent study describing how cell lines from established colon cancer (macro-)metastases are particularly sensitive to inhibition of cystine import via xCT (Tarrago-Celada et al., 2021). Nevertheless, although a level of glutathione synthesis is clearly a requirement for growth of primary tumors and of established metastases, this does not preclude transient adoption of metabolic states characterized by low glutathione levels during early metastatic seeding. Indeed, a recent study has demonstrated that, although glutathione production is required for tumor initiation in mammary carcinogenesis, it becomes dispensable later in disease progression (Harris et al., 2015). These authors showed that inhibition of glutathione synthesis decreases tumor burden only when administered prior to primary tumor onset. Inhibition of glutathione synthesis following tumor onset evokes alternative antioxidant responses, enabling glutathione-depleted cells to thrive despite oxidative insults. Another recent study shows that high expression levels of xCT promote primary tumor growth but suppress metastasis (Yan et al., 2023). The mechanistic interpretation offered by these authors is that

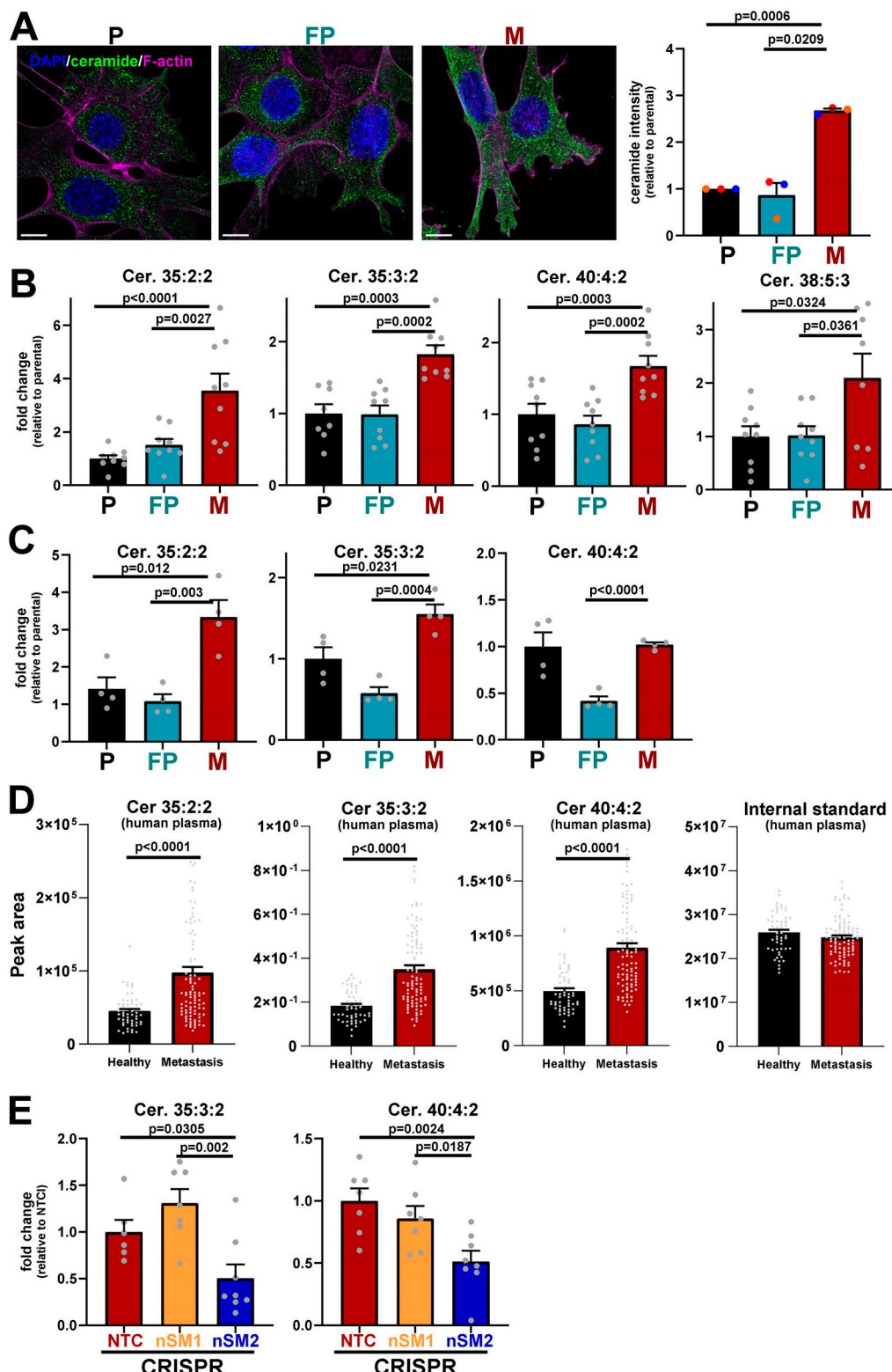

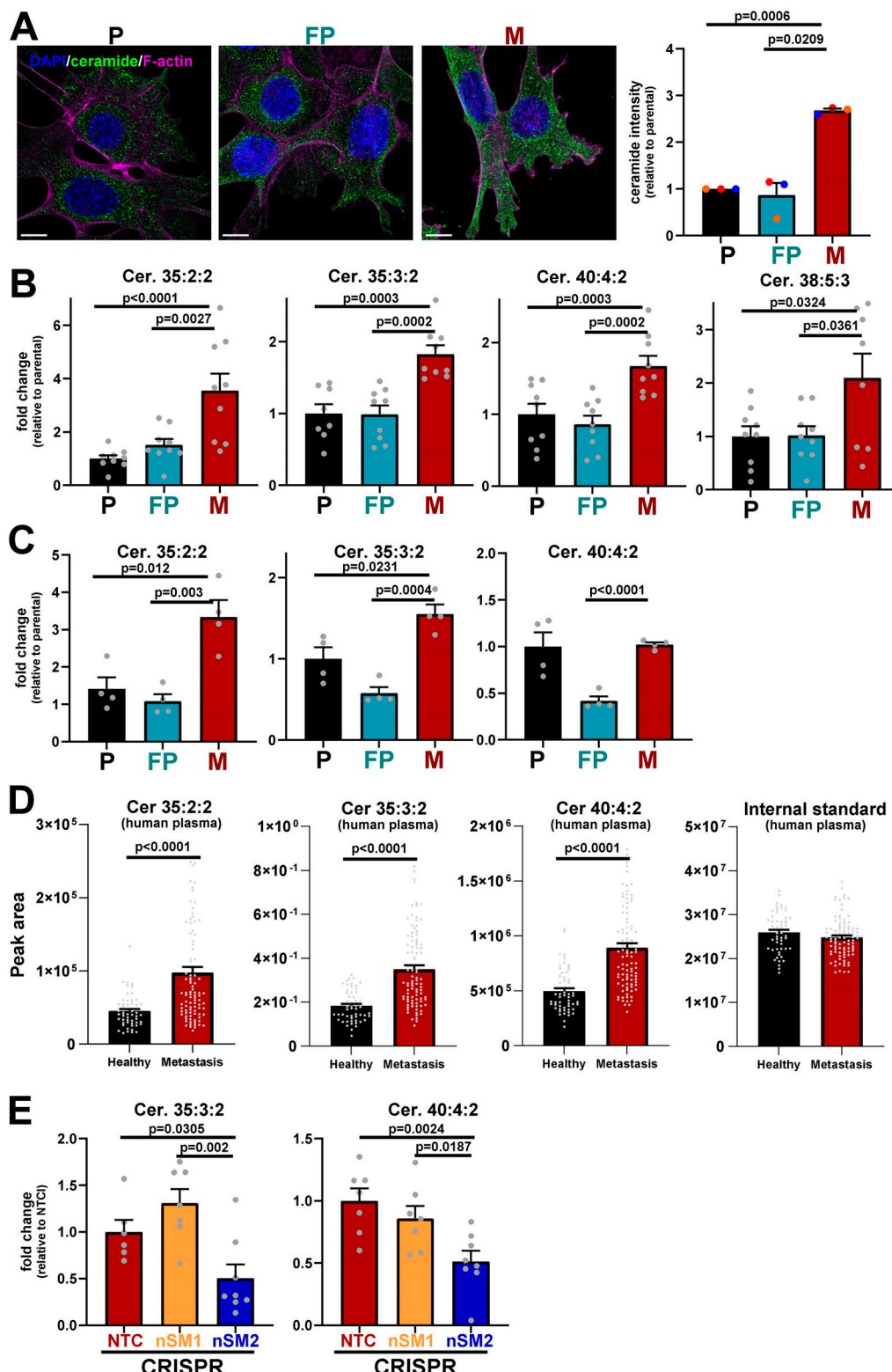

Figure 6. **Micrometastatic cells and their EVs are enriched in ceramide species. (A)** Parental (P), FP, and micrometastatic (M) cells were cultured for 48 h on glass-bottomed dishes and fixed using paraformaldehyde. Ceramides (green), F-actin (phalloidin; magenta), and nuclei (DAPI; blue) were visualized by immunofluorescence; bar is 10 µm. ImageJ was used to quantify the mean intensity of ceramide (sum of z-stacks, 10 stacks/field of view). Values are mean ± SEM, n = 3 independent experiments (colored dots); data were analyzed by one-way ANOVA with Tukey's multiple comparisons. **(B)** The intracellular levels of ceramide species in P, FP, and M cells were determined as for Fig. 5, F and G. Ceramide species found to be present at significantly different levels between M and FP or P cells were normalized to cell number and expressed as fold change relative to (P) cells; values are mean ± SEM; data were analyzed using one-way ANOVA with Tukey's multiple comparison test, n = 3 independent experiments. Greater-than two technical replicates/experiment were performed, and each technical replicate is represented by a dot. **(C)** P, FP, and M cells were cultured for 48 h. Conditioned media were harvested from these cultures, and EVs

purified using differential centrifugation. LC-MS–based lipidomics was used to determine the levels of the indicated ceramide species in the EV pellets; values are mean ± SEM, $n$ = 4 independent experiments (each dot represents one experiment); data were analyzed by one-way ANOVA. Lipid extracts of EVs were normalized to the number of EV-releasing cells prior to lipidomic analysis. **(D)** LC-MS metabolomics was used to determine levels of the indicated ceramide species in plasma collected from metastatic breast cancer patients ($n$ = 96) and matched healthy volunteers ($n$ = 55). An internal standard (Splash II, Avanti) was spiked in the lipid extraction buffer and was analyzed to account for technical variation among samples (right graph). Values are mean ± SEM; data were analyzed by unpaired Student's $t$ test. Each dot represents an individual patient. **(E)** M cells were transduced with lentiviral vectors bearing gRNAs against non-targeting sequences (NTC) or recognizing sequences in nSMase1 (nSM1) or nSMase2 (nSM2). Ceramides were determined using LC-MS as for (B). Values are mean ± SEM, $n$ = 3 independent experiments; data were analyzed by one-way ANOVA with Tukey's multiple comparison test. Greater-than two technical replicates/experiment were performed, and each technical replicate is represented by a dot.

metastasizing cancer cells with high xCT activity are particularly susceptible to oxidative stress because this increases intracellular cystine. Subsequent conversion of intracellular cystine to cysteine—a redox reaction necessary for glutathione synthesis—mops up intracellular NADPH, rendering the metastasizing cells sensitive to disulfide stress and death. Our finding that micrometastatic cells cultured from the lungs of mice bearing MMTV-*PyMT* tumors display reduced xCT and GCL also suggests that moderate expression levels of xCT—and, therefore, glutathione synthesis—may be necessary during the early stages of metastatic seeding because of the need to upregulate proline synthesis. Increased proline synthesis and its release not only help to maintain redox balance but also offer the possibility of contributing to ECM synthesis by other cells in the lung microenvironment to condition the early metastatic niche.

It is possible that diversion of glutamine-derived carbons toward proline and away from glutathione synthesis contributes to the increased invasiveness in a cell-autonomous manner. However, inhibition of PYCR (which, respectively, decreases proline and increases glutathione synthesis) does not reduce transmigration of micrometastatic cells (Fig. S4 A), indicating that more complex mechanisms exert metabolic control over invasiveness of micrometastatic cells and their microenvironment. Indeed, decreased glutathione, in combination with increased ceramide levels, contributes to switching off Rab27-dependent EV release and encourages micrometastatic cells to release EVs via an nSM2-dependent pathway. It is currently unclear how low glutathione levels switch on nSM2-dependent EV production. As serine is a precursor for ceramide and serine can be generated from glycine, one possibility is that decreased glutathione synthesis increases glycine levels and, in turn, the ability to synthesize ceramide from serine. However, glycine and serine levels are not elevated in micrometastatic cells (Fig. S4 B) nor following BSO-mediated blockade of glutathione synthesis (Fig. S4 C). Thus, it remains unclear why nSM2-dependent generation of ceramide increases in micrometastatic cells, and further experimentation will be needed to address this.

nSM2-dependent production of EVs supports deposition of pro-invasive ECM by fibroblasts, and it is interesting to compare this with other situations in which tumor cells can, via EV release, foster invasiveness in other cells. Metabolic stress in the MDA-MB-231 breast cancer cell line promotes PINK1-dependent packaging of mtDNA into EVs that are released via the Rab27 pathway, and these EVs alter invasive behavior of other tumor cells by activating TLR9 signalling (Rabas et al., 2021). Cells from MMTV-*PyMT* primary

tumors (P cells) also release mtDNA-containing EVs ([Rabas et al., 2021] and Fig. S5, A–C). However, and despite previous studies linking nSM2 with mitochondrial function and the release of mtDNA-containing EVs (Torralba et al., 2018), micrometastatic (M) cells release much less mtDNA in association with EVs than do cells from primary tumors (P) (Fig. S5, A–C). Thus, although the present study and previous work from our laboratory (Rabas et al., 2021) both indicate that altered metabolism can communicate invasiveness to other cells via EV release, the EV cargoes mediating this communication must differ between micrometastatic cells and their primary tumor counterparts. Other studies from our laboratory have shown that acquisition of gain-of-function mutations in p53 alters the podocalyxin content of EVs (Novo et al., 2018). This, in turn, alters integrin trafficking in fibroblasts to encourage them to deposit a pro-invasive ECM. However, we have measured α5β1 integrin recycling and found this not to differ between fibroblasts pre-treated with EVs from control or nSM2-CRISPR micrometastatic cells. Moreover, TWOMBLI analysis (Wershof et al., 2021) indicates that EVs from micrometastatic cells do not influence the fibronectin filament length or organization in the ECM deposited by fibroblasts (Fig. S5, D–G), further indicating that its pro-invasive nature is not a consequence of altered α5β1 integrin-dependent ECM deposition. Thus, further work will be necessary to determine how EVs released via an nSM2-dependent pathway influence ECM deposition by fibroblasts, and it will be interesting to evaluate the role of lipidic EV cargoes, such as ceramides and cholesterol esters, in this process.

Taken together, our study has elucidated the metabolic reprogramming that occurs in mammary cancer cells as they seed early lung metastases, how this influences the cellular machinery controlling EV production, and how these EVs, in turn, can foster a pro-invasive microenvironment. Thus, we provide mechanistic insights that further our understanding of how changes in cellular metabolic programs can coordinate membrane trafficking processes to influence the lung microenvironment in a way that is likely to render this organ more fertile for subsequent metastatic outgrowth. We believe that the mechanistic understanding outlined in the present study will enable further exploration of the metabolic vulnerabilities that may be targeted to oppose metastatic outgrowth of breast cancer in the lung. Finally, it is important to highlight that although the lung tropism of MMTV-*PyMT* metastasis restricts our interpretation to niche priming events in this organ, further work will be necessary to establish potential roles of proline and ceramide metabolism in establishment of metastases in other target organs, such as liver, brain, and bone, to which breast cancer metastasizes.

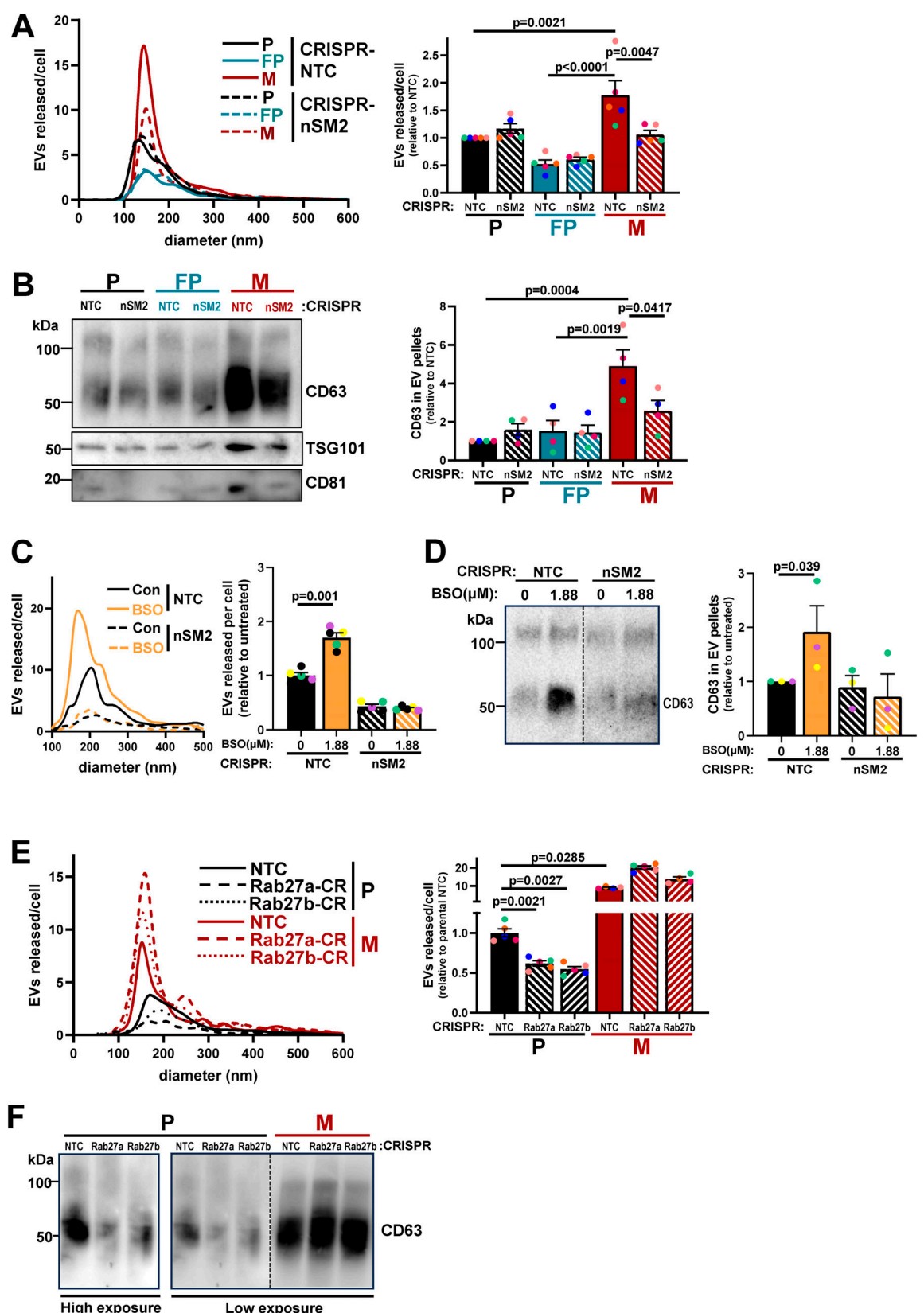

**Figure 7. Micrometastatic cells release EVs via an nSMase2-dependent and Rab27-independent mechanism.** Parental (P), FP, or micrometastatic (M) cells were transduced with a lentiviral vector bearing gRNAs recognizing non-targeting sequences (NTC) or sequences in *nSMase2* (nSM2), *Rab27a*, or *Rab27b*. (See Fig. S3 for confirmation of suppression of *nSMase* [Fig. S3 A] and *Rab27* [Fig. S3B] expression at the gene and protein levels, respectively). **(A–F)** EVs were purified from conditioned medium collected over a 48-h period and analyzed using nanoparticle tracking (A, C, and E) as for Fig. 5 A, and levels of CD63 in EV

pellets were determined by western blotting (B, D, and F). In C–D, P cells transduced with non-targeting (NTC) lentiviral vectors or those targeting *nSMase-2* (nSM2) were incubated with BSO (1.88 µM) during the 48-h EV collection period. Values are mean ± SEM; data were analyzed by one-way ANOVA, n = 5 (A, C, and E), n = 4 (B), n = 3 (D) independent experiments. Source data are available for this figure: SourceData F7.

## Materials and methods

### MMTV-*PyMT* model of mammary cancer

All mice carrying a MMTV promoter-driven *PyMT* transgene were backcrossed >20 generations in an FVB/N background. The MMTV-*PyMT* mice have been described previously (Guy et al., 1992). MMTV-*PyMT* mice (Jackson Laboratory) were housed in individual ventilated cages with environmental enrichment in a barrier facility (12-h light/dark cycle). Monitoring of mice for tumor development was performed two to three times per week. Tumor growth was monitored by caliper measurement, with the clinical endpoint being any tumor reaching 15 mm in diameter when mice were humanely euthanized by Schedule 1 methods. Metastatic burden (number of metastases and metastatic index) in the lungs was assessed by H&E staining. All work was carried out with ethical approval from the local animal welfare (AWERB) committee and performed under the revised Animal (Scientific Procedures) Act 1986 and the EU Directive 2010/63/EU (PP6345023). All animal experiments were performed in accordance with relevant guidelines and regulations as per Animal Research: Reporting of In Vivo Experiments guidelines. For determination of metastatic burden, mice were sacrificed at endpoint (as described above), and lungs were removed, fixed in formalin, and paraffin embedded (FFPE). FFPE lungs were cut into a series of 4-µm serial sections, and metastatic burden assessed by manual assessment of H&E stain in sections 5, 10, and 15 (using HALO image analysis software v3.6.4134.137). For each mouse the average number of metastases identified across the three sections of the lung was recorded, in addition to the average metastatic burden across the three lung sections (defined as [area of lung covered by metastases/total lung area] × 1,000).

### Mammary FP transplantation and resection

For orthotopic transplantation experiments, 8–10-wk-old female FVB/N mice were obtained from Charles River (London, UK). 0.5 × 10⁶ cells generated from MMTV-*PyMT* tumors (either P or P′; see generation of *PyMT* cell lines below) in 50 µl PBS/Matrigel (1:1) were transplanted into the fourth mammary FP (inguinal) of FVB mice (Campbell et al., 2021). Analgesia was used following all surgical procedures, which were carried out under anesthesia. Following recovery from surgery, mice were monitored for tumor development at least three times per week. Once the tumor was palpable, growth was assessed by caliper measurement, and the tumor was surgically removed when the diameter reached 8–10 mm. The tumor was kept in ice-cold PBS prior to downstream processing for cell line generation (see below). After surgical resection, mice were closely monitored (three or more times per week) for clinical signs of metastasis, which included weight loss, altered respiration, or abdominal distension. Animals were humanely euthanized by Schedule 1 methods if they displayed one or more moderate signs (e.g., 20% weight loss and moderate piloerection). At clinical endpoint,

lungs were harvested and kept in ice-cold PBS prior to further processing for the generation of cell lines.

### Generation and maintenance of cell lines

Mammary tumors (FP and FP′ lines) or lungs (M and M′ lines) were minced using sterile surgical scalpels into a fine paste, and the finely chopped tissue was added to culture media (DMEM), followed by centrifugation at 1,000 rpm for 5 min at RT. The cell pellet was resuspended in DMEM, filtered using a 70-µm cell strainer (to remove remaining clumps), and added to a 10-cm dish. The media was supplemented with 10% FBS, 2 mM glutamine, 100 U/ml penicillin streptomycin, 20 ng/ml hEGF, 10 µg/ml insulin, and 0.25 µg/ml fungizone and cultured at 37°C/5% $CO_2$ in a humidified incubator. Prior to mincing, resected lungs were inspected for metastatic lesions using a dissection microscope. Any macrometastases that were visible were dissected from the lung tissue at this stage and minced and cultured independently. The macrometastatic cells (maM′) used in this study originated as a macrometastasis dissected from a lung belonging to a mouse implanted with the P′ parental MMTV-*PyMT* line. To establish mammary tumor cell lines, cells were trypsinized and replated (once they reached around 90% confluence) for at least five passages (representing >25 cell divisions) and maintained in culture for a maximum of 20 passages. The same protocol was used for the generation of metastatic cell lines, with the exception that following centrifugation, the resuspended cell pellet was seeded in 6-well plates. Following this period of ex vivo culture (>25 cell divisions), we found that all cells expressed the *PyMT* transgene and were E-cadherin positive, indicating that they were tumor cells of epithelial origin that had outgrown cells, such as lung parenchymal and stromal cells. For cell culture growth and maintenance, cells were kept in the aforementioned conditions (without fungizone supplementation after the fifth passage) and passaged every 2 days. All cells were regularly tested for mycoplasma contamination using the Venor GeM qONEStep Mycoplasma detection kit (11-91025; Cambio) for qPCR.

### Immunofluorescence

Cells were seeded (3 × 10⁴ cells/dish) in 35-mm glass-bottomed dishes (MatTek) and incubated at 37°C/5% $CO_2$ for 40 h and fixed with 4% paraformaldehyde (in PBS) at RT for 15 min. Cells were permeabilized with 0.1% Triton X-100 (in PBS) at RT for 15 min and blocked in 1% bovine serum albumin (in PBS) for 1 h. Cells were then incubated with an antibody recognizing ceramides (ALX-804-196; dilution 1:100; Enzo) in blocking solution at 4°C overnight, followed by secondary antibody (Alexa Fluor 488 anti-Mouse IgM, [A-21042], dilution 1:500; Thermo Fisher Scientific) and phalloidin stain (Alexa Fluor 555, [A-34055], 1:1,000 dilution in PBS; Thermo Fisher Scientific). Cells were then mounted using Vectashield (containing DAPI; Vectashield Laboratories) and visualized by confocal microscopy. Airyscan

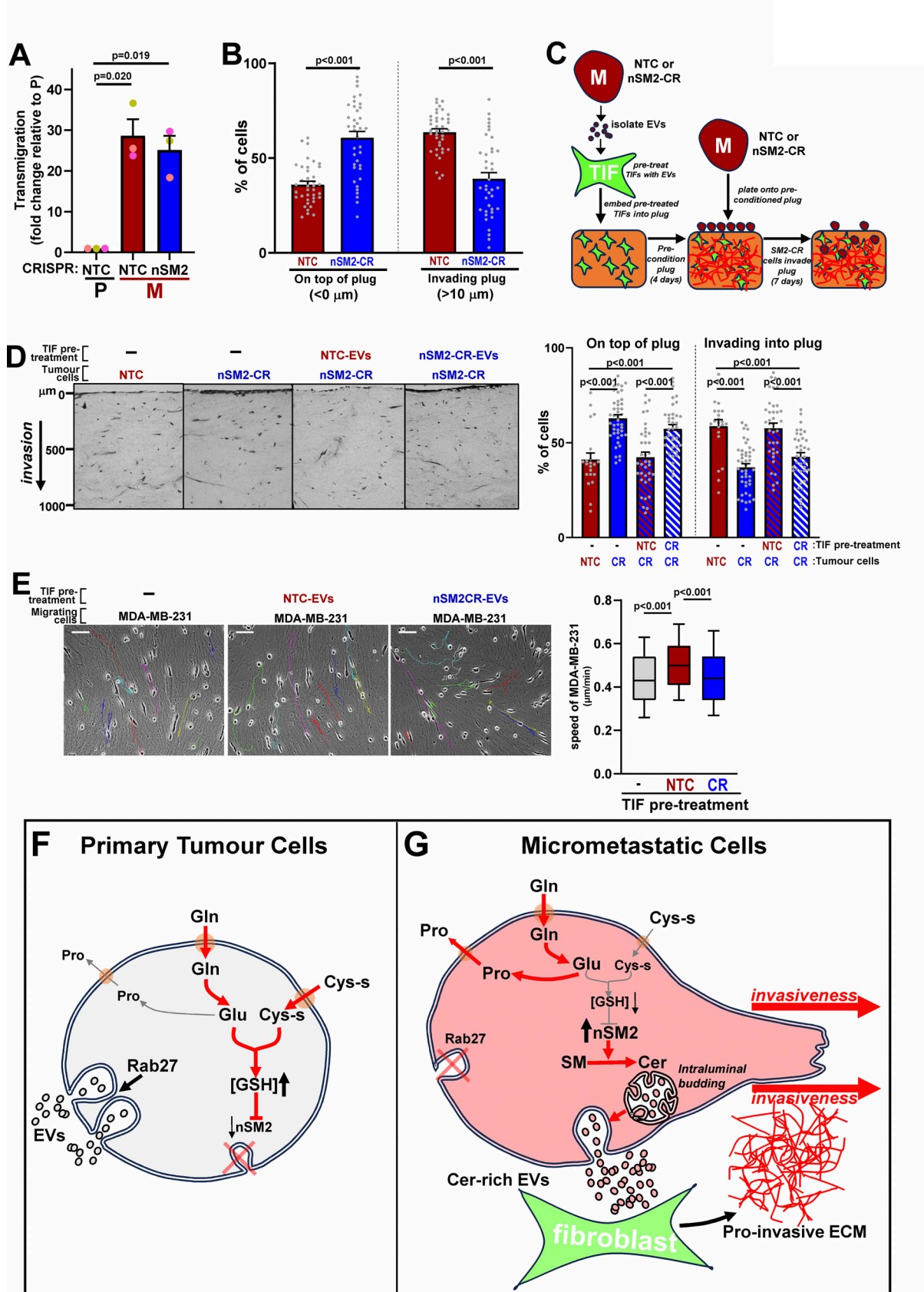

Figure 8. **nSMase2-dependent EV release favors generation of a pro-invasive microenvironment. (A and B)** Micrometastatic (M) cells were transduced with lentiviruses bearing gRNAs recognizing non-targeting sequences (NTC) or sequences in *nSMase2* (nSM2-CR). Transduced cells were plated into the upper chamber of Transwells and transmigration determined as for Fig. 1 C (A). Values are mean ± SEM, *n* = 3 independent experiments (colored dots); data were

analyzed using a paired *t* test. Alternatively, transduced cells were plated onto fibroblast preconditioned collagen plugs, and invasion into these was determined as for Fig. 1 E (B). Values are mean ± SEM, two plugs/condition, *n* = 39–47 fields of view; data were analyzed by mixed effects ANOVA with Tukey's multiple comparison test. **(C)** Schematic depiction of protocol for determining EVs' influence on the invasive microenvironment of organotypic collagen plugs. EVs released by control (NTC) or *nSMase-2* CRISPR (nSM2-CR) M cells were incubated with TIFs for 72 h. EV-treated TIFs were then allowed to precondition organotypic plugs of rat tail collagen for 4 days. NTC or nSM2-CR M cells were then plated onto preconditioned plugs and allowed to invade for 7 days. **(D)** Control (NTC) or *nSMase-2* CRISPR (nSM2-CR) M cells were plated onto organotypic collagen plugs that had been preconditioned as outlined in C. Invasion was quantified as for Fig. 1 E. Values are mean ± SEM, *n* > 22 fields of view/condition (*n* = 2 independent experiments); data were analyzed using mixed effects ANOVA with Tukey's multiple comparison test. **(E)** TIFs were incubated with EVs from control (NTC) or *nSMase-2* CRISPR (nSM2-CR) M cells for 72 h or were left untreated. EV pre-treated TIFs were then replated and allowed to generate ECM for 7 days, which was then de-cellularized. MDA-MB-231 breast cancer cells were plated onto de-cellularized ECMs, and time-lapse microscopy (over 16 h) followed by cell tracking (ImageJ) was employed to measure their migration (*n* > 100 cells, 2 independent experiments). Colored lines indicate representative tracks of individual migrating cells (left panels). Bar, 100 μm. Box and whisker plots are 10–90 percentile, and the line displays median. Data were analyzed by Kruskal–Wallis test with Dunn's multiple comparisons test. **(F and G)** Schematic summary of how metabolic control of EV biogenesis in cells from primary mammary tumors and lung micrometastases may influence ECM microenvironments. Cells from primary tumors synthesize glutathione using glutamine-derived carbons and release EVs in a Rab27-dependent/nSMase2-independent manner (F). In micrometastatic cells, more glutamine-derived carbons are used for proline production, leading to decreased glutathione synthesis. This, in combination with increased nSMase2 (nSM2)-dependent ceramide production, promotes nSM2-dependent intraluminal vesicle budding and the release of the resulting EVs in a Rab27-independent way, which encourages fibroblasts to generate a more invasive microenvironment (G) (Gln, glutamine; Glu, glutamate; Cys-s, cystine; GSH, reduced glutathione; Pro, proline; SM, sphingomyelin; Cer, ceramide).

images were collected on a Zeiss 880 point-scanning confocal microscope, built on an inverted Zeiss Axio Observer.Z1 stand. Airyscan images were acquired using a 63×/1.4 Plan-Apochromat objective lens with a confocal pinhole diameter of 156 μm and the Airyscan v1.0 array detector (Zeiss). Multi-channel images were captured sequentially: DAPI using 405-nm excitation and BP 420–480 + LP 605-nm emission filter, ceramide using 488-nm excitation and BP 420–480 + BP 495–550-nm emission, and phalloidin using 561-nm excitation and BP 420–using 480 + LP 605-nm emission. Images were collected with a 1.8× zoom with an image size of 2,124 × 2,124 pixels, yielding a pixel size of 0.035 × 0.035 μm and a 1.98-μs pixel dwell time. Z-stacks were collected using a step size of 0.37 μm. Images were acquired and processed using the software Zen LSM 2.1 Black (Zeiss), using the Airyscan processing module set to "Auto." Quantification of ceramide intensity (green channel) was performed on projections of cells per field of view (sum of z-slices; 10 slices/field of view, 6 fields of view per biological replicate) and normalized to phalloidin area (red channel) using ImageJ software (Schneider et al., 2012).

De-cellularized ECM, which were generated by TIFs that had been either pre-treated or not with EVs from control or nSM2-CR M cells for 72 h (see below), were fixed with 4% paraformaldehyde (in PBS) for 15 min and blocked in 1% bovine serum albumin (in PBS) for 1 h. ECM was then incubated with an antibody recognizing fibronectin (#610078; dilution 1:100; BD transduction Laboratories) in blocking solution at 4°C overnight, followed by incubation with secondary antibody (Alexa Fluor 488 anti-Mouse IgG, [A-21202], dilution 1:400; Thermo Fisher Scientific). Fibronectin-stained ECM was then imaged on an Opera Phenix high-content spinning disk confocal microscope using a 20×/1.0 NA Plan Apochromat water immersion objective and Harmony 5.0 acquisition software. Fibronectin images were captured using 488-nm excitation and a 500–550-nm emission filter and 300-ms exposure time. Images were acquired with an Andor Xyla camera using 2x binning, resulting in pixel size 0.6 × 0.6 μm and image size 1,080 × 1,080 pixels. At each position, four planes were captured with 2-μm spacing, and the resulting images were flat-field corrected and exported as TIFF files using

an Acapella script, then maximum intensity projected using ImageJ. Fibronectin fiber analysis was then performed by running fluorescent fibronectin images through the FIJI macro TWOMBLI (Wershof et al., 2021) with the following parameters: contrast saturation 0.35, line width 5, curvature window 20, and minimum branch length 10.

## ISH

FFPE tissues were obtained from MMTV-*PyMT* primary mammary tumors and matched metastatic lungs. FFPE tissue was cut into 4-μm sections, which were then heated to 60°C for 2 h. H&E staining was performed on a Leica autostainer, where sections were dewaxed, taken through graded alcohols, and stained with Haem Z (CellPath) for 13 min. Sections were then washed in tap water, differentiated in 1% acid alcohol, washed, and the nuclei blued in Scott's tap water substitute (in-house). After washing, sections were placed in Putt's eosin (in-house) for 3 min. ISH detection for Slc7a11 (xCT) mRNA was performed using RNAScope 2.5 LSx (Brown) detection kit (Bio-Techne) on a Leica Bond Rx autostainer strictly according to the manufacturer's instructions. To complete H&E and ISH staining, sections were rinsed in tap water, dehydrated through graded ethanols, and placed in xylene. The stained sections were mounted in xylene using DPX mountant (CellPath). Following staining, slides were scanned at 20× magnification using NanoZoomer NDP scanner (Hamamatsu), and HALO software (Indica Labs) was used to quantify the average optical density of the *Slc7a11* stain in sections of primary mammary tumors and corresponding lung metastases.

## Western blotting

Proteins were resolved in 4–12% (NP0335BOX; Thermo Fisher Scientific) or 10% (NP0315BOX) precast NuPAGE Bis-Tris gels using MOPS running buffer (NP0001) at 120 V. Resolved proteins were transferred to a PVDF membrane (IPVH85R; Millipore) in NuPAGE transfer buffer (NP00061; Thermo Fisher Scientific) at 120 V for 90 min. Following blocking (in 4% milk), membranes were incubated with primary antibodies diluted in 1% milk overnight at 4°C while gentle agitation was applied. Primary antibodies were anti-CD63, (ab217345), dilution 1:1,000;

Abcam; anti-CD81, (sc-166029), dilution 1:100; Santa-Cruz; anti-TSG101, (GTX70255), dilution 1:1,000; GeneTex; and anti–β-actin, (A1978), dilution 1:5,000; Sigma-Aldrich. Following incubation with appropriate HRP-conjugated secondary antibodies (dilution 1:5,000 for HRP-linked anti-rabbit IgG, #7074; CST and HRP-linked anti-mouse IgG, #7076; CST) protein bands were visualized using enhanced chemiluminescence (34095; SuperSignal West Femto; Thermo Fisher Scientific) in a Bio-Rad ChemiDoc imaging system. Densitometry analysis of CD63 and TSG101 was performed at non-saturating exposures while applying the same background settings across the samples using Image Lab software (Bio-Rad).

## qPCR

RT quantitative real-time PCR (RT-qPCR) reactions were performed in 96-well plates (Bio-Rad CFX platform) by using SYBR Green in a total reaction volume of 10 μl per well. Each reaction contained a 1/20 dilution of cDNA template, 1X PerfeCTa SYBR Green Fast mix, and 0.5-μM forward and reverse primers, and samples were plated in duplicate.

Primer sequences were as follows:

ARPP P0 (forward) 5′-GCACTGGAAGTCCAACTACTTC-3′
ARPP P0 (reverse) 5′-TGAGGTCCTCCTTGGTGAACAC-3′
Gclc (forward) 5′-ACACCTGGATGATGCCAACGAG-3′
Gclc (reverse) 5′-CCTCCATTGGTCGGAACTCTAC-3′
PyMT (forward) 5′-CTGCTACTGCACCCAGACAA-3′
PyMT (reverse) 5′-GCAGGTAAGAGGCATTCTGC-3′
SLC7A11 (xCT) (forward) 5′-CTTTGTTGCCCTCTCCTGCTTC-3′
SLC7A11 (xCT) (reverse) 5′-CAGAGGAGTGTGCTTGTGGACA-3′
Smpd2 (neutral sphingomyelinase-1) (forward) 5′-CATCCCCTACCTGAGCAAAC-3′
Smpd2 (neutral sphingomyelinase-1) (reverse) 5′-CCAGGAGAGCCAGATCAAAGTT-3′
Smpd3 (neutral sphingomyelinase-2) (forward) 5′-TGGTGGTGTTTGACGTCATCT-3′
Smpd3 (neutral sphingomyelinase-2) (reverse) 5′-GCGAGTAAAGAGCGAGTGCT-3′

Standard curves were generated using serially diluted cDNA from pooled samples to assess reaction efficiency, and no-template controls (without cDNA) were also included for each target gene to check for potential primer dimer formation. Reactions were run on the BioRad C1000 thermal cycler using a three-step protocol, during which cDNA was denatured for 3 min at 95°C, followed by 40 cycles of denaturation at 95°C for 20 s, annealing at 60°C for 20 s, and extension at 72°C for 20 s, with a final extension step of 72°C for 5 min and a melting curve from 65 to 95°C in 0.5°C increments. Data acquisition and analysis were performed in CFX Manager Software, and the ΔΔCt method was used to compare gene expression levels between different cell lines. In all analyses, the ARPP P0 mRNA was used as an internal control.

## Digital-drop PCR

The levels of mitochondrial (mitochondrially and nuclear-encoded) genes (presented in Fig. S5, A–C) were determined using digital-drop PCR. This was performed as described by Mahmood and collaborators (Mahmood et al., 2024). Briefly, samples were prepared in triplicate in a 96-well plate using 4 μl of DNA, 100 nM of each primer, 10 μl of QX200 ddPCR EvaGreen Supermix, and water to a final volume of 20 μl. Droplet generation and PCR and measurements were then performed on the QX200 Droplet Digital PCR System (Bio-Rad Laboratories) as per the manufacturer's instructions, with the primer annealing temperature set at 60°C.

Primers for ND1 and ND2 were described by Rabas et al. (2021). Primers for ND5 and VDAC1 were as follows:

ND5 (forward) 5′-TGCCTAGTAATCGGAAGCCTCGC-3′
ND5 (reverse) 5′-TCAGGCGTTGGTGTTGCAGG-3′
VDAC1 (forward) 5′-CTCCCACATACGCCGATCTT-3′
VDAC1 (reverse) 5′-GCCGTAGCCCTTGGTGAAG-3′

## Cell proliferation

Cells were seeded at $2 \times 10^4$ cells per well in 6-well culture plates and incubated at 37°C/5% $CO_2$ in a humidified incubator for 6 h to adhere. Plates were transferred to a tissue culture incubator equipped with an IncuCyte ZOOM live-cell imaging system, and images were acquired (10× objective) every 2 h over a period of 120 h (5 days). At the end of the incubation, images were analyzed using the Incucyte Analysis Software, and confluence was measured using the same phase segmentation parameters across all *PyMT*-derived cell lines. Confluence was calculated as a % of the phase image area covered by cells (three technical replicates or fields of view/condition) corresponding to individual time points, and these values (60 time points) were plotted over the culture period to generate the growth curves for each cell line.

## Generation of nSMase and Rab27 CRISPR cells

gRNA sequences targeting neutral sphingomyelinases 1 and 2 and Rab27s a and b were cloned into a lentiCRISPR vector (Addgene [52961]) (O'Prey et al., 2017; Shalem et al., 2014). gRNAs were as follows:

Rab27a (forward) 5′-CACCG CCAGGAGCATCTCAATCGCG-3′;
Rab27a (reverse) 5′-AAACCGCGATTGAGATG CTCCTGGC-3′;
Rab27b (forward) 5′-CACCGGCTGCGCTTGTTTCGAAGTA-3′;
Rab27b (reverse) 5′-AAACTACTTCGAAACAAGCGCAGCC-3′;
neutral sphingomyelinase-1 (forward) 5′-CACCGCGCCCTATGTTTGCTCAGGT-3′;
neutral sphingomyelinase-1 (reverse) 5′-AAACACCTGAGCAAACATAGGGCGC-3′;
neutral sphingomyelinase-2 (forward) 5′-CACCGGTGGTGTTTGACGTCATCTG-3′;
neutral sphingomyelinase-2 (reverse) 5′-AAACCAGATGACGTCAAACACCACC-3′.

Then, to produce lentivirus-encoding Cas9, along with the desired gRNAs, HEK 293T cells were used as the host-packaging cell line (1 million cells were seeded overnight and transfected on the next day with 10 μg of plentiCRISPR for each of the gRNAs, 7.5 μg psPAX2, #12260; Addgene, and 4 μg pVSV-G, #138479; Addgene). Virus-containing HEK 293T supernatant was then filtered through a 0.45-μm PTFE filter membrane, polybrene supplemented to a final concentration of 4 μg/ml,

and the medium incubated with recipient P, FP, or M cells overnight at 37°C/5% CO$_2$. Virus-transfected cells were then selected using puromycin (2 µg/ml), the antibiotic selection medium being replenished every 2 days, and the cells were passaged according to their confluence to generate stable cell lines. Seven passages after infection, cells were harvested for qPCR analysis of *nSMases* and western blot for Rab27s.

## Metabolite extraction, calculation of metabolite exchange, and metabolic tracing

Cells were seeded in 6-well plates (three technical replicates/cell line) in full culture medium (2 ml) at $1.5 \times 10^4$ cells per well. After 24 h (day 1), 2 ml of culture medium was added to each well to prevent nutrient exhaustion. On day 2, the culture medium was changed to 2 or 7 ml for medium metabolite extractions (calculation of exchange rates) or intracellular metabolite extractions, respectively. For the metabolic tracing experiments, DMEM in which $^{13}C_5$-labelled glutamine (CLM-1822-H, Cambridge Isotope Laboratories) was substituted for unlabelled glutamine (2 mM) was added at this stage at the indicated concentrations. The volumes of culture medium used enabled measurement of exchange rates of nutrients with slow flux while detecting intracellular metabolites that are rapidly consumed. In parallel, at this time point (day 2), cells were harvested for determination of total cell number per condition. After 24 h (day 3), media and intracellular metabolites were extracted. For extraction of metabolites from culture media, 20 µl of medium was added to 980 µl of an ice-cold polar extraction solution (methanol, acetonitrile [ACN], and water in a 5:3:2 ratio), and the samples were then shaken in a thermomixer at 1,400 rpm for 10 min (4°C). Metabolites were extracted from serum and plasma using the same approach as for cell culture medium. For cell extracts, cells were quickly washed three times with ice-cold PBS, followed by extraction with 600 µl of ice-cold extraction solution for 5 min at 4°C, while gentle agitation was applied. To remove insoluble material, media and cell extracts were centrifuged at 16,000 *g* for 10 min, and the supernatant was transferred to glass vials, which were stored at –80°C prior to liquid chromatography-mass spectrometry (LC-MS) analysis.

Lipids were extracted from cells using 600 µl, or from EV pellets using 50 µl, of a solution composed of butanol and methanol (1:1), supplemented with Splash II standard (330709; Avanti). Lipid cell extracts were centrifuged at 16,000 *g* for 10 min to remove insoluble material, and the supernatant was transferred to glass vials, which were stored at –80°C prior to LC-MS analysis.

To allow estimation of consumption/secretion rates of nutrients and metabolites, data normalization was achieved by counting the total number (#) of cells on days 2 and 3, and the exchange rate per day for each metabolite was calculated according to the following equation: $x = \frac{\Delta\text{metabolite}}{(\#\text{cell day2}+\#\text{cell day3})/2}$, where Δmetabolite= (detected peak area spent medium - detected peak area cell-free medium), and whether the x value is positive or negative indicates that the metabolite is secreted or consumed by the cells, respectively (Vande Voorde et al., 2019).

## LC-MS metabolomics and data analysis
All polar metabolites were analyzed by LC-MS as previously described in Villar et al. (2023).

### Polar metabolites
Metabolites from the biological extracts (5 µl) were separated using a ZIC-pHILIC guard and analytical column (SeQuant; 150 mm by 2.1 mm, 5 µm; Merck) on an Ultimate 3000 HPLC system (Thermo Fisher Scientific). Chromatographic separation was performed using a 15-min linear gradient from 20% ammonium carbonate (20 mM [pH 9.2]) and 80% ACN to 20% ACN at a constant flow rate of 200 µl/min. The column temperature was constant at 45°C. A Q Exactive orbitrap mass spectrometer (Thermo Fisher Scientific) equipped with electrospray ionization was coupled to the HPLC system; polarity switching mode was used with a resolution of 70,000 at 200 mass/charge ratio (m/z) to enable both positive and negative ions to be detected across a mass range of 75–1,000 m/z (automatic gain control target 1e6 and maximal injection time 250 ms). Polar metabolomics analysis was performed using TraceFinder (version 4.1; Thermo Fisher Scientific). Extracted ion chromatograms were generated for each compound and its isotopologues using the m/z of the singly charged ions (XIC, ±5 ppm) and the RT ( ±2 min) from our in-house metabolite library, which was generated using reference standards on the same LC-MS method. Thiols were preserved as previously described in Sun et al. (2020). Standards were derivatized and analyzed with the same methodology to generate optimized XIC parameters. All peak areas were normalized to total cell number, e.g., peak area per million cells.

### Lipidomics (targeted analysis)
Lipid analysis was performed using an Ultimate 3000 HPLC (Thermo Fisher Scientific) coupled to a Q-Exactive Orbitrap mass spectrometer (Thermo Fisher Scientific). Lipids were separated on an Acquity UPLC CSH C18 column (100 × 2.1 mm; 1.7 µm; Waters Corporation) maintained at 60°C. The mobile phases consisted of 60:40 ACN: H$_2$O with 10 mM ammonium formate, 0.1% formic acid, and 5 µM of phosphoric acid (A) and 90:10 IPA:ACN with 10 mM ammonium formate, 0.1% formic acid, and 5 µM phosphoric acid (B). The gradient was as follows: 0–2 min 30% (B), 2–8 min 50% (B), 8–15 min 99% (B), 15–16 min 99% (B), and 16–17 min 30% (B). Sample temperature was maintained at 6°C in the autosampler, and 5 µl of sample were injected into the LC-MS instrument. Thermo Q-Exactive Orbitrap MS instrument was operated in both positive and negative polarities over the following mass range: 240–1,200 m/z (positive) and 240–1,600 (negative) at resolution 70,000. Data-dependent fragmentation (dd-MS/MS) was carried out for each polarity. Feature detection and peak alignment from .Raw files were performed using Compound Discoverer 3.1 (Thermo Fisher Scientific). Files were also converted to .mgf format using MSConvert software (ProteoWizard) and searched against the LipiDex_ULCFA database using LipiDex software (Hutchins et al., 2018). Data preprocessing, filtering, and basic statistics were performed using Perseus software (Tyanova et al.,

2016) version 1.6.2.2 after importing the final results table from LipiDex.

## Lipidomics (untargeted analysis of ceramides)

A lipidomic platform was established to identify and measure the ceramides present in the biological extracts using tandem mass spectrometry. The analysis was performed using an Ultimate 3000 HPLC (Thermo Fisher Scientific) coupled to a TSQ Altis triple quadrupole mass spectrometer. The method used the same chromatographic conditions applied in the untargeted lipidomic analysis. MS was performed employing SRM lists generated using LipidCreator version 1.2.1 (Peng et al., 2020). SRM lists included all possible fragments for species with saturated or unsaturated fatty acids of chain length (C16–C20) based on preliminary experiments on the abundance of ceramide species of variable chain lengths. After peak deconvolution and integration of the chromatograms using TF software (version 4.1; Thermo Fisher Scientific). Initially, 1,579 peaks were detected; species were considered only if they showed fragments for both the sphingoid units and the expected acyl chains, all at the same retention time. This strategy was applied for structural identification of ceramides. Data cleaning and filtering were all done using R software. Species were excluded if detected in blanks or were of ≥40% missingness, with relative standard error of >20%. This approach narrowed down the number of detected species across the groups to 12 species. Fold change analysis was performed and 1.5 was selected as an arbitrary threshold in the metastatic compared with parental and FPs, which left only four species found to be significantly different in the metastatic group versus control.

## Plasma from metastatic breast cancer patients and healthy volunteers

Plasma samples from patients with metastatic breast cancer and female healthy volunteers were originally collected under a protocol approved by West of Scotland REC4 (reference 10/S0704/32). Analysis of these archival samples for the current research was approved by the Office for Research Ethics Committees Northern Ireland HSC REC B (reference 17/NI/0228).

## Organotypic collagen plug invasion assay

Organotypic plugs were generated from rat tail-derived collagen I, as previously described (Timpson et al., 2011). Briefly, $1 \times 10^6$ confluent TIFs were mixed with collagen I (2 mg/ml) under neutral or slightly acidic conditions (MEM, approximate pH = 7.2 by using 0.22 M NaOH), and the collagen/fibroblast matrix was allowed to contract for ~4 days, until it fit in a 24-well culture dish. For some experiments, TIFs were pre-treated with EVs for 72 h. EVs purified by differential centrifugation were resuspended in DMEM containing 10% EV-depleted fetal calf serum. The quantity of EVs used for TIF pre-treatment was equivalent to that present in the cell-conditioned medium from which they were purified. Following EV pre-treatment, TIFs were washed extensively in PBS to remove EV-containing medium, trypsinized, and then mixed with collagen I. $3 \times 10^3$ parental (P), FP, or micrometastatic (M) cells were seeded on top of these plugs (in duplicates) and cultured for 2 days. Plugs

were then transferred on top of a metal grid and cultured in full medium by creating an air-liquid interface (invasion day 0) for 7 days. At the end of the incubation time, plugs were cut in half and fixed in 4% paraformaldehyde before paraffin embedding (proper orientation of cross sections), and 4 µm sections were then cut and stained (H&E). Single-cell tracking was performed manually using ImageJ software under enhanced brightness/contrast settings (black and white mode), and a distance value, along with an angle value (representing the orientation relative to the invasion baseline; negative angle: cell entering the plug, positive angle: cell residing on top of the plug) was assigned to each cell per field of view. Images were acquired using a brightfield microscope (Zeiss Axio Imager.A2) with Axiocam 105 using a 10×/0.3 Plan Neofluar objective, and raw data were exported in batches (.ome.tiff) using the Zeiss Zen lite software (blue edition).

## TIF-derived ECM

TIFs ($2 \times 10^5$ cells/well) that had previously been exposed to EVs for 72 h were washed, trypsinized, and then reseeded in 6-well plates coated with gelatin, and cells were incubated at 37°C/5% $CO_2$. EV quantity used for TIF pre-treatment was adjusted as described in the previous section describing the organotypic collagen plug assays. Cell-derived matrices (CDMs) were then generated as described in Bass et al. (2007); Cukierman (2005). Once cells reached confluence, the medium was changed to full medium supplemented with 50 µg/ml ascorbic acid, which was refreshed every other day for 7 days. ECMs/CDMs were denuded by incubation in PBS containing 20 mM $NH_4OH$ and 0.5% Triton X-100 (2 min at RT), residual DNA was digested with DNase I (10 µg/ml in PBS containing calcium and magnesium; D-PBS), and denuded ECMs/CDMs were stored at 4°C. Before use, ECMs/CDMs were washed twice in D-PBS and incubated with full medium for 30 min at 37°C/5% $CO_2$. MDA-MB-231 cells were seeded at $0.8 \times 10^5$ cells per well and imaged in time-lapse for 16 h using an inverted phase-contrast microscope (Ti Eclipse, Plan Fluor 10×/0.3 objective; Nikon), while kept at 37°C/5% $CO_2$ (6 fields of view per well were imaged every 10 min for 16 h). After completion of image acquisition, movies were analyzed using ImageJ software (cytokinesis plugin) to determine cell migration speed.

## Transwell migration assay

These assays were performed in Boyden Chambers (3422; Corning; Merck; Transwell Costar, 6.5-mm diameter, 8-µm pore size) according to the manufacturer's instructions (Justus et al., 2023). Briefly, to assess migration, the lower faces of the inserts were coated with 2 µg/ml fibronectin (F0895; Merck). Subsequently, cells ($1 \times 10^5$ cells/well) were resuspended in serum-free medium and added to the upper chamber, while 10% FBS-containing medium was added to the lower chamber. 2 h later, inserts were removed, and the upper face of each insert was washed to remove cells that did not transmigrate. Cells that had transmigrated to the lower chamber were stained with 0.1% crystal violet in 2% ethanol and visualized. Brightfield images were collected on an inverted microscope (Nikon Ts2) with a 20×/0.40 Plan Fluor objective (Nikon). Images were acquired

using a Nikon DS-Fi3 CMOS camera and NIS-Elements F 5.22.00 (Nikon) software.

## EV collection and nanoparticle tracking analysis

EV-free medium was prepared by overnight (16 h) ultracentrifugation of 5% FBS-containing DMEM at 100,000 $g$ (4°C) and further filtration of the collected supernatant using a 0.2-μm PES filter unit. $0.3 \times 10^6$ cells were then seeded in 15-cm dishes (three 15-cm dishes/condition) and cultured in full medium for 16 h at 37°C/5% $CO_2$. Cells were then washed twice with PBS, medium was changed to EV-free medium (15 ml/dish), and cultured for 48 h at 37°C/5% $CO_2$. This conditioned medium was then collected and subjected to sequential centrifugation steps to remove live cells (300 $g$ for 10 min), dead cells (2,000 $g$ for 10 min), and finally to remove cell debris and larger lipid membrane particles, including the majority of microvesicles (10,000 $g$ for 20 min). EVs were then pelleted in thin-wall polypropylene tubes after ultracentrifugation at 100,000 $g$ for 70 min using an SW32 rotor (Beckman coulter). The EV pellet was then washed in filtered PBS (0.2-μm PES) and subjected to a final ultracentrifugation step at 100,000 $g$ for 70 min. All centrifugation steps were performed at 4°C. At the end of the last spin, the supernatant was carefully aspirated by tilting the tube, EVs were resuspended in 150 μl PBS, which was previously filtered using a Whatman Anotop filter (0.02 μm), and stored at 4°C.

Nanoparticle tracking analysis was performed using a NanoSight LM10 instrument (Malvern Panalytical) with a high-sensitivity camera according to the manufacturer's instructions. Isolated EVs were diluted 1:50–1:200 in PBS (filtered through 0.02 μm). Diluted EV samples were then flushed through the chamber until vesicles were visible on the camera by using a 1-ml syringe. The focus and gain settings were optimized for each run and kept consistent across the samples, which were injected into the flow cell, and five recordings of 60 s each were acquired. The chamber was washed with ethanol and deionized water between samples to ensure no residual particles remained. The data were analyzed using the NTA3.1 analysis software, and averages of technical replicates were plotted per experiment following normalization to the total number of EV-releasing cells and the dilution applied per condition.

## Statistical analyses

Statistical analyses were performed with GraphPad Prism 9 on all relevant experiments using unpaired or paired $t$ tests to compare two groups with normal data distribution. ANOVA tests (one-way, two-way, or repeated measures) were used to compare more than two groups if the data were normally distributed, while a Kruskal–Wallis test was performed for not normally distributed datasets. A Friedman test (nonparametric repeated measures ANOVA) was used when no assumptions were made for the data distribution. Data distribution was assumed to be normal, but this was not formally tested except for the graphs in Fig. 3 E, Fig. 5 E, and Fig. 8 E. In this case, the data were tested for normal distribution using the Kolmogorov–Smirnov test, the Shapiro–Wilk test, and the Anderson–Darling test. Since the P value was <0.05 in one of the aforementioned

tests, the data in Fig. 3 E, Fig. 5 E, and Fig. 8 E were assumed not to be normally distributed. Statistical significance is annotated in the figures (P values are shown on the figures, with P <0.05 considered significant), and the associated tests are indicated in each figure legend.

## Online supplemental material

The supplemental materials are as follows: Fig. S1 shows the comparison of metabolite levels of cells from micrometastases and frank metastases with their primary tumor counterparts. Fig. S2 shows the treatment of parental (P) cells with BSO. Fig. S3 shows the validation of CRISPR disruption of nSMases and Rab27s and EV release from nSMase-1 CRISPR cells. Fig. S4 shows that the PYCR activity is not required for transmigration of MMTV-PyMT mammary cancer cells and serine/glycine levels in mammary cancer cells from primary tumor and micrometastases and following treatment with BSO. Fig. S5 shows that the micrometastatic (M) cells release less EV-associated mtDNA than cells from primary tumors (P), and EVs from micrometastatic cells do not influence the fibronectin filament length or organization in the ECM deposited by fibroblasts.

## Data availability

The data supporting the findings of this study are available within the article and its supplementary information files and from the corresponding author upon request. The LC-MS metabolomic and lipidomic data of this study are deposited on the MassIVE repository, computer science and engineering, the University of California, San Diego, CA, USA. Dataset ID: MassIVE MSV000097473 publicly available at https://massive.ucsd.edu/ProteoSAFe/dataset.jsp?accession=MSV000097473. Assistance with navigating this MassIVE dataset may be obtained from the corresponding authors upon request.

## Acknowledgments

We would like to thank Anna Koessinger for assistance with excision of lung macrometastases and core staff in the Biological Services Unit, the Beatson Advanced Imaging Resource (Research Resource Identifier: SCR_023875), Molecular Services, Histology Facility, Metabolomic Facility, and Central Services (CRUK Scotland Institute) for their support, which facilitated the work described in this manuscript. This paper was critically reviewed by Catherine Winchester (CRUK Scotland Institute).

This work was funded by Cancer Research UK core program funding to J.C. Norman (A18277 and A28291), S. Tardito (A23982), and K. Blyth (A29799); Breast Cancer Now (2019NovPR1268); the Medical Research Council (MR/P01058X/1); C.J. Clarke (the Pancreatic Cancer UK Research Innovation Fund [2021RIF_22_Voorde]); the Wellcome Trust ISSF (318046); and CRUK (PRCBTP-May24/100005). We acknowledge the Cancer Research UK Glasgow Centre (C596/A18076) and the BSU facilities at the Cancer Research UK Scotland Institute (C596/A17196 and A31287).

Author contributions: M. Gounis: conceptualization, data curation, formal analysis, investigation, methodology, validation, visualization, and writing—original draft, review, and

editing. A.V. Campos: data curation, formal analysis, investigation, methodology, project administration, validation, visualization, and writing—review and editing. E. Shokry: formal analysis, investigation, and writing—review and editing. L. Mitchell: investigation and methodology. R. Deshmukh: investigation. E. Dornier: investigation and methodology. N. Rooney: methodology. S. Dhayade: investigation. L. Pardo: investigation. M. Moore: formal analysis and investigation. D. Novo: investigation and methodology. J. Mowat: resources. C. Jamieson: resources. E. Kay: resources. S. Zanivan: resources. N.R. Paul: investigation. C. Mitchell: formal analysis, investigation, and writing—review and editing. C. Nixon: investigation. I. Macpherson: funding acquisition, investigation, and writing—review and editing. S. Tardito: data curation and supervision. D. Sumpton: conceptualization, investigation, and writing—review and editing. K. Blyth: conceptualization, funding acquisition, investigation, project administration, supervision, and writing—review and editing. J.C. Norman: conceptualization, formal analysis, funding acquisition, investigation, methodology, project administration, resources, supervision, visualization, and writing—original draft, review, and editing. C.J. Clarke: conceptualization, formal analysis, funding acquisition, investigation, project administration, supervision, and writing—original draft, review, and editing.

Disclosures: The authors declare no competing interests exist.

Submitted: 13 May 2024

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

# Supplemental material

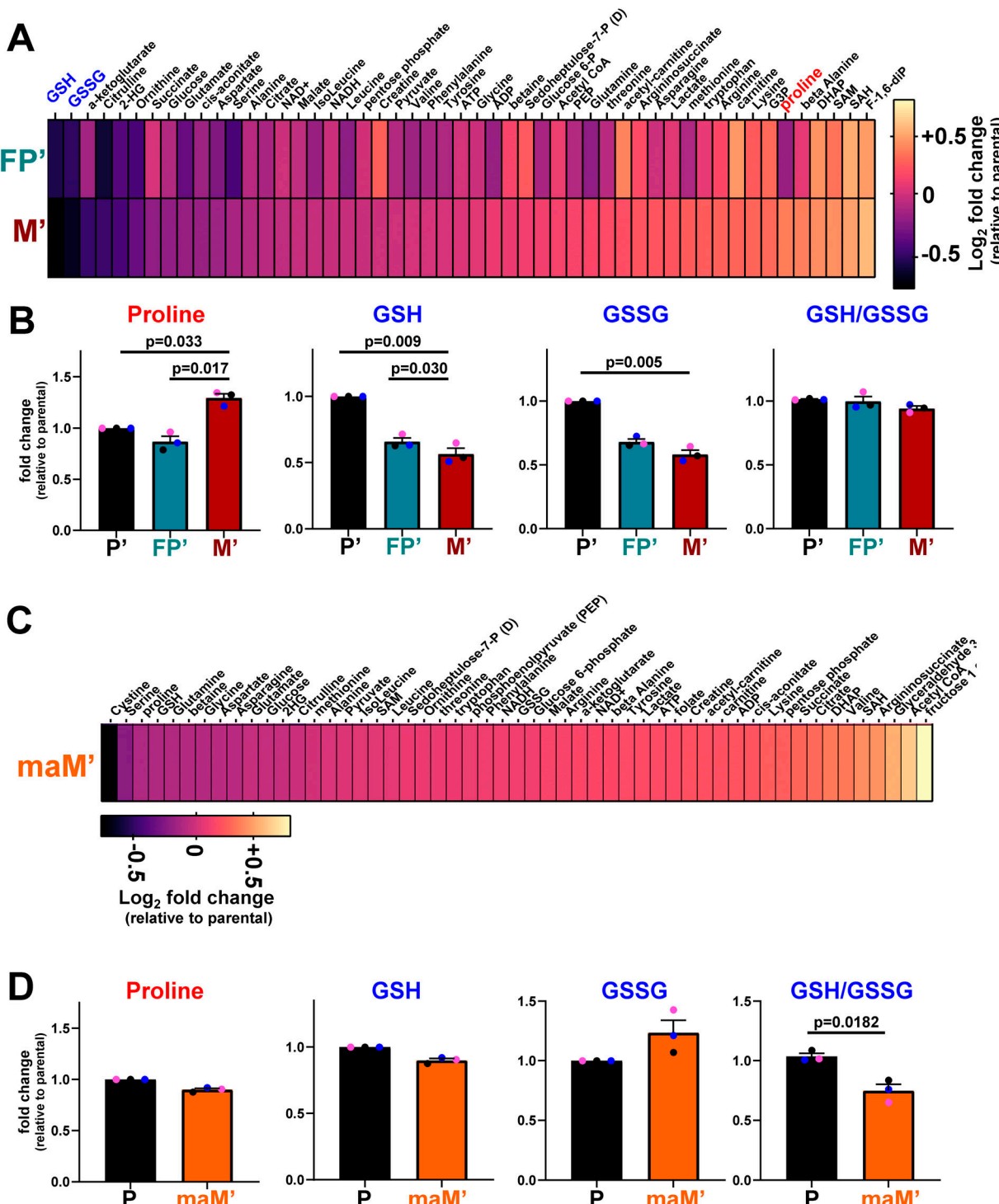

Figure S1. **Comparison of metabolite levels of cells from micrometastases and frank metastases with their primary tumor counterparts. (A–D)** The abundance of intracellular metabolites in fat pad' (FP'), micrometastatic' (M') (A and B) and frank macrometastatic (maM') (C and D) cells was determined using LC-MS and expressed relative to levels of the same metabolites in parental (P') cells. For the heatmaps (A and C), values are log₂-fold changes, and for the graphs (B and D), values are mean fold change ± SEM; statistics are repeated measures one-way ANOVA with Tukey's multiple comparison test, $n = 3$ independent biological replicates (each colored dot represents an individual experiment).

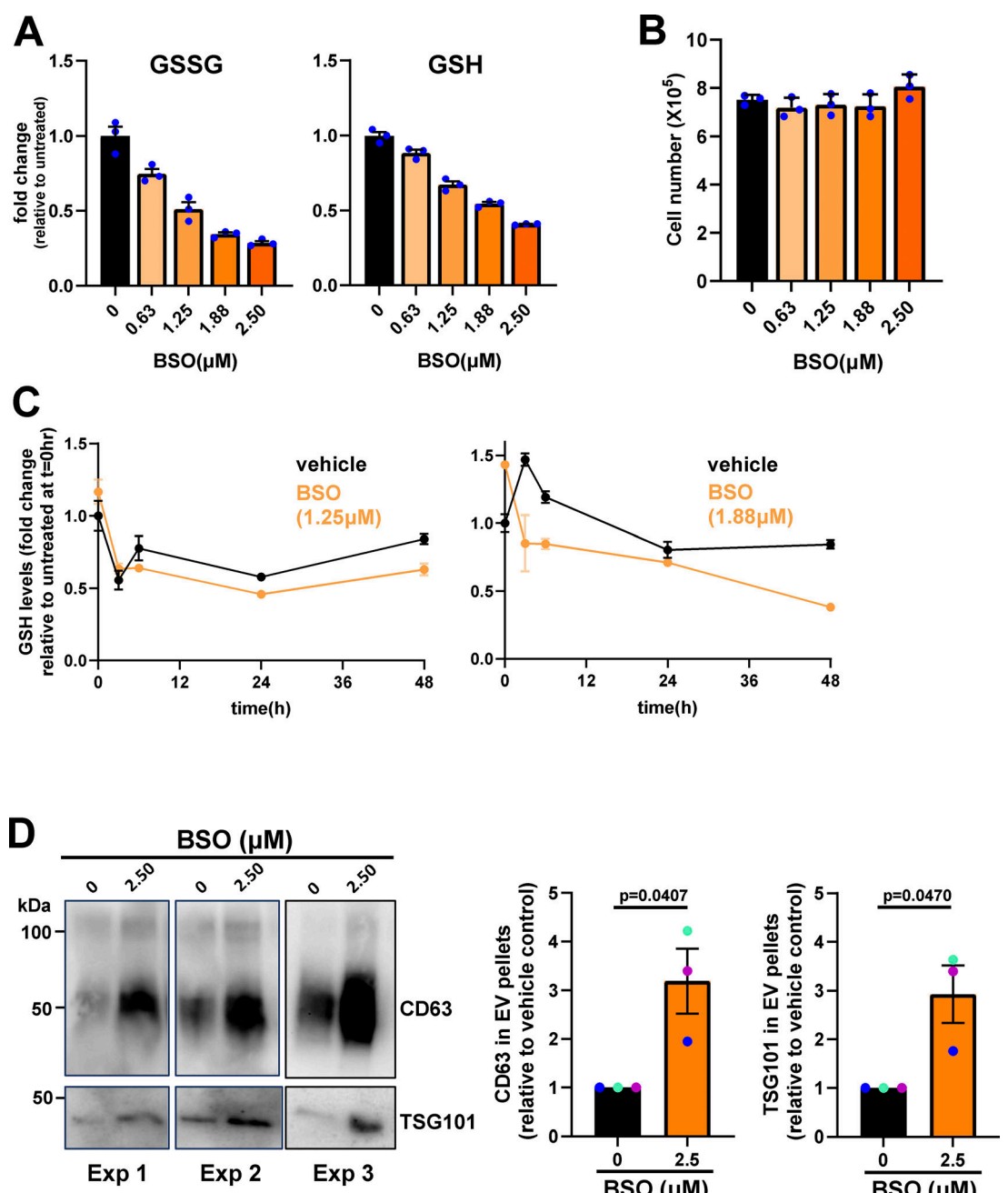

Figure S2. **Treatment of parental (P) cells with BSO. (A–D)** Parental (P) cells were incubated with the indicated concentrations of BSO for 24 h (A and B), 48 h (D), or for the indicated times (C). Following this, cell-conditioned medium was collected for isolation of sEVs by differential centrifugation (D), and the cells lysed for determination of intracellular metabolites by LC-MS (A and C). Levels of oxidized (GSSG) and reduced (GSH) glutathione are expressed as fold change relative to untreated cells (A and C), and the total number of cells was determined (B). Values in A–C are mean ± SEM, three technical replicates per condition. The CD63 and TSG101 content of EV pellets from untreated or BSO-treated (2.5 μM) P cells was determined by western blotting. Each western blot represents an individual experiment. CD63 levels in EV pellets were quantified as for Fig. 5 B, and values represent mean ± SEM, $n$ = 3 independent experiments (each colored dot represents an individual experiment); data were analyzed by paired $t$ test. Source data are available for this figure: SourceData FS2.

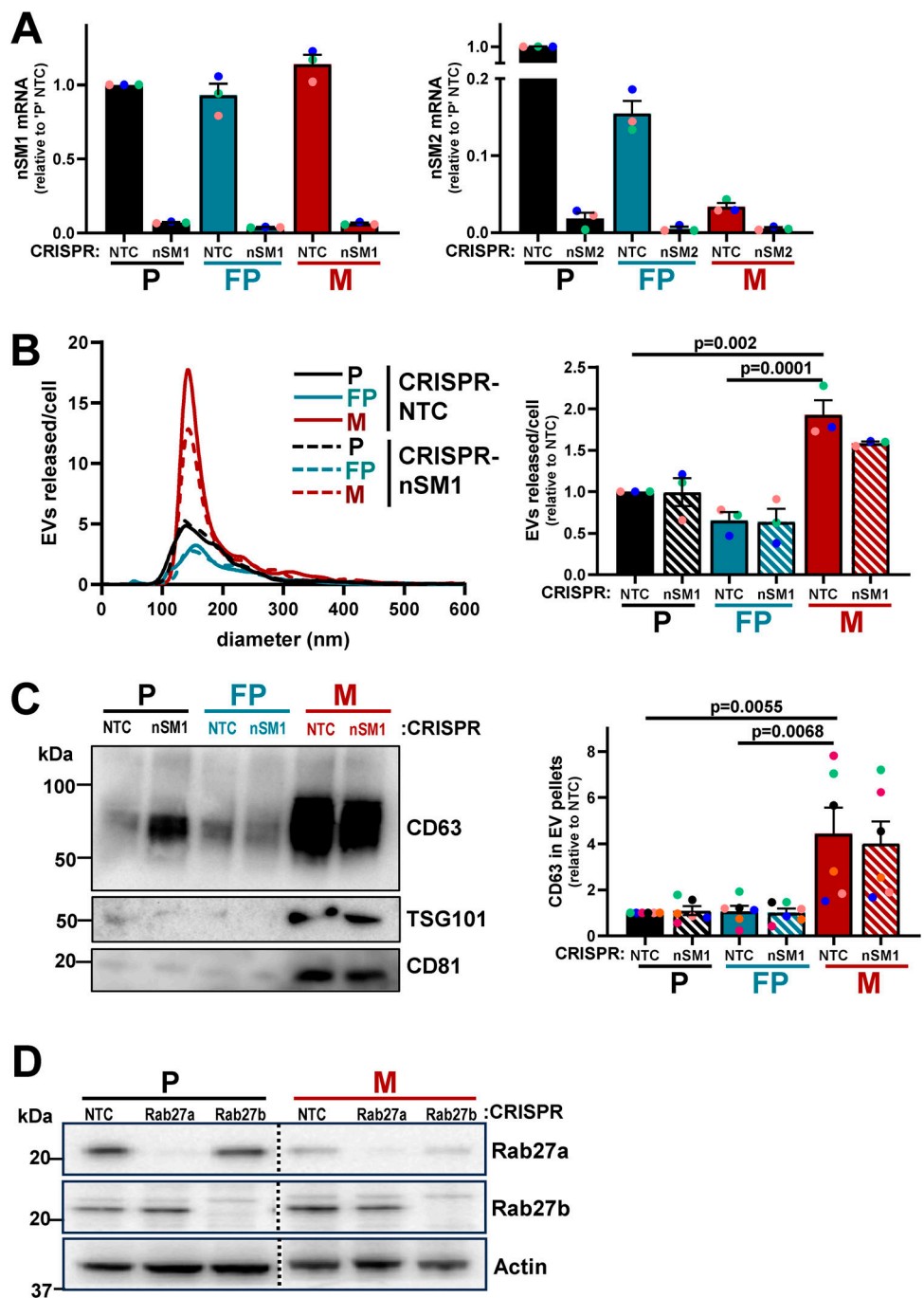

Figure S3. **Validation of CRISPR disruption of nSMases and Rab27s and EV release from nSMase-1 CRISPR cells. (A)** Parental (P), FP, and micrometastatic (M) cells in which nSMase1 (nSM1) or nSMase2 (nSM2) were disrupted by CRISPR were lysed. Levels of mRNA-encoding nSM1 (left graph) or nSM2 (right graph) were determined using qPCR. All data were normalized to ARPP P0 and presented relative to expression in parental NTC cells; values are mean ± SEM, n = 3, colored dots are independent experiments. **(B and C)** Parental (P), FP, or micrometastatic (M) cells were transduced with a lentiviral vector bearing gRNAs recognizing non-targeting sequences (CRISPR-NTC) or sequences targeting nSMase1 (CRISPR-nSM1). EVs were purified from conditioned medium collected over a 48-h period and analyzed using nanoparticle tracking (B) as for Fig. 5 A, and levels of CD63 in EV pellets were determined by western blotting (C) as for Fig. 5 B. Values are mean ± SEM, one-way ANOVA, with Tukey's multiple comparison, n = 3 (B), n = 6 (C), colored dots are independent experiments. **(D)** Western blotting was used to determine levels of Rab27a and Rab27b in parental (P) and micrometastatic (M) cells in which the genes for the Rab GTPases were disrupted using CRISPR. Actin is used as loading control. Source data are available for this figure: SourceData FS3.

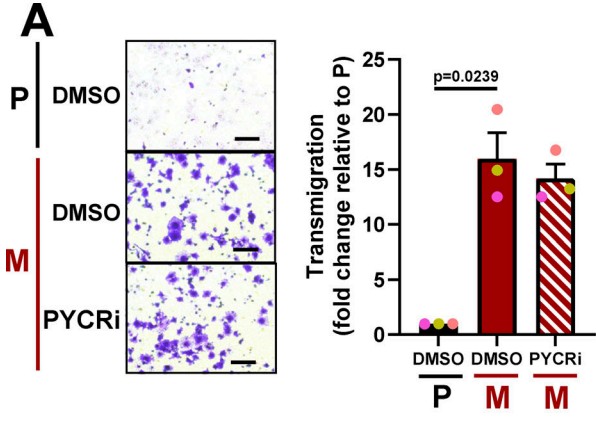

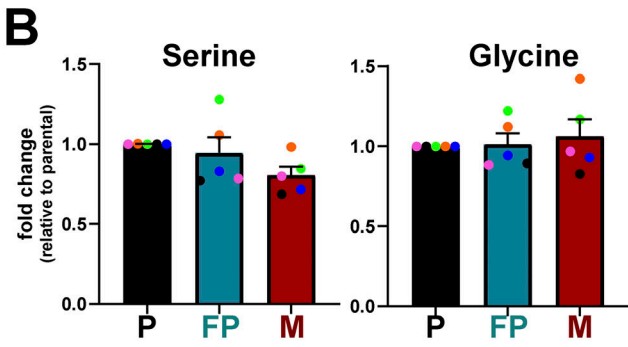

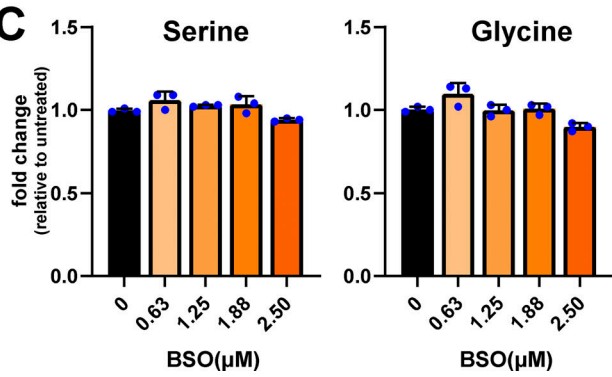

**Figure S4.  PYCR activity is not required for transmigration of MMTV-PyMT mammary cancer cells and serine/glycine levels in mammary cancer cells from primary tumor and micrometastases and following treatment with BSO. (A)** Parental (P) or micrometastatic (M) cells were treated with PYCR inhibitor (PYCRi; 20 µM) or control vehicle as indicated. Cells were then seeded into the upper chambers of Transwells (8-µm pore size) and allowed to transmigrate over a 2-h period toward a gradient of serum and fibronectin (applied to the lower chamber) in the presence or absence of 20 µM PYCRi or vehicle control (DMSO). Representative images (left panels) and quantification (right panels) of the number of cells adherent to the upper surface of the lower chamber are displayed. Bars, 100 µm. Values are mean ± SEM, n = 3 experiments (each colored dot represents an individual experiment); data were analyzed using a paired t test. **(B)** Parental (P), FP or micrometastatic (M) cells were cultured as for Fig. 2 A, and levels of intracellular metabolites were determined using LC-MS–based metabolomics. The abundance of intracellular serine and glycine are expressed as mean fold change ± SEM relative to parental cells, n = 5 experiments (each colored dot represents an individual experiment). **(C)** Parental (P) cells were incubated with the indicated concentrations of BSO for 24 h and then cells were lysed for determination of intracellular metabolites by LC-MS. Levels of glycine and serine are expressed as fold change relative to untreated cells. Values are mean ± SEM, three technical replicates per condition.

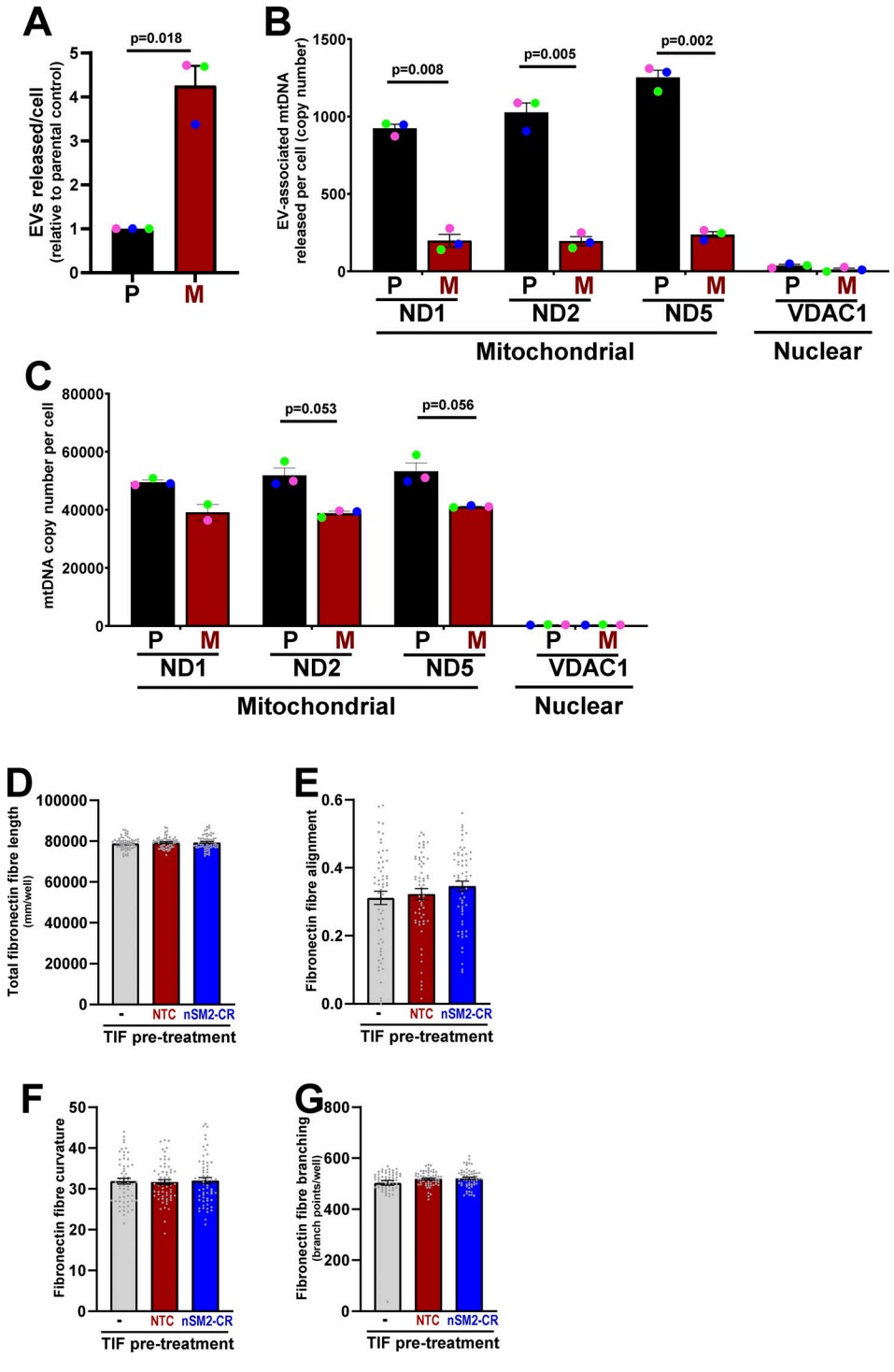

Figure S5. **Micrometastatic (M) cells release less EV-associated mtDNA than cells from primary tumors (P), and EVs from micrometastatic cells do not influence the fibronectin filament length or organization in the ECM deposited by fibroblasts. (A–C)** EVs were purified from the conditioned media of parental (P) and micrometastatic (M) cells as for Fig. 5 A and B. The number of EVs was determined by nanoparticle tracking (A). Purified EVs were incubated in the presence of DNase immobilized to agarose beads to remove DNA associated with the external surface of the EVs. DNase-treated EVs were then lysed, and mtDNA and nuclear DNA was determined by digital-drop PCR using primers complementary to sequences within the mitochondrial ND1, ND2, and ND5 genes or the nuclear-encoded VDAC1 gene (B). The cellular content of the ND1, ND2, ND5, and VDAC1 genes were confirmed using digital-drop PCR (C). Values are mean ± SEM, n = 3 independent experiments (each colored dot represents an individual experiment); data were analyzed using a paired t test. **(D–G)** TIFs were incubated with EVs from control (NTC) or nSMase-2 CRISPR (nSM2-CR) M cells for 72 h or were left untreated (−). EV pre-treated or untreated TIFs were then replated and allowed to generate ECM for 7 days, which was then de-cellularized. Fibronectin was then visualized in the de-cellularized ECM using immunofluorescence followed by confocal microscopy. Quantitative image analysis (TWOMBLI) (Wershof et al., 2021) was then used to determine the characteristics of the deposited fibronectin fibers, including the fiber length (D), alignment (E), curvature (F), and branching (G). Values are mean ± SEM, 60 fields of view were quantified per condition (each grey dot corresponds to a field of view).

