## [Peer Review File · The Journal of Cell Biology]

Metabolic adaptations of micrometastases alter EV production to generate invasive microenvironments

Michalis Gounis, America Campos, Engy Shokry, Louise Mitchell, Ruhi Deshmukh, Emmanuel Dornier, Nicholas Rooney, Sandeep Dhayade, Luis Pardo-Fernandez, Madeleine Moore, David Novo, Jenna Mowat, Craig Jamieson, Emily Kay, Sara Zanivan, Colin Nixon, Iain Macpherson, Saverio Tardito, David Sumpton, Karen Blyth, Jim Norman, Cassie Clarke, Nikki Paul, and Claire Mitchell

Corresponding Author(s): Jim Norman, CRUK-Beatson Institute and Cassie Clarke, Cancer Research UK Beatson Institute

Review Timeline:

Submission Date:	2024-05-13
Editorial Decision:	2024-06-25
Revision Received:	2025-01-29
Editorial Decision:	2025-03-14
Revision Received:	2025-05-01

Monitoring Editor: Brendan Manning

Scientific Editor: Tim Fessenden

Transaction Report:

DOI: <https://doi.org/10.1083/jcb.202405061>

June 25, 2024

Re: JCB manuscript #202405061

Prof. Jim C Norman
CRUK-Beatson Institute
Integrin Cell Biology Lab Beatson Institute for Cancer Research Garscube Estate, Switchback Road
Glasgow, East Dumbartonshire G61 1BD
United Kingdom

Dear Prof. Norman,

Thank you for submitting your manuscript entitled "Metabolic adaptations of micrometastases alter EV production to generate invasive microenvironments". Your manuscript has been assessed by expert reviewers, whose comments are appended below. Although the reviewers express potential interest in this work, significant concerns unfortunately preclude publication of the current version of the manuscript in JCB.

You will see that all reviewers appreciated the conceptual advance connecting proline metabolism with the spread of breast cancer cells to the lung microenvironment. However, reviewers varied in their concerns over validations and controls, and sought greater details on the proposed mechanism related to metabolism itself as well as its connection with small extracellular vesicles. A suitably revised manuscript must strengthen this connection, and provide essential validation of the metabolic adaptations proposed here to promote metastasis. Accordingly all points by Reviewers 2 and 3 must be resolved, as well as points 2 and 3 by Reviewer 1, whereas their first point, which we agree makes an intriguing suggestion, is left to your discretion. Finally, all reviewers remarked on the appearance of claims followed by "data not shown," which are not permitted by JCB.

Please let us know if you are able to address the major issues outlined above and wish to submit a revised manuscript to JCB. Note that a substantial amount of additional experimental data likely would be needed to satisfactorily address the concerns of the reviewers. The typical timeframe for revisions is three to four months. While most universities and institutes have reopened labs and allowed researchers to begin working at nearly pre-pandemic levels, we at JCB realize that the lingering effects of the COVID-19 pandemic may still be impacting some aspects of your work, including the acquisition of equipment and reagents. Therefore, if you anticipate any difficulties in meeting this aforementioned revision time limit, please contact us and we can work with you to find an appropriate time frame for resubmission. Please note that papers are generally considered through only one revision cycle, so any revised manuscript will likely be either accepted or rejected.

If you choose to revise and resubmit your manuscript, please also attend to the following editorial points. Please direct any editorial questions to the journal office.

GENERAL GUIDELINES:

Text limits: Character count is < 40,000, not including spaces. Count includes title page, abstract, introduction, results, discussion, and acknowledgments. Count does not include materials and methods, figure legends, references, tables, or supplemental legends.

Figures: Your manuscript may have up to 10 main text figures. To avoid delays in production, figures must be prepared according to the policies outlined in our Instructions to Authors, under Data Presentation, <https://jcb.rupress.org/site/misc/ifora.xhtml>. All figures in accepted manuscripts will be screened prior to publication.

Supplemental information: There are strict limits on the allowable amount of supplemental data. Your manuscript may have up to 5 supplemental figures. Up to 10 supplemental videos or flash animations are allowed. A summary of all supplemental material should appear at the end of the Materials and methods section.

Please note that JCB now requires authors to submit Source Data used to generate figures containing gels and Western blots with all revised manuscripts. This Source Data consists of fully uncropped and unprocessed images for each gel/blot displayed in the main and supplemental figures. Since your paper includes cropped gel and/or blot images, please be sure to provide one Source Data file for each figure that contains gels and/or blots along with your revised manuscript files. File names for Source Data figures should be alphanumeric without any spaces or special characters (i.e., SourceDataF#, where F# refers to the associated main figure number or SourceDataFS# for those associated with Supplementary figures). The lanes of the gels/blots should be labeled as they are in the associated figure, the place where cropping was applied should be marked (with a box),

and molecular weight/size standards should be labeled wherever possible.

If you choose to resubmit, please include a cover letter addressing the reviewers' comments point by point. Please also highlight all changes in the text of the manuscript.

Regardless of how you choose to proceed, we hope that the comments below will prove constructive as your work progresses. We would be happy to discuss them further once you've had a chance to consider the points raised. You can contact the journal office with any questions at cellbio@rockefeller.edu.

Thank you for thinking of JCB as an appropriate place to publish your work.

Sincerely,

Brendan Manning
Monitoring Editor
Journal of Cell Biology

Tim Fessenden
Scientific Editor
Journal of Cell Biology

Reviewer #1 (Comments to the Authors (Required)):

In the manuscript by Gounis M. et al., titled "Metabolic adaptations of micrometastases alter EV production to generate invasive microenvironments," the authors interrogated the drivers of early metastasis (i.e., micrometastasis). Specifically, they studied metabolic alterations associated with micrometastasis and how these metabolic alterations supported the migratory capabilities of tumor cells. They developed a new model of micrometastasis by isolating breast cancer cells that have disseminated to the lungs of MMTV-PyMT mice. They confirmed these cells have increased invasive capacity compared to parental breast cancer cells. They profiled metabolites and showed that proline levels are increased in the plasma of mice with metastasis and in the serum of breast cancer patients with metastasis. Glutamate, derived from glutamine, can contribute to the generation of either proline or glutathione (GSH). They showed that, along with lower proline levels, micrometastatic cells had lower levels of reduced GSH compared to parental cells. Further, they showed micrometastatic cells had a lower flux of glutamine/glutamate into GSH and a higher flux into proline. Additionally, they showed that these phenotypes are reversed upon treatment with an inhibitor of PYCR, the enzyme that synthesizes proline. Next, they showed that micrometastatic cells have increased release of small extracellular vesicles (sEV) compared to parental cells. Importantly, sEV can support tumor metastasis. Mechanistically, they showed that following inhibition of GSH synthesis, parental cells increased their release of sEVs, suggesting that the generation of GSH inhibited the release of sEVs. Ceramides are a class of lipids that contribute to the production of sEV, and they found that micrometastatic cells have an increased abundance of ceramides compared to parental cells. Further, they showed that the production of ceramides in micrometastatic cells is controlled by neutral sphingomyelinase-2 (nSM2). They showed that increased release of sEV in micrometastatic cells is controlled by nSM2 and that the increased release of sEV in parental cells following inhibition of GSH synthesis is controlled by nSM2. Finally, they showed that the activity of nSM2 and the production of sEV by micrometastatic cells govern their increased migratory capability.

This is an excellent manuscript. The authors outline the hypotheses, experiments, and conclusions in a clear and logical manner. Clear mechanistic insight is provided regarding metabolic regulators of breast cancer metastasis. Suggestions are provided to potentially improve the manuscript.

Major Concerns

1. While the connections between ceramide generation and migratory capabilities of breast cancer cells, less evidence is provided for the connection between proline generation and migratory phenotypes. It would be interesting to potentially explore whether the PYCR inhibitor blocks the pro-invasive phenotype of micrometastatic cells. Alternatively, it would be interesting to examine if overexpression of PYCR in parental cells can decrease GSH levels and promote migration.

2. An outstanding question is the mechanisms involved in the GSH-dependent control of ceramide generation. This would be interesting to explore, or at least discuss, in the manuscript. Serine is required for ceramide generation, and serine can be generated from glycine via SHMT. Since glycine is required for GSH synthesis, one hypothesis is that GSH synthesis lowers glycine, and subsequently serine, levels, thereby lowering the ability to generate ceramide.
3. Statistical analysis is not provided for Figure 4B, and it should be provided.

Minor Concerns

1. Page 9: when mentioning CD63- and TSG101-positive sEV for the first time, it would add clarity if the biological significance of these markers were explained.
2. Page 9: It would be helpful if the "(data not shown)" regarding GSH production inhibition by BSO were provided.
3. Additional information on how you eliminated fat pad stromal and parenchymal cells during the generation of FP cells and lung stromal and parenchymal cells during the generation of M cells would be helpful.

Reviewer #2 (Comments to the Authors (Required)):

Summary

In the manuscript "Metabolic adaptations of micrometastases alter EV production to generate invasive microenvironments" Gounis et al. used a mouse model of mammary carcinoma syngeneic transplantation and primary tumor resection (MMTV-PyMT), to generate isogenic cell lines from both primary tumors and corresponding lung micrometastases. Key findings using this model include: micrometastatic cells increase proline production, reduced glutathione synthesis and reduced total glutathione levels. Micrometastatic cells also exhibit altered sphingomyelin metabolism, resulting in higher intracellular levels of specific ceramides. These metabolic changes affect the production of small extracellular vesicles (sEVs). Micrometastatic cells switch from Rab27-dependent sEV production to neutral sphingomyelinase-2 (nSM2)-dependent sEV release. The nSM2-dependent sEVs from micrometastatic cells impact fibroblasts' ability to deposit extracellular matrix, promoting cancer cell invasiveness. The study demonstrates that metabolic rewiring in micrometastatic cells influences sEV release, thus driving the formation of an invasive microenvironment during metastasis. This work contributes new fundamental understanding linking cellular metabolism to the invasive characteristics of cancer cells during metastasis via sEV/extracellular matrix deposition, and the impact of this work is appropriate for publication in JCB. However, a few key clarifications regarding the data interpretation as well as additional experiments are needed to ensure robustness of the findings prior to publication in JCB:

Major

1. In breast cancer, macrometastasis are quantitatively defined as > 2mm in the largest dimension and micrometastasis < 2mm (AJCC Cancer Staging). It is unclear how macro- versus micro- metastasizes are defined in this study and care should be taken to quantitatively define and separate these terms or to use an alternate term for the micrometastasis to avoid confusion with these naming conventions. Are the "larger/frank metastases" all > 2mm?
2. Is the reason why glutamate gets shunted toward proline is because cysteine is limiting? One experiment (among others) to address this would be to determine in if in cysteine deplete media conditions and/or system xCT inhibition with Erastin2 with the P lines, will this cause the P lines to shunt labeled glutamine toward proline? In cysteine deplete conditions, are there differences in cell survival/invasion in the P, FP and M cells?
3. The observed metabolic profile changes in the ex vivo P and M cultured cell lines could be due to transient changes in the microenvironment, as the authors suggest. However, given the authors are studying this mechanism ex vivo in cell lines recovered from these microenvironments it could be due to a more stable, non-transient selection. To clarify whether these changes are transient/reversible, the authors should inject the M lines back into the mammary fat pad and determine if the observed proline/glutathione changes revert to those seen in the primary tumors.
4. A possible interpretation of the data supporting the conclusion "thus micrometastatic cells may avoid excessive ROS generation which would constitute a risk in the highly oxidative environment of the lung" is that the cells are recovering from ROS exposure experienced in the blood during metastasis. The depleted glutathione in the M cells could be a consequence of selected metastasizing cells which survived the high ROS environment in the blood, those that were able to regenerate NAD(P)⁺ without activating oxidative phosphorylation. Experiments should be performed to aid in the interpretation of this stated conclusion and consideration of this possibility, such as measuring ROS levels in the breast cancer cells +/-PYCR inhibition and +/-BSO treatment in the P versus M lines or i.v. injection of the P versus M lines to ensure equivalent survival and formation of lung metastasis - thus, confirming the metabolic selection for this phenotype takes place during lung colonization and not in the blood.

5. Several key works in support of the authors claims should be added to the manuscript discussion at the appropriate points, and the authors' findings should be contextualized with regard to these prior findings: Roshanzamir et al., PNAS 2022; PMID: 35994654; Piskounova et al., Nature 2015 PMID: 26466563; Le Gal et al., Sci Transl Med 2015 PMID: 26446958; Schafer ZT et al. Nature 2009 PMID: 19693011; Metallo CM, et al. Nature 2012 PMID: 22101433

Minor

1 The introduction should carefully distinguish between metabolic adaptations that are thought to take place in the blood compared to that at distant organ environments - these are believed to be two different types of different metabolic adaptations the cancer cells undergo, with the blood being a significant bottleneck for metastasis (as is mentioned in the cited Luzzi et al., 1998). This blood bottleneck prior to the metastatic colonization at different microenvironments should also be more extensively considered in conclusions made from the data throughout the manuscript text.

2 The images included in Figure 1c are difficult to visualize; resolution and/or magnification needs to be enhanced

3 Please show a representative image of how lung metastasis burden was determined by retrospective histological examination (currently referenced as 'not shown'), as well as the data on GSH at 48 hours and showing that BSO does not compromise the growth/viability (currently referenced as 'not shown' before Fig. 5C)

4 For the patient study that was used in Fig 2c to compare proline levels, and in Fig 6D to compare ceramide species) in patients with and without metastatic breast cancer, please describe where this population came from/what database in the manuscript text as well as the methods. It is unclear where this data came from.

5 Proline, ceramides, and sphingolipids have already been shown to be increased in patients with breast cancer in epidemiological studies of plasma metabolomics of human samples (Brantley et al., Cancer Epidemiol Biomarkers Prev. 2022 PMID: PMC8983458), please site this work and include comparison the analyses with the findings in Fig. 2c.

6 The ratios of GSH to GSSG should be shown in Fig 3b to allow for interpretations about the oxidative status of the P, FP, and M cells to be made

7 SE and SD are both used at various points in the manuscript; clarify in the methods when each were selected to be used for data representation

8 The authors should speculate in the discussion if they expect their findings to be generalizable to other metastatic sites of colonization, or specific to the lung microenvironment niche.

Reviewer #3 (Comments to the Authors (Required)):

The goal of this manuscript was to identify metabolic differences between metastatic and non-metastatic cells. The authors generated isogenic cell models of parental (P and P'), primary tumor-derived (FP and FP'), and micrometastatic breast cancer cells (M and M') from the MMTV-PyMT model and analyzed metabolite changes in conditioned media. They found that proline is uniquely secreted from micrometastatic cells and that serum proline is elevated in sera from mice and humans with breast cancer metastasis. Analysis of the cells themselves demonstrated that proline levels were elevated and that GSH and GSSG levels were decreased in micrometastatic cell models (M and M's). Glutathione and proline both use glutamate as a common precursor and metabolic tracking was used to show that glutamate derived from glutamine is shunted towards proline synthesis in metastatic cells and away from glutathione and TCA cycle pathways. About halfway through the paper, the authors shift their focus to characterizing extracellular vesicle (EV) secretion by the metastatic cells, connecting it to the decrease in glutathione pathway metabolites. Lipidomics analysis of EVs secreted from the cells revealed an increase in lipids (ceramides, sphingomyelins) associated with the neutral sphingomyelinase 2 pathway, which is known to regulate EV biogenesis. KO of nSMase1 and 2 revealed that nSMase2 but not nSMase1 is important for SEV release from micrometastatic but not parental cells, decreasing the release by about half. KO of the exosome secretion regulators Rab27a or Rab27b inhibited exosome secretion from parental but not the micrometastatic cells, potentially suggesting different regulatory mechanisms. The effect of EVs purified from control and nSMase2-KO M cells on migration through collagen plugs and induction of fibroblasts to synthesize and deposit ECM is investigated. Based on these data, the authors propose a model in which the decrease in glutathione metabolism in micrometastatic cells induces EV secretion, dependent on nSMase2. Those EVs promote invasion through collagen plugs containing fibroblasts, presumably through inducing fibroblasts to synthesize ECM. Overall, it is an interesting manuscript with a lot of interesting data. The metabolic findings about the proline pathway /shunt from glutathione is interesting and backed up with human correlative data. Some of the EV data is interesting too, particularly the upregulation of ceramides in EVs and the role of the nSMase pathway. It does feel like two stories are cobbled together (metabolism and EV) with little link between the two. Also, there are a lot of rough edges, in terms of missing information and some technical issues (missing replicates, incomplete statistical comparisons, some potential issues with the way the EV studies seem to have been carried out). The points below address those issues.

1. How were cancer cells sorted from fat pad tumors and micrometastases? Was collagenase used? How were the cells separated from other cell types?

2. What does WT mean in Fig 2B? Is that the non-tumor-bearing mice? In Fig 2C, are the metastatic cancer patients lung metastatic (same plasma samples in Fig 6D)? What about the level of plasma proline in non-metastatic breast cancer patients? Is the reason that you didn't test this, that you can't rule out the presence of micrometastases? If so, it would probably help the reader to say so.

3. I think there is an error in the schematic diagram in Figure 4C. Doesn't shunting to alpha-KG go down, not up, in the

micrometastatic cells?

4. It is hard to agree that the secretion of CD63-positive small EVs is increased from M and M' compared to parental (P and P') and primary tumor cells (FP and FP') in Fig 5B and S3B, as well as by BSO in Fig 5E, if the loading was normalized to cell numbers. Other cargoes were increased from M and M', too. The data seem to just be increased small EV secretion. The loading should be based on equal number of small EVs or equal protein amount and then the specific effect on cargo proteins can be properly compared. Similarly, for Fig 5E, nanoparticle tracking analysis (NTA) assessment of actual EV number should be used to characterize the effect of the redox agent BSO on EV secretion from parental cells.
5. The only real link between the metabolism story and the EV story is the inhibitor experiment in Fig 5E (with very rough Western blot) in which an inhibitor upregulates EV secretion in parental cells (as assessed by Western blot, not NTA).
6. The statement in the results, "Deletion of nSM2 did not influence transmigration toward a gradient of serum and fibronectin (not shown), indicating that cell-autonomous migratory behavior depended on neither nSM2 nor EV release from micrometastatic cells." The last half of that sentence does not take into account the likelihood that serum and/or fibronectin may rescue effects of EV loss and so is an overinterpretation of the (not shown) data. Also, data should be shown if it will be mentioned in the manuscript.
7. For Fig 8, were the TIFs washed after the addition of the EVs or was the EV-TIF mixture added to the collagen plug? If the latter, how do you separate the effect of the EVs vs. the effects of the TIFs?
8. What media was used is when the TIFs were pre-treated with small EVs? Also, how were the EV treatments done of the TIFs? Equal vesicle number? Equal protein? If it was done based on starting cell number, it's a bit less informative since the KO cells secrete fewer EVs.
9. The method for preparing de-cellularized ECM is missing. In addition, quantitation of the amount of de-cellularized ECM deposited by fibroblasts under the different small EV pre-treatment conditions (as well as in the absence of EVs) would provide support for the authors' conclusion that the micrometastatic cancer cell EVs are inducing the fibroblasts to deposit more ECM.
10. The schematic summary in Fig 8 should be edited. Currently, it looks like nSMase2 controls exosome secretion by promoting the fusion event of MVB with the plasma membrane. nSMase2 is involved in MVB biogenesis, not the release step, as stated correctly in the Introduction.
11. The discussion is rather long and rambling, and the connections between the metabolic part of the paper and the EVs is not well made. Some references also seem to be missing, e.g. the authors mention that nSMase2-dependent EV production is not associated with release of mtDNA but without references. Torralba et al. has described the role of Rab27a and nSMase2 in mitochondrial function (doi:10.1038/s41467-018-05077-9) - perhaps this is relevant?
12. Small EVs are not another name for exosomes but a term encompassing exosomes and other small-sized EVs. The sentences at the end of the 4th paragraph of Introduction are ambiguous.
13. Please show all data if they appear in the manuscript or do not mention results if they are listed as "not shown".
14. At least 3 biological replicates for the assay in Fig 1E should be performed.
15. In Fig 3C and 4B, what is a technical repeat? Why did the authors perform 3 technical repeats instead of biological repeats? Please describe what the limitations are in performing biological repeats.
16. Graphs in Fig S2 show only 3 data points unlike the replicates stated in the figure legend, n=5.
17. How was the sEV amount for lipidomics normalized in Fig 6C?
18. Were the CRISPR cells validated with diagnostic PCR of genomic DNA to distinguish parental from heterozygous from homozygous? In some cases it can be challenging to distinguish between heterozygous and homozygous KO, e.g. for the light Western blot shown for Rab27a in the M cells in Fig S4. If it's a heterozygous rather than homozygous KO in the M cells, that could account for the unique lack of effect of Rab27a in the M cells. And it looks like there is a whisper of Rab27a in the KO lane of that very faint blot. Since Rab27a and nSMase2 affect distinct parts of the exosome biogenesis (nSMase2) and secretion (Rab27a) pathway, one wouldn't really expect switching from one mechanism to another anyway.
19. In Fig 6, how was the ceramide intensity normalized, per cell, per area, per field of view, or something else? It is unclear what "n" in Fig 6A and 6E indicates. If the "n" doesn't indicate independent experiments, "n=5" sounds too small for the quantitation. Typically one would want many images each from 3 biological replicates.
20. Presenting all statistics available between each group will be helpful for readers to understand the results. For instance, is the difference in ceramide intensity between NTC and nSM1 or between nSM1 and nSM2 in Fig 6E significant? Also, in Fig S5A, is EV release by nSM1 KO in M cells significantly decreased compared to NTC control?

Response to reviewers

Gounis et al.

Metabolic adaptations of micrometastases alter EV production to generate invasive microenvironments

Reviewer 1

We would like to thank this reviewer for their positive view of the manuscript and for their constructive criticisms/concerns which we have addressed with the inclusion of new data.

Major Concerns

1. While the connections between ceramide generation and migratory capabilities of breast cancer cells, less evidence is provided for the connection between proline generation and migratory phenotypes. It would be interesting to potentially explore whether the PYCR inhibitor blocks the pro-invasive phenotype of micrometastatic cells. Alternatively, it would be interesting to examine if overexpression of PYCR in parental cells can decrease GSH levels and promote migration.

We have addressed this as follows:

We have determined the consequences of inhibiting PYCR on transmigration of micrometastatic cells (which exhibit elevated proline synthesis from glutamate). These data are now presented in a new supplementary figure S8 and indicate that inhibition of proline synthesis, via Pycr, does not influence transmigration of micrometastatic cells. This point is also now addressed in the discussion.

2. An outstanding question is the mechanisms involved in the GSH-dependent control of ceramide generation. This would be interesting to explore, or at least discuss, in the manuscript. Serine is required for ceramide generation, and serine can be generated from glycine via SHMT. Since glycine is required for GSH synthesis, one hypothesis is that GSH synthesis lowers glycine, and subsequently serine, levels, thereby lowering the ability to generate ceramide.

We have addressed this as follows:

We have determined the levels of serine and glycine in primary tumour (P & FP) and micrometastatic (M) cells; the latter having reduced levels of glutathione (Fig. 3A-C). We have also tested whether incubation of primary tumour cells with various concentrations of BSO to reduce glutathione (Fig. S5 A,C) may alter serine and glycine content. These data are now presented in a new supplementary figure S9 and indicate that neither serine nor glycine levels were altered under either of these circumstances. We have outlined this possibility in the discussion (page 15) and the call-out for figure S9 is at this point.

3. Statistical analysis is not provided for Figure 4B, and it should be provided.

We have addressed this as follows:

We have repeated this experiment and now present 3 independent experiments determining the consequences of inhibition of PYCR on the routing of glutamine-derived carbons to various metabolites, including glutathione. This has enabled proper statistical analysis of this experiment with each individual experiment being represented as a coloured dot. These data are presented in a new figure 4B.

Minor Concerns

1. Page 9: when mentioning CD63- and TSG101-positive sEV for the first time, it would add clarity if the biological significance of these markers were explained.

We have addressed this as follows:

We have now amended the results text on page 9-10 to introduce these EV markers.

2. Page 9: It would be helpful if the "(data not shown)" regarding GSH production inhibition by BSO were provided.

We have addressed this as follows:

We have now added the data describing the ability of BSO (in a low μM concentration range) to suppress cellular glutathione levels in a temporally stable manner, and to do so without compromising cell growth/viability. These data are now included in a new supplementary figure S5A-C.

3. Additional information on how you eliminated fat pad stromal and parenchymal cells during the generation of FP cells and lung stromal and parenchymal cells during the generation of M cells would be helpful.

We have addressed this as follows:

We find that cells other than those expressing the MMTV-PyMT transgene do not survive for more than a few generations in culture. Therefore, we find it not necessary to incorporate steps to actively eliminate these. We have included the following statements in the appropriate methods section to cover this point (new text is highlighted in red):

[To establish mammary tumour cell lines, cells were **trypsinised and re-plated** (once they reached around 90% confluence) for at least five **passages (representing >25 cell divisions)** and maintained in culture for a maximum of twenty passages. The same protocol was used for the generation of metastatic cell lines, with the exception that following centrifugation, the resuspended cell pellet was seeded in 6 well plates. **Following this period of ex-vivo culture (>25 cell divisions) we found that all cells expressed the MMTV-PyMT transgene and were E-cadherin positive, indicating that they were tumour cells of epithelial origin that had outgrown cells such as lung parenchymal and stromal cells.]**

Reviewer 2

We would like to thank this reviewer for their conclusion that our manuscript *'contributes new fundamental understanding linking cellular metabolism to the invasive characteristics of cancer cells during metastasis via sEV/extracellular matrix deposition, and the impact of this work is appropriate for publication in JCB'*.

We also thank this reviewer for their constructive criticisms/concerns which we have addressed, mostly with the inclusion of new data, and which are described below.

Major

1. In breast cancer, macrometastasis are quantitatively defined as > 2mm in the largest dimension and micrometastasis < 2mm (AJCC Cancer Staging). It is unclear how macro- versus micro- metastasizes are defined in this study and care should be taken to quantitatively define and separate these terms or to use an alternate term for the micrometastasis to avoid confusion with these naming conventions. Are the "larger/frank metastases" all > 2mm?

We have addressed this as follows:

We have conducted a quantitative analysis to define the distinction between lung micro – and macrometastases in the MMTV-PyMT transplantation/resection model that we have deployed. These data are included in a new supplementary figure S1A,B. They indicate that most metastatic nodules in the lung at clinical end-point (approximately 1-2 months following primary tumour resection) are micrometastases, and that these have a mean diameter of approximately 150 µm. If macrometastases are present, there are normally only 1 or 2 of them, and they have a mean diameter of approximately 2 mm. We have added a detailed description of this characterisation to the results section on page 6.

2. Is the reason why glutamate gets shunted toward proline is because cysteine is limiting? One experiment (among others) to address this would be to determine in if in cysteine deplete media conditions and/or system xCT inhibition with Erastin2 with the P lines, will this cause the P lines to shunt labeled glutamine toward proline? In cysteine deplete conditions, are there differences in cell survival/invasion in the P, FP and M cells?

We have addressed this as follows:

This is an excellent suggestion, and we have now performed metabolic tracing experiments to determine the fate of glutamine-derived carbons in cells from primary tumours (P cells) when cultured in reduced concentrations of cystine. As the reviewer hypothesises, these experiments indicate that glutamine-derived carbons are, indeed, shunted away from glutathione synthesis and toward proline when cystine/cysteine is limiting. These data are now presented in a new figure 4C.

3. The observed metabolic profile changes in the ex vivo P and M cultured cell lines could be due to transient changes in the microenvironment, as the authors suggest. However, given the authors are studying this mechanism ex vivo in cell lines recovered from these microenvironments it could be due to a more stable, non-transient selection. To clarify whether these changes are transient/reversible, the authors should inject the M lines back into the mammary fat pad and determine if the observed proline/glutathione changes revert to those seen in the primary tumors.

We have addressed this as follows:

As the reviewer suggests, we have now performed an experiment in which we have (re)-injected P and M cells into the mammary fat pad and compared their capacity to grow. This indicates that the cells from micrometastases (M) have markedly reduced capacity to grow in the mammary fat pad. These data are now presented in a new

supplementary figure S1C. We feel that these data support the reviewer's view that M cells have undergone changes that are 'stable' (possibly due to selection) and that are not easily reversible. We have, therefore, also described these results as follows in the results section on page 6-7: [.....to assess the stability of the differences between the micrometastatic and primary tumour cells, we re-introduced M cells into the 4th mammary fat pad of FVB mice and monitored their growth. Whilst all (5/5) of the primary tumour cells (P) grew efficiently to form lesions of 300 – 1000 mm², micrometastatic cells were unable to establish tumours in the mammary fat pad within 10 weeks (Fig. S1C). This indicated that, by moving to the lung to establish micrometastases, 'M' cells have undergone a relatively stable non-transient selection and/or adaptation which renders them incompetent to grow in the microenvironment of the mammary fat pad].

4. A possible interpretation of the data supporting the conclusion "thus micrometastatic cells may avoid excessive ROS generation which would constitute a risk in the highly oxidative environment of the lung" is that the cells are recovering from ROS exposure experienced in the blood during metastasis. The depleted glutathione in the M cells could be a consequence of selected metastasizing cells which survived the high ROS environment in the blood, those that were able to regenerate NAD(P)+ without activating oxidative phosphorylation. Experiments should be performed to aid in the interpretation of this stated conclusion and consideration of this possibility, such as measuring ROS levels in the breast cancer cells +/-PYCR inhibition and +/-BSO treatment in the P versus M lines or i.v. injection of the P versus M lines to ensure equivalent survival and formation of lung metastasis - thus, confirming the metabolic selection for this phenotype takes place during lung colonization and not in the blood.

We have addressed this as follows:

We have performed several experiments aimed at measuring ROS generation by P and M cells. These experiments, so far, indicate that neither of these cell populations generate much ROS and, moreover, that this is not altered by addition of the Pycr inhibitor. Consistently, preliminary observations indicate that the oxygen consumption rate does not differ between P and M cells. Thus, we are not anticipating including data on ROS production in the current manuscript. Additionally, we agree with this reviewer that our data do not conclude that micrometastatic cells may be wired to avoid excessive ROS generation '.....in the highly oxidative environment of the lung'. Firstly, because it is, indeed, possible, that the oxidative environment of the circulation may also contribute to such a phenomenon. And, secondly, because we feel that our wording offers this as a possible mechanism for the drive to metabolic re-wiring of the micrometastatic cells and not a firm conclusion. We have modified the appropriate sentence in the discussion thus: [Thus, micrometastatic cells may avoid excessive ROS generation to maximise their chances of survival in the highly oxidative environment of the lung and/or the circulation.] to moderate the tone of our suggestion.

5. Several key works in support of the authors claims should be added to the manuscript discussion at the appropriate points, and the authors' findings should be contextualized with regard to these prior findings: Roshanzamir et al., PNAS 2022; PMID: 35994654; Piskounova et al., Nature 2015 PMID: 26466563; Le Gal et al., Sci Transl Med 2015 PMID: 26446958; Schafer ZT et al. Nature 2009 PMID: 19693011; Metallo CM, et al. Nature 2012 PMID: 22101433

We have addressed this as follows:

We have now added Roshanmir et al., and Pikounova et al. to the discussion and modified the text as appropriate to contextualise them.

Minor

1 The introduction should carefully distinguish between metabolic adaptations that are thought to take place in the blood compared to that at distant organ environments - these are believed to be two different types of different metabolic adaptations the cancer cells undergo, with the blood being a significant bottleneck for metastasis (as is mentioned in the cited Luzzi et al., 1998). This blood bottleneck prior to the metastatic colonization at different microenvironments should also be more extensively considered in conclusions made from the data throughout the manuscript text.

We have addressed this as follows:

We have modified the text of the introduction and discussion at certain points to draw attention to this distinction.

2 The images included in Figure 1c are difficult to visualize; resolution and/or magnification needs to be enhanced

We have addressed this as follows:

We believe that this problem was in the PDF conversion. We have increased the resolution of the PDF to try to reduce this problem. However, proper resolution of the images may need to await submission of TIF files to the journal.

3 Please show a representative image of how lung metastasis burden was determined by retrospective histological examination (currently referenced as 'not shown'), as well as the data on GSH at 48 hours and showing that BSO does not compromise the growth/viability (currently referenced as 'not shown' before Fig. 5C)

We have addressed this as follows:

We have now included a description of the determination of metastatic burden in the methods section on page 17. We have also added the data describing the ability of BSO (in a low μM concentration range) to suppress cellular glutathione levels in a temporally stable manner, and to do so, without compromising cell growth/viability. These data are now included in a new supplementary figure S5A-C.

4 For the patient study that was used in Fig 2c to compare proline levels, and in Fig 6D to compare ceramide species) in patients with and without metastatic breast cancer, please describe where this population came from/what database in the manuscript text as well as the methods. It is unclear where this data came from.

We have addressed this as follows:

We have now included a section in the methods to outline the provenience of these samples. As follows: 'Plasma samples from patients with metastatic breast cancer and female healthy volunteers were originally collected under a protocol approved by West of Scotland REC4 (reference 10/S0704/32). Analysis of these archival samples for the current research was approved by Office for Research Ethics Committees Northern Ireland HSC REC B (reference 17/NI/0228).'

5 Proline, ceramides, and sphingolipids have already been shown to be increased in patients with breast cancer in epidemiological studies of plasma metabolomics of human samples (Brantley et al., Cancer Epidemiol Biomarkers Prev. 2022 PMID: PMC8983458), please cite this work and include comparison the analyses with the findings in Fig. 2c.

We have addressed this as follows:

We have now cited this Brantley et al. paper plus other studies which have identified associations between circulating proline levels and breast cancer, as follows: 'Consistently, elevated levels of plasma proline have previously been identified as being positively

associated with breast cancer in several studies (Brantley et al., 2022; Li et al., 2020; Miyagi et al., 2011).'

6 The ratios of GSH to GSSG should be shown in Fig 3b to allow for interpretations about the oxidative status of the P, FP, and M cells to be made

We have addressed this as follows:

We have now included these data in Fig. 3B and S3B&D.

7 SE and SD are both used at various points in the manuscript; clarify in the methods when each were selected to be used for data representation

We have addressed this as follows:

We have updated all figures so that all error bars are now SEM. In the original submission SD had been used when there were insufficient biological repeats. The incorporation of additional biological repeats into this revision has clearly firmed-up the study, and we now use SEM throughout.

8 The authors should speculate in the discussion if they expect their findings to be generalizable to other metastatic sites of colonization, or specific to the lung microenvironment niche.

We have addressed this as follows:

We have now covered this point in the concluding paragraph of the discussion.

Reviewer 3

We would like to thank this reviewer for their evaluation of our manuscript as ‘*an interesting manuscript with lots of interesting data*’ and for drawing attention to the instances of missing information, and some of the areas in which the presentation of the data seems to be incomplete. We have rectified all these shortcomings as detailed below:

Reviewer points

1. How were cancer cells sorted from fat pad tumors and micrometastases? Was collagenase used? How were the cells separated from other cell types?

We have addressed this as follows:

We have included the following statements in the appropriate methods section to cover this point (new text is highlighted in red):

[To establish mammary tumour cell lines, cells were **trypsinised and re-plated** (once they reached around 90% confluence) for at least five **passages (representing >25 cell divisions)** and maintained in culture for a maximum of twenty passages. The same protocol was used for the generation of metastatic cell lines, with the exception that following centrifugation, the resuspended cell pellet was seeded in 6 well plates. **Following this period of ex-vivo culture (>25 cell divisions) we found that all cells expressed the MMTV-PyMT transgene and were E-cadherin positive, indicating that they were tumour cells of epithelial origin that had outgrown cells such as lung parenchymal and stromal cells.]**

2. What does WT mean in Fig 2B? Is that the non-tumor-bearing mice? In Fig 2C, are the metastatic cancer patients lung metastatic (same plasma samples in Fig 6D)? What about the level of plasma proline in non-metastatic breast cancer patients? Is the reason that you didn't test this, that you can't rule out the presence of micrometastases? If so, it would probably help the reader to say so.

We have addressed this as follows:

Fig. 2B - Yes, WT meant non-tumour bearing mice. However, to clear-up this misunderstanding we have modified the figure to refer to these categories of mice as either non-tumour bearing ‘FVB’ or ‘MMTV-PyMT’ mice. Moreover, this is now outlined clearly in the figure legend.

Fig. 2C/Fig. 6D – The patient cohorts used for figures 2C and 6D are, indeed, the same. However, the samples differ because figure 2C presents data from determination of polar metabolites in serum, whereas figure 6D presents data from a lipidomic analysis of plasma.

All our patient samples are either from women with metastatic breast cancer or healthy age-matched volunteers as a control. We do not yet have access to plasma or serum from breast cancer patients who do not have metastases. Indeed, as this reviewer points-out, it is not normally possible to be sure of the absence of metastases in patient populations.

3. I think there is an error in the schematic diagram in Figure 4C. Doesn't shunting to alpha-KG go down, not up, in the micrometastatic cells?

We have addressed this as follows:

Indeed, we have now modified this figure accordingly.

4. It is hard to agree that the secretion of CD63-positive small EVs is increased from M and M' compared to parental (P and P') and primary tumor cells (FP and FP') in Fig 5B and S3B, as well as by BSO in Fig 5E, if the loading was normalized to cell numbers. Other cargoes were increased from M and M', too. The data seem to just be increased small EV secretion. The loading should be based on equal number of small EVs or equal protein amount and then the specific effect on cargo proteins can be properly

compared. Similarly, for Fig 5E, nanoparticle tracking analysis (NTA) assessment of actual EV number should be used to characterize the effect of the redox agent BSO on EV secretion from parental cells.

We have addressed this as follows:

i) We do not claim that changes in EV release are specific for CD63-positive EVs. Rather that there are alterations in EV release as measured by nanoparticle tracking and that these are reflected by similar changes in the CD63 content of EV pellets and in the levels of other EV markers. We feel that this misunderstanding may arise from the way that we have worded certain points. We have, therefore, re-worded statements to be more accurate in our descriptions. Furthermore, we have removed claims concerning release of small EVs and now refer more generally to EVs.

ii) All loading of EV pellets into Western blots are normalised to the number of 'donor' mammary cancer cells. Therefore, the CD63 Western blots reflect the CD63 content of EV pellets released by equivalent numbers of mammary cancer cells – be they P or M cells.

iii) As recommended by this reviewer, we have now included nanoparticle tracking analysis to support the Western blot demonstrating the effect of BSO on EV release. These data are now included in a new figure 5D.

5. The only real link between the metabolism story and the EV story is the inhibitor experiment in Fig 5E (with very rough Western blot) in which an inhibitor upregulates EV secretion in parental cells (as assessed by Western blot, not NTA).

We have addressed this as follows:

We have firmed-up the experiments testing the effect of BSO on EV release. We have now performed nanoparticle tracking analysis which demonstrates that BSO-treatment increases EV release by P cells. These data are now presented in Fig. 5D. We have also conducted another 3 repeats of the experiment shown in Fig. 5E but using a single concentration (2.5 μ M) of BSO. These 3 individual experiments each demonstrate that BSO-treatment leads to increased CD63 and TSG101 content of the EV pellets isolated from medium conditioned by P cells and are now presented in a new supplementary Fig. S5D.

6. The statement in the results, "Deletion of nSM2 did not influence transmigration toward a gradient of serum and fibronectin (not shown), indicating that cell-autonomous migratory behavior depended on neither nSM2 nor EV release from micrometastatic cells." The last half of that sentence does not take into account the likelihood that serum and/or fibronectin may rescue effects of EV loss and so is an overinterpretation of the (not shown) data. Also, data should be shown if it will be mentioned in the manuscript.

We have addressed this as follows:

We have now included these data in a new supplementary figure S8A. We have also modified the last half of the sentence interpreting these results thus: [...indicating that cell-autonomous **haptotactic/chemotactic** migratory behaviour **did not depend on sSM2-dependent EV** release from micrometastatic cells.] We feel that this interpretation is now appropriate.

7. For Fig 8, were the TIFs washed after the addition of the EVs or was the EV-TIF mixture added to the collagen plug? If the latter, how do you separate the effect of the EVs vs. the effects of the TIFs?

We have addressed this as follows:

Yes, TIFs were extensively washed and trypsinised following 72hr of EV-treatment and prior to using them for the contraction of collagen plugs. Additionally, EV pre-treated

TIFs were also washed (to remove EV-containing medium), trypsinised and re-plated prior to production of ECM for the experiments presented in Fig. 8D. We have now amended the appropriate methods sections to explain these protocol details more clearly.

8. What media was used is when the TIFs were pre-treated with small EVs? Also, how were the EV treatments done of the TIFs? Equal vesicle number? Equal protein? If it was done based on starting cell number, it's a bit less informative since the KO cells secrete fewer EVs.

We have addressed this as follows:

This information has now been added to the appropriate methods sections. Indeed, EV concentrations in the pre-treatment medium are corrected to those present in the condition/cell-exposed medium from which they were purified and, thus, are appropriate to starting/donor cell number. We believe that this is informative because it is providing information on the efficacy of the EV dose provided by a given number of donor cells.

9. The method for preparing de-cellularized ECM is missing. In addition, quantitation of the amount of de-cellularized ECM deposited by fibroblasts under the different small EV pre-treatment conditions (as well as in the absence of EVs) would provide support for the authors' conclusion that the micrometastatic cancer cell EVs are inducing the fibroblasts to deposit more ECM.

We have addressed this as follows:

We have now amended the appropriate methods section to include a description of ECM de-cellularisation, and the references for these approaches.

We have also quantified the fibronectin deposited by TIFs with and without pre-treatment with EVs from control and nSM2-CRISPR M cells. These data are presented in a new supplementary figure S11 and indicate that the amount and organisation of fibronectin deposited is not detectably altered by EV pre-treatment. We have also added the following text (see below) to the discussion to clarify this observation, and to indicate that the altered ability of ECM deposited by EV pre-treated fibroblasts to support increased cell migration is not due to increased fibronectin deposition:

[.....alters integrin trafficking in fibroblasts to encourage them to deposit a more pro-invasive ECM. However, we have measured $\alpha 5\beta 1$ integrin recycling and found this not to differ between fibroblasts treated with EVs from control and nSM2-CRISPR micrometastatic cells. **Moreover, EVs from micrometastatic cells do not influence the fibronectin filament length or organisation in the ECM deposited by fibroblasts (Fig. S10), further indicating that its pro-invasive nature is not a consequence of altered $\alpha 5\beta 1$ integrin dependent ECM deposition.** Thus, further work will be necessary.....]

10. The schematic summary in Fig 8 should be edited. Currently, it looks like nSMase2 controls exosome secretion by promoting the fusion event of MVB with the plasma membrane. nSMase2 is involved in MVB biogenesis, not the release step, as stated correctly in the Introduction.

We have addressed this as follows:

The schematic in Fig. 8 has now been amended to reflect this point.

11. The discussion is rather long and rambling, and the connections between the metabolic part of the paper and the EVs is not well made. Some references also seem to be missing, e.g. the authors mention that nSMase2-dependent EV production is not associated with release of mtDNA but without references. Torralba et al. has described the role of Rab27a and nSMase2 in mitochondrial function (doi:10.1038/s41467-018-05077-9) - perhaps this is relevant?

We have addressed this as follows:

We have now added the Torralba reference (and appropriate text) to the discussion. We have also added the data indicating that, although Rab27 commands release of mtDNA-containing EVs from P cells (as we have reported for other mammary cancer cell lines, such as MDA-MB-231), micrometastatic (M) cells release very little mtDNA in association with EVs. These data are now presented in a new supplementary figure S10.

12. Small EVs are not another name for exosomes but a term encompassing exosomes and other small-sized EVs. The sentences at the end of the 4th paragraph of Introduction are ambiguous.

We have addressed this as follows:

As mentioned previously, and in response to some of the other reviewers' comments, we have adjusted our text to no longer refer to small EVs (or exosomes). Now that we are not drawing distinctions between small and other EVs, and because we are no longer claiming that our data pertain to any particular sub-category of EVs, we feel that these statements are no longer incorrect.

13. Please show all data if they appear in the manuscript or do not mention results if they are listed as "not shown".

We have addressed this as follows:

We have now presented all data previously referred to as 'not shown'. This has necessitated the inclusion of many supplementary figures, but we feel that the manuscript is certainly much stronger for the inclusion of these data.

14. At least 3 biological replicates for the assay in Fig 1E should be performed.

We have addressed this as follows:

We have now performed 3 independent biological repeats for this experiment. The new data are presented in Fig. 1E. The data in Fig. 1E are represented by presenting a single average value for each experiment performed, and these are colour-coded so that the reader can assign values to the 3 individual experiments. The companion figure to this – supplementary figure S2C - now represents the overall variation in the data (conflating both the inter- and intra-assay variation) plotted as a violin plot.

15. In Fig 3C and 4B, what is a technical repeat? Why did the authors perform 3 technical repeats instead of biological repeats? Please describe what the limitations are in performing biological repeats.

We have addressed this as follows:

We have now performed 4 independent biological repeats for the experiments represented in figures 3C and 4B. These are represented by colour-coded dots each denoting the average values from each experiment. Technical variation (between intra-experimental replicates) is small and is now not represented.

16. Graphs in Fig S2 show only 3 data points unlike the replicates stated in the figure legend, n=5.

We have addressed this as follows:

This reviewer is correct. The N for these experiments is 3 and we have corrected this error.

17. How was the sEV amount for lipidomics normalized in Fig 6C?

We have addressed this as follows:

Lipid extracts of EVs for the lipidomics presented in Fig. 6C were normalised to the number of EV-releasing donor cells. This is now stated in the legend to this figure.

18. Were the CRISPR cells validated with diagnostic PCR of genomic DNA to distinguish parental from heterozygous from homozygous? In some cases it can be challenging to distinguish between heterozygous and homozygous KO, e.g. for the light Western blot shown for Rab27a in the M cells in Fig S4. If it's a heterozygous rather than homozygous KO in the M cells, that could account for the unique lack of effect of Rab27a in the M cells. And it looks like there is a whisper of Rab27a in the KO lane of that very faint blot. Since Rab27a and nSMase2 affect distinct parts of the exosome biogenesis (nSMase2) and secretion (Rab27a) pathway, one wouldn't really expect switching from one mechanism to another anyway.

We have addressed this as follows:

These CRISPR cells are not cloned, they were selected using puromycin and then used as the mixed population that results from this. We find that this obviates the problems associated with cloning cells and then needing to control for what can be, in our experience, a wide range of clonal variation. The resulting antibiotic-selected cells were then screened for reduced Rab GTPase expression by Western blotting. As these cells are, therefore, not 'knock-out' for Rab27s, we realised that it is inaccurate to refer to them as such. We have, therefore, modified our terminology of these lines and we now refer to them as CRISPR lines (not knockout) and talk about 'suppression' of Rab27s, rather than knockout, throughout the paper. We have also repeated the Western blots (now presented in a new supplementary figure S6C) several times to obtain a realistic view of the actual level of suppression of Rab27 expression.

19. In Fig 6, how was the ceramide intensity normalized, per cell, per area, per field of view, or something else? It is unclear what "n" in Fig 6A and 6E indicates. If the "n" doesn't indicate independent experiments, "n=5" sounds too small for the quantitation. Typically one would want many images each from 3 biological replicates.

We have addressed this as follows:

We have now repeated the experiment presented in Fig. 6A. The data are now represented by presenting a single average value for each experiment performed and these are colour-coded so that the reader can assign these values to the 3 individual experiments. We have not repeated figure 6E as we consider that the lipidomics presented is a better way to determine functional suppression of nSMs. We have also updated the 'immunofluorescence' methods section to more clearly describe the normalisation to phalloidin intensity (red channel) that was used for quantification of ceramide intensity (green channel).

20. Presenting all statistics available between each group will be helpful for readers to understand the results. For instance, is the difference in ceramide intensity between NTC and nSM1 or between nSM1 and nSM2 in Fig 6E significant? Also, in Fig S5A, is EV release by nSM1 KO in M cells significantly decreased compared to NTC control?

We have addressed this as follows:

We have now added statistics to illustrate any differences that are significant at $p < 0.05$ – as is the convention – throughout the paper. This has entailed adding – as this reviewer points out – statistics to Fig. 6E. The difference between nSM1 and NTC control in Fig. S5A (now Fig. S7A) is not significant at $p < 0.05$.

March 14, 2025

RE: JCB Manuscript #202405061R

Jim Norman
CRUK-Beatson Institute

Dear Prof. Norman:

Thank you for submitting your revised manuscript entitled "Metabolic adaptations of micrometastases alter EV production to generate invasive microenvironments". The reviewers are in broad agreement that the conclusions are more fully supported in the revised study. We would be happy to publish your paper in JCB pending final revisions necessary to meet our formatting guidelines (see details below). We appreciate your rigor in bringing these interesting findings to the JCB readership.

A. MANUSCRIPT ORGANIZATION AND FORMATTING:

Full guidelines are available on our Instructions for Authors page, <http://jcb.rupress.org/submission-guidelines#revised>. Submission of a paper that does not conform to JCB guidelines will delay the acceptance of your manuscript.

- 1) Text limits: Character count for Articles is < 40,000, not including spaces. Count includes abstract, introduction, results, discussion, and acknowledgments. Count does not include title page, figure legends, materials and methods, references, tables, or supplemental legends.
- 2) Figures limits: Articles may have up to 10 main figures and 5 supplemental figures/tables.
- 3) Figure formatting: Scale bars must be present on all microscopy images, including inset magnifications. Molecular weight or nucleic acid size markers must be included on all gel electrophoresis. Please avoid pairing red and green for images and graphs to ensure legibility for color-blind readers. If red and green are paired for images, please ensure that the particular red and green hues used in micrographs are distinctive with any of the colorblind types. If not, please modify colors accordingly or provide separate images of the individual channels.
- 4) Statistical analysis: Error bars on graphic representations of numerical data must be clearly described in the figure legend. The number of independent data points (n) represented in a graph must be indicated in the legend. Statistical methods should be explained in full in the materials and methods. For figures presenting pooled data the statistical measure should be defined in the figure legends. Please also be sure to indicate the statistical tests used in each of your experiments (either in the figure legend itself or in a separate methods section) as well as the parameters of the test (for example, if you ran a t-test, please indicate if it was one- or two-sided, etc.). Also, if you used parametric tests, please indicate if the data distribution was tested for normality (and if so, how). If not, you must state something to the effect that "Data distribution was assumed to be normal but this was not formally tested."
- 5) Abstract and title: The abstract should be no longer than 160 words and should communicate the significance of the paper for a general audience. The title should be less than 100 characters including spaces. Make the title concise but accessible to a general readership.
** We suggesting changing this sentence in the abstract: "The combination of these two metabolic adaptations alters extracellular vesicle (EV) production to drive generation of an invasive microenvironment."
We suggest a clearer phrasing: "The combination of these two metabolic adaptations alters extracellular vesicle (EV) production to render the microenvironment more permissive for invasion."
- 6) Materials and methods: Should be comprehensive and not simply reference a previous publication for details on how an experiment was performed. Please provide full descriptions in the text for readers who may not have access to referenced manuscripts. We also provide a report from SciScore and an associate score, which we encourage you to use as a means of evaluating and improving the methods section.
- 7) Please be sure to provide the sequences for all of your primers/oligos, plasmids, and RNAi constructs in the materials and methods. You must also indicate in the methods the source, species, and catalog numbers (where appropriate) for all of your antibodies. Please also indicate the acquisition and quantification methods for immunoblotting/western blots.
- 8) Microscope image acquisition: The following information must be provided about the acquisition and processing of images:
 - a. Make and model of microscope
 - b. Type, magnification, and numerical aperture of the objective lenses

- c. Temperature
- d. Imaging medium
- e. Fluorochromes
- f. Camera make and model
- g. Acquisition software
- h. Any software used for image processing subsequent to data acquisition. Please include details and types of operations involved (e.g., type of deconvolution, 3D reconstitutions, surface or volume rendering, gamma adjustments, etc.).

10) Supplemental materials: There are strict limits on the allowable amount of supplemental data. Articles may have up to 5 supplemental figures. Please also note that tables, like figures, should be provided as individual, editable files. A summary of all supplemental material should appear at the end of the Materials and methods section.

13) ORCID IDs: ORCID IDs are unique identifiers allowing researchers to create a record of their various scholarly contributions in a single place. At resubmission of your final files, please provide an ORCID ID for all authors.

15) A data availability statement is required for all research article submissions. The statement should address all data underlying the research presented in the manuscript. Please visit the JCB instructions for authors for guidelines and examples of statements at (<https://rupress.org/jcb/pages/editorial-policies#data-availability-statement>).

** We strongly encourage deposition of all mass spectrometry data in the ProteomeXchange Consortium, for instance via the PRIDE partner repository.

Please note that JCB requires authors to submit Source Data used to generate figures containing gels and Western blots with all revised manuscripts. This Source Data consists of fully uncropped and unprocessed images for each gel/blot displayed in the main and supplemental figures. Since your paper includes cropped gel and/or blot images, please be sure to provide one Source Data file for each figure that contains gels and/or blots along with your revised manuscript files. File names for Source Data figures should be alphanumeric without any spaces or special characters (i.e., SourceDataF#, where F# refers to the associated main figure number or SourceDataFS# for those associated with Supplementary figures). The lanes of the gels/blots should be labeled as they are in the associated figure, the place where cropping was applied should be marked (with a box), and molecular weight/size standards should be labeled wherever possible. Source Data files will be directly linked to specific figures in the published article.

WHEN APPROPRIATE: The source code for all custom computational methods published in JCB must be made freely available as supplemental material hosted at www.jcb.org. Please contact the JCB Editorial Office to find out how to submit your custom macros, code for custom algorithms, etc. Generally, these are provided as raw code in a .txt file or as other file types in a .zip file. Please also include a one-sentence summary of each file in the Online Supplemental Material paragraph of your manuscript.

B. FINAL FILES:

Thank you for your attention to these final processing requirements. Please revise and format the manuscript and upload materials within 7 days. If you need an extension for whatever reason, please let us know and we can work with you to determine a suitable revision period.

Thank you for this interesting contribution, we look forward to publishing your paper in Journal of Cell Biology.

Sincerely,

Brendan Manning
Monitoring Editor
Journal of Cell Biology

Tim Fessenden
Scientific Editor
Journal of Cell Biology

Reviewer #1 (Comments to the Authors (Required)):

The revisions have addressed concerns. Great work.

Reviewer #2 (Comments to the Authors (Required)):

The authors have now sufficiently addressed the reviewer comments. The inclusion of metabolic tracing experiments in Fig 4C greatly enhance the conclusions in this work.

Reviewer #3 (Comments to the Authors (Required)):

The authors have appropriately addressed my suggestions. The data are much tighter now, and the manuscript is ready for publication.

A. Manuscript Organisation and formatting

1) Text limits: Character count for Articles is < 40,000, not including spaces. Count includes abstract, introduction, results, discussion, and acknowledgments. Count does not include title page, figure legends, materials and methods, references, tables, or supplemental legends.

The character count for the paper is 39,428 not including spaces, but including abstract, introduction, results, discussion and acknowledgements.

2) Figures limits: Articles may have up to 10 main figures and 5 supplemental figures/tables.

There are 8 main figures. We have reduced the number of supplementary figures from 11 to 5 as recommended. The number of supplementary figures in the previous submission was high as the reviewers requested to see a substantial amount of data which was referred to as 'not shown'. Although we had previously agreed to discuss an increased supplementary figure allowance with Dr Fessenden, we have now managed to re-structure the figures to keep the supplementary figures within the allowance of 5. We have achieved this without resorting to data not shown, but it has necessitated us to be more selective with showing exemplar data. For instance, we no longer display nanoparticle tracking plots for both the P, FP & M and the P', FP' & M' series, but still include quantification of these as bar graphs. Similarly, we have reduced the presentation of exemplar images for the organotypic invasion assays but have included quantification of all of these as bar graphs.

3) Figure formatting: Scale bars must be present on all microscopy images, including inset magnifications. Molecular weight or nucleic acid size markers must be included on all gel electrophoresis. Please avoid pairing red and green for images and graphs to ensure legibility for color-blind readers. If red and green are paired for images, please ensure that the particular red and green hues used in micrographs are distinctive with any of the colorblind types. If not, please modify colors accordingly or provide separate images of the individual channels.

We have provided scale bars for all images. However, when these are part of a matched series with identical magnification, such as the H&E and RNA scope images in Fig. 3E, to avoid cluttering the images, we have only added scale bars to the upper pairs. Similarly, in Fig. 1D, there are no scale bars as the images are calibrated by the scale on the left-hand side. Secondly, and as requested, we have modified the immunofluorescence images in Fig. 6A to deploy colour-blind types.

4) Statistical analysis: Error bars on graphic representations of numerical data must be clearly described in the figure legend. The number of independent data points (n) represented in a graph must be indicated in the legend. Statistical methods should be explained in full in the materials and methods. For figures presenting pooled data the statistical measure should be defined in the figure legends. Please also be sure to indicate the statistical tests used in each of your experiments (either in the figure legend itself or in a separate methods section) as well as the parameters of the test

(for example, if you ran a t-test, please indicate if it was one- or two-sided, etc.). Also, if you used parametric tests, please indicate if the data distribution was tested for normality (and if so, how). If not, you must state something to the effect that "Data distribution was assumed to be normal but this was not formally tested."

All scale bars and statistical tests are described in the figure legends. T-tests are specified as paired or unpaired as appropriate. We have now included a statement describing our assumptions re normal data distribution in the 'statistical analyses' section in the methods.

5) Abstract and title: The abstract should be no longer than 160 words and should communicate the significance of the paper for a general audience. The title should be less than 100 characters including spaces. Make the title concise but accessible to a general readership.

** We suggesting changing this sentence in the abstract: "The combination of these two metabolic adaptations alters extracellular vesicle (EV) production to drive generation of an invasive microenvironment." We suggest a clearer phrasing: "The combination of these two metabolic adaptations alters extracellular vesicle (EV) production to render the microenvironment more permissive for invasion."

The title is 99 characters including spaces. We have modified the abstract as recommended.

6) Materials and methods: Should be comprehensive and not simply reference a previous publication for details on how an experiment was performed. Please provide full descriptions in the text for readers who may not have access to referenced manuscripts. We also provide a report from SciScore and an associate score, which we encourage you to use as a means of evaluating and improving the methods section.

We have made a substantial number of additions and modifications to the methods section to ensure comprehensive and clear descriptions of ALL methods. Hopefully the SciScore will be good, as the level of detail, we feel, is sufficient for others to reproduce the experimentation. All textual modifications are added in red type.

7) Please be sure to provide the sequences for all of your primers/oligos, plasmids, and RNAi constructs in the materials and methods. You must also indicate in the methods the source, species, and catalog numbers (where appropriate) for all of your antibodies. Please also indicate the acquisition and quantification methods for immunoblotting/western blots.

These details have now been added to the methods

8) Microscope image acquisition: The following information must be provided about the acquisition and processing of images:

- a. Make and model of microscope
- b. Type, magnification, and numerical aperture of the objective lenses
- c. Temperature
- d. Imaging medium

e. Fluorochromes

f. Camera make and model

g. Acquisition software

h. Any software used for image processing subsequent to data acquisition. Please include details and types of operations involved (e.g., type of deconvolution, 3D reconstitutions, surface or volume rendering, gamma adjustments, etc.).

We have included all this information in the methods section. N.B. We have not included a separate subsection for these details but have added them to the methods subsections as appropriate.

All references are cited formatted according to JCB guidelines

10) Supplemental materials: There are strict limits on the allowable amount of supplemental data. Articles may have up to 5 supplemental figures. Please also note that tables, like figures, should be provided as individual, editable files. A summary of all supplemental material should appear at the end of the Materials and methods section.

We have reduced the number of supplementary figures to 5 as requested and included a summary after the methods section.

A 20-word eTOC summary is now included interposed between the supplementary figure legends and the conflict-of-interest statement.

We have now included a conflict-of-interest statement following the eTOC.

13) ORCID IDs: ORCID IDs are unique identifiers allowing researchers to create a record of their various scholarly contributions in a single place. At resubmission of your final files, please provide an ORCID ID for all authors.

We have listed ORCID IDs for all authors that have them in a section following the conflict-of-interest statement.

14) A separate author contribution section following the Acknowledgments. All authors should be mentioned and designated by their full names. We encourage use

of the CRediT nomenclature.

Our acknowledgements section comprises an author contributions subsection which is consistent with the CRediT system.

15) A data availability statement is required for all research article submissions. The statement should address all data underlying the research presented in the manuscript. Please visit the JCB instructions for authors for guidelines and examples of statements at (<https://rupress.org/jcb/pages/editorial-policies#data-availability-statement>).

** We strongly encourage deposition of all mass spectrometry data in the ProteomeXchange Consortium, for instance via the PRIDE partner repository. We have included a data availability statement in a subsection of the acknowledgement section. Mass spectrometry data been deposited in the Massive repository and this is already on-line.

16) Source data files

We have uploaded all uncropped Western blots as individual PDFs and these are named as instructed; (i.e., SourceDataF#, where F# refers to the associated main figure number or SourceDataFS# for those associated with Supplementary figures).